# Foundations of Multivariate Distributional Reinforcement Learning

**Harley Wiltzer**
Mila — Québec AI Institute
McGill University
harley.wiltzer@mail.mcgill.ca

**Jesse Farebrother**
Mila — Québec AI Institute
McGill Unversity
jfarebro@cs.mcgill.ca

**Arthur Gretton**
Google DeepMind
Gatsby Unit, University College London
gretton@google.com

**Mark Rowland**
Google DeepMind
markrowland@google.com

## Abstract

In reinforcement learning (RL), the consideration of multivariate reward signals has led to fundamental advancements in multi-objective decision-making, transfer learning, and representation learning. This work introduces the first oracle-free and computationally-tractable algorithms for provably convergent multivariate *distributional* dynamic programming and temporal difference learning. Our convergence rates match the familiar rates in the scalar reward setting, and additionally provide new insights into the fidelity of approximate return distribution representations as a function of the reward dimension. Surprisingly, when the reward dimension is larger than 1, we show that standard analysis of categorical TD learning fails, which we resolve with a novel projection onto the space of mass-1 signed measures. Finally, with the aid of our technical results and simulations, we identify tradeoffs between distribution representations that influence the performance of multivariate distributional RL in practice.

## 1 Introduction

Distributional reinforcement learning [DRL; MSK+10, BDM17b, BDR23] focuses on the idea of learning probability distributions of an agent's random return, rather than the classical approach of learning only its mean. This has been highly effective in combination with deep reinforcement learning [YZL+19, BCC+20, WBK+22], and DRL has found applications in risk-sensitive decision making [LM22, KEF23], neuroscience [DKNU+20], and multi-agent settings [ROH+21, SLL21].

In general, research in distributional reinforcement learning has focused on the classical setting of a scalar reward function. However, prior non-distributional approaches to multi-objective RL [RVWD13, HRB+22] and transfer learning [BDM+17a, BHB+20] model value functions of multivariate cumulants,[1] rather than a scalar reward. Having learnt such a multivariate value function, it is then possible to perform zero-shot evaluation and policy improvement for any scalar reward signal contained in the span of the coordinates of the multivariate cumulants, opening up a variety of applications in transfer learning, and multi-objective and constrained RL.

*Multivariate distributional RL* combines these two ideas, and aims to learn the full probability distribution of returns given a multivariate cumulant function. Successfully learning the multivariate

---

[1]Cumulants refer to accumulated quantities in RL (e.g., rewards or multivariate rewards)—not to be confused with statistical cumulants.

38th Conference on Neural Information Processing Systems (NeurIPS 2024).

reward distribution opens up a variety of unique possibilities, such as zero-shot return distribution estimation [WFG$^+$24] and risk-sensitive policy improvement [CZZ$^+$24].

Pioneering works have already proposed algorithms for multivariate distributional RL. While these works all demonstrate benefits from the proposed algorithmic approaches, each suffers from separate drawbacks, such as not modelling the full joint distribution [GBSL21], lacking theoretical guarantees [FSMT19, ZCZ$^+$21], or requiring a maximum-likelihood optimisation oracle for implementation [WUS23]. Concurrently, the work of [LK24] analyzed algorithms for DRL with Banach-space-valued rewards, and provided convergence guarantees for dynamic programming with non-parametric (intractable) distribution models.

Our central contribution in this paper is to propose algorithms for dynamic programming and temporal-difference learning in multivariate DRL which are computationally tractable and theoretically justified, with convergence guarantees. We show that reward dimensions strictly larger than 1 introduce new computational and statistical challenges. To resolve these challenges, we introduce multiple novel algorithmic techniques, including a randomized dynamic programming operator for efficiently approximating projected updates with high probability, and a novel TD-learning algorithm operating on mass-1 *signed* measures. These new techniques recover existing bounds even in the scalar reward case, and provide new insights into the behavior of distributional RL algorithms as a function of the reward dimension.

## 2 Background

We consider a Markov decision process with Polish state space $\mathcal{X}$, action space $\mathcal{A}$, transition kernel $P : \mathcal{X} \times \mathcal{A} \to \mathscr{P}(\mathcal{X})$, and discount factor $\gamma \in [0, 1)$. Unlike the standard RL setting, we consider a vector-valued reward function $r : \mathcal{X} \to [0, R_{\max}]^d$, as in the literature on successor features [BDM$^+$17a]. Given a policy $\pi : \mathcal{X} \to \mathcal{P}(\mathcal{A})$, we write the policy-conditioned transition kernel $P^\pi(\cdot \mid x) = \int P(\cdot \mid x, a)\pi(\mathrm{d}a \mid x)$.

**Multi-variate return distributions.** We write $(X_t)_{t \geq 0}$ for a trajectory generated by setting $X_0 = x$, and for each $t \geq 0$, $X_{t+1} \sim P^\pi(\cdot|X_t)$. The return obtained along this trajectory is then defined by $G^\pi(x) = \sum_{t=0}^\infty \gamma^t r(X_t)$, and the *(multi-)return distribution function* is $\eta^\pi(x) = \mathrm{Law}\,(G^\pi(x))$.

**Zero-shot evaluation.** An intriguing prospect of estimating multivariate return distributions is the ability to predict (scalar) return distributions for any reward function in the span of the cumulants. Indeed, [ZCZ$^+$21, CZZ$^+$24] show that for any reward function $\tilde{r} : x \mapsto \langle r(x), w \rangle$ for some $w \in \mathbf{R}^d$, $\langle G^\pi(x), w \rangle =_{\mathrm{law}} \sum_{t \geq 0} \gamma^t \tilde{r}(X_t)$ for $X_0 = x$. Likewise, one might consider $r(x) = \delta_x$, in which case $G^\pi(x)$ corresponds to the per-trajectory discounted state visitation measure, and [WFG$^+$24] demonstrated methods for learning the distribution of $G^\pi$ to infer the return distribution for any bounded deterministic reward function.

**Multivariate distributional Bellman equation.** It was shown in [ZCZ$^+$21] that multi-return distributions obey a distributional Bellman equation, similar to the scalar case [MSK$^+$10, BDM17b], and defines the multivariate distributional Bellman operator $\mathcal{T}^\pi : \mathcal{P}(\mathbf{R}^d)^{\mathcal{X}} \to \mathcal{P}(\mathbf{R}^d)^{\mathcal{X}}$ by

$$(\mathcal{T}^\pi \eta)(x) = \mathop{\mathbf{E}}_{X' \sim P^\pi(\cdot|x)} \left[ (\mathrm{b}_{r(x),\gamma})_\sharp \eta(X') \right], \tag{1}$$

where $\mathrm{b}_{y,\gamma}(z) = y + \gamma z$ and $f_\sharp \mu = \mu \circ f^{-1}$ is the *pushforward* of a measure $\mu$ through a measurable function $f$. In particular, [ZCZ$^+$21] showed that $\eta^\pi$ satisfies the *multi-variate distributional Bellman equation* $\mathcal{T}^\pi \eta^\pi = \eta^\pi$, and that $\mathcal{T}^\pi$ is a $\gamma$-contraction in $\overline{W}_p$, where $\overline{W}_p(\eta_1, \eta_2) = \sup_{x \in \mathcal{X}} W_p(\eta_1(x), \eta_2(x))$ and $W_p$ is the $p$-Wasserstein metric [Vil09]. This suggests a convergent scheme for approximating $\eta^\pi$ in $\overline{W}_p$ by *distributional dynamic programming*, that is, computing the iterates $\eta_{k+1} = \mathcal{T}^\pi \eta_k$, following Banach's fixed point theorem.

**Approximating multivariate return distributions.** In practice, however, the iterates $\eta_{k+1} = \mathcal{T}^\pi \eta_k$ cannot be computed efficiently, because the size of the support of $\eta_k$ may increase exponentially with $k$. A variety of alternative approaches that aim to circumvent this computational difficulty have been considered [FSMT19, ZCZ$^+$21, WUS23]. Many of these approaches have proven effective in combination with deep reinforcement learning, though as tabular algorithms, either lack theoretical guarantees, or rely on oracles for solving possibly intractable optimisation problems. A more complete account of multivariate DRL is given in Appendix A. A central motivation of our work

is the development of computationally-tractable algorithms for multivariate distributional RL with theoretically guarantees.

**Maximum mean discrepancies.** A core tool in the development of our proposed algorithms, as well as some prior work [NTGV20, ZCZ$^+$21], is the notion of distance over probability distributions given by maximum mean discrepancies [GBR$^+$12, MMD]. A maximum mean discrepancy $\mathrm{MMD}_\kappa :$ $\mathscr{P}(\mathcal{Y}) \times \mathscr{P}(\mathcal{Y}) \to \mathbf{R}_+$ assigns a notion of distance to pairs of probability distributions, and is parametrised via a choice of kernel $\kappa : \mathcal{Y} \times \mathcal{Y} \to \mathbf{R}$, defined by

$$\mathrm{MMD}_\kappa(p,q) = \mathbb{E}_{(Y_1,Y_2)\sim p\otimes p}[\kappa(Y_1,Y_2)] - 2\mathbb{E}_{(Y,Z)\sim p\otimes q}[\kappa(Y,Z)] + \mathbb{E}_{(Z_1,Z_2)\sim q\otimes q}[\kappa(Z_1,Z_2)].$$

A useful alternative perspective on MMD is that the choice of kernel $\kappa$ induces a reproducing kernel Hilbert space (RKHS) of functions $\mathcal{H}$, namely the closure of the span of functions of the form $z \mapsto \kappa(y,z)$ for each $y \in \mathcal{Y}$, with respect to the norm $\|\cdot\|_\mathcal{H}$ induced by the inner product $\langle \kappa(y_1,\cdot), \kappa(y_2,\cdot)\rangle = \kappa(y_1,y_2)$. With this interpretation, $\mathrm{MMD}_\kappa(p,q)$ is equal to $\|\mu_p - \mu_q\|_\mathcal{H}$, where $\mu_p = \int_\mathcal{Y} \kappa(\cdot,y)p(\mathrm{d}y) \in \mathcal{H}$ is the *mean embedding* of $p$ (similarly for $\mu_q$). When $p \mapsto \mu_p$ is injective, the kernel $\kappa$ is called *characteristic*, and $\mathrm{MMD}_\kappa$ is then a proper metric on $\mathcal{P}(\mathcal{Y})$ [GBR$^+$12]. In the remainder of this work, we will assume that all spaces of measures will be over compact sets $\mathcal{Y}$; thus with continuous kernels, we are ensured that distances between probability measures are bounded. When comparing return distributions, this is achieved by asserting that rewards are bounded.

We conclude this section by recalling a particular family of kernels, introduced in [SSGF13], that will be particularly useful for our analysis. These are the kernels induced by semimetrics.

**Definition 1.** *Let $\mathcal{Y}$ be a nonempty set, and consider a function $\rho : \mathcal{Y} \times \mathcal{Y} \to \mathbf{R}_+$. Then $\rho$ is called a* semimetric *if it is symmetric and $\rho(y_1,y_2) = 0 \iff y_1 = y_2$. Additionally, $\rho$ is said to have* strong negative type *if $\int \rho \, \mathrm{d}([p-q] \times [p-q]) < 0$ whenever $p,q \in \mathcal{P}(\mathcal{Y})$ with $p \neq q$.*

Notably, certain semimetrics naturally induce characteristic kernels and probability metrics.

**Theorem 1** ([SSGF13, Proposition 29]). *Let $\rho$ be a semimetric on a space $\mathcal{Y}$ have* strong negative type*, in the sense that $\int \rho \, \mathrm{d}([p-q] \times [p-q]) < 0$ whenever $p \neq q$ are probability measures on a compact set $\mathcal{Y}$. Moreover, let $\kappa : \mathcal{Y} \times \mathcal{Y} \to \mathbf{R}$ denote the kernel induced by $\rho$, that is*

$$\kappa(y_1,y_2) = \frac{1}{2}(\rho(y_1,y_0) + \rho(y_2,y_0) - \rho(y_1,y_2))$$

*for some $y_0 \in \mathcal{Y}$. Then $\kappa$ is characteristic, so $\mathrm{MMD}_\kappa$ is a metric.*

**Remark 1.** *An important class of semimetrics are the functions $\rho_\alpha : \mathbf{R}^d \times \mathbf{R}^d \to \mathbf{R}_+$ given by $\rho_\alpha(y_1,y_2) = \|y_1 - y_2\|_2^\alpha$ for $\alpha \in (0,2)$. It is known that these semimetrics have strong negative type, and thus the kernels $\kappa_\alpha$ induced by $\rho_\alpha$ are characteristic [SR13, SSGF13]. The resulting metric $\mathrm{MMD}_{\kappa_\alpha}$ is known as the* energy distance.

## 3 Multivariate Distributional Dynamic Programming

To warm up, we begin by demonstrating that indeed the (multivariate) distributional Bellman operator is contractive in a supremal form $\overline{\mathrm{MMD}_\kappa}$ of MMD, given by $\overline{\mathrm{MMD}_\kappa}(\eta_1,\eta_2) = \sup_{x\in\mathcal{X}} \mathrm{MMD}_\kappa(\eta_1(x),\eta_2(x))$, for a particular class of kernels $\kappa$. Our first theorem generalizes the analogous results of [NTGV20] in the scalar case to multivariate cumulants. The proof of Theorem 2, as well as proofs of all remaining results, are deferred to Appendix B.

**Theorem 2** (Convergent MMD dynamic programming for the multi-return distribution function). *Let $\kappa$ be a kernel induced by a semimetric $\rho$ on $[0, (1-\gamma)^{-1}R_{\max}]^d$ with strong negative type, satisfying*

1. ***Shift-invariance**. For any $z \in \mathbf{R}^d$, $\rho(z+y_1, z+y_2) = \rho(y_1,y_2)$.*

2. ***Homogeneity**. For any $\gamma \in [0,1)$, there exists $c > 0$ for which $\rho(\gamma y_1, \gamma y_2) = \gamma^c \rho(y_1,y_2)$.*

*Consider the sequence $\{\eta_k\}_{k=1}^\infty$ given by $\eta_{k+1} = \mathcal{T}^\pi \eta_k$. Then $\eta_k \to \eta^\pi$ at a geometric rate of $\gamma^{c/2}$ in $\overline{\mathrm{MMD}_\kappa}$, as long as $\overline{\mathrm{MMD}_\kappa}(\eta_0, \eta^\pi) \leq C < \infty$.*

Notably, the energy distance kernels $\kappa_\alpha$ satisfy the conditions of Theorem 2, and $\rho_\alpha(\gamma y_1, \gamma y_2) \leq \gamma^\alpha \rho_\alpha(y_1,y_2)$ by the homogeneity of the Euclidean norm, so $\mathcal{T}^\pi$ is a $\gamma^{\alpha/2}$-contraction in the energy distances. This generalizes the analogous result of [NTGV20] in the one-dimensional case.

While Theorem 2 illustrates a method for approximating $\eta^\pi$ in MMD, it leaves a lot to be desired. Firstly, even in tabular MDPs, just as in the case of scalar distributional RL, return distribution functions have infinitely many degrees of freedom, precluding a tractable exact representation. As such, it will be necessary to study approximate, finite parameterizations of the return distribution functions, requiring more careful convergence analysis. Moreover, in RL it is generally assumed that the transition kernel and reward function are not known analytically—we only have access to sampled state transitions and cumulants. Thus, $\mathcal{T}^\pi$ cannot be represented or computed exactly, and instead we must study algorithms for approximating $\eta^\pi$ from samples. We provide algorithms for resolving both of these concerns—the former in Section 5 and the latter in Section 6—where we illustrate the difficulties that arise once the cumulant dimension exceeds unity.

## 4 Particle-Based Multivariate Distributional Dynamic Programming

Our first algorithmic contribution is inspired by the empirically successful *equally-weighted particle* (EWP) representations of multivariate return distributions employed by [ZCZ+21].

**Temporal-difference learning with EWP representations.** EWP representations, expressed by the class $\mathscr{C}_{\mathrm{EWP},m}$, are defined by

$$\mathscr{C}_{\mathrm{EWP},m} = (\mathscr{C}^\circ_{\mathrm{EWP},m})^{\mathcal{X}}, \qquad \mathscr{C}^\circ_{\mathrm{EWP},m} = \left\{ \frac{1}{m} \sum_{i=1}^m \delta_{\theta_i} \; : \; \theta_i \in \mathbf{R}^d \right\}. \tag{2}$$

For simplicity, we consider the case here where at each state $x$, the multi-return distribution is approximated by $N(x) = m$ atoms. We can represent $\eta \in \mathscr{C}_{\mathrm{EWP},m}$ by $\eta(x) = \frac{1}{m} \sum_{i=1}^m \delta_{\theta_i(x)}$ for $\theta_i : \mathcal{X} \to \mathbf{R}^d$. The work of [ZCZ+21] introduced a TD-learning algorithm for learning a $\mathscr{C}_{\mathrm{EWP},m}$ representation of $\eta^\pi$, computing iterates of the particles $(\theta_i^{(k)})_{i=1}^m$ according to

$$\theta_i^{(k+1)}(x) = \theta_i^{(k)}(x) - \lambda_k \nabla_{\theta_i(x)} \mathrm{MMD}_\kappa^2 \left( \frac{1}{m} \sum_{i=j}^m \delta_{\theta_j^{(k)}(x)}, \frac{1}{m} \sum_{j=1}^m \delta_{r(x) + \gamma \overline{\theta}_j^{(k)}(X')} \right) \tag{3}$$

for step sizes $(\lambda_k)_{k \geq 0}$ and sampled next states $X' \sim P^\pi(\cdot \mid x)$, where $\overline{\theta} = \texttt{stop-gradient}(\theta^{(k)})$ is a copy of $\theta^{(k)}$ that does not propagate gradients. Despite the empirical success of this method in combination with deep learning, no convergence analysis has been established, owing to the nonconvexity of the MMD objective with respect to the particle locations. In this section we aim to understand to what extent analysis is possible for dynamic programming and temporal-difference learning algorithms based on the EWP representations in Equation (2).

**Dynamic programming with EWP representations.** As is often the case in approximate distributional dynamic programming [RBD+18, RMA+24], we have $\mathcal{T}^\pi \mathscr{C}_{\mathrm{EWP},m} \not\subset \mathscr{C}_{\mathrm{EWP},m}$; in words, the distributional Bellman operator does not map EWP representations to themselves. Concretely, as long as there exists a state $x$ at which the support of $P^\pi(\cdot \mid x)$ is not a singleton, $(\mathcal{T}^\pi \eta)(x)$ will consist of more than $m$ atoms even when $\eta \in \mathscr{C}_{\mathrm{EWP},m}$; and secondly, if $P(\cdot \mid x)$ is not uniform, $(\mathcal{T}^\pi \eta)(x)$ will not consist of equally-weighted particles.

Consequently, to obtain a DP algorithm over EWP representations, we must consider a *projected* operator of the form $\Pi_{\mathrm{EWP}} \mathcal{T}^\pi$, for a projection $\Pi_{\mathrm{EWP}} : \mathcal{P}(\mathbf{R}^d)^{\mathcal{X}} \to \mathscr{C}_{\mathrm{EWP},m}$. A natural choice for this projection is the operator that minimizes the MMD of each multi-return distribution in $\mathscr{C}_{\mathrm{EWP},m}$,

$$(\Pi^m_{\mathrm{EWP},\kappa} \eta)(x) \in \underset{p \in \mathscr{C}^\circ_{\mathrm{EWP},m}}{\mathrm{argmin}} \; \mathrm{MMD}_\kappa(p, \eta(x)). \tag{4}$$

Unfortunately, even in the scalar-reward ($d = 1$) case, the operator $\Pi^m_{\mathrm{EWP},\kappa}$ is problematic; $(\Pi^m_{\mathrm{EWP},\kappa} \eta)(x)$ is not uniquely defined, and $\Pi^m_{\mathrm{EWP},\kappa}$ is not a non-expansion in $\overline{\mathrm{MMD}_\kappa}$ [LB22, RMA+24]. These pathologies present significant complications when analyzing even the convergence of dynamic programming routines for learning an EWP representation of the multi-return distribution — in particular, it is not even clear that $\Pi^m_{\mathrm{EWP},\kappa} \mathcal{T}^\pi$ has a fixed point (let alone a unique one). Another complication arises due to the computational difficulty of computing the projection (4): even in the case where $\eta(x)$ has finite support for each state $x$, the projection $(\Pi^m_{\mathrm{EWP},\kappa} \eta)(x)$ is very similar to clustering, which can be intractable to compute exactly for large $m$ [She21]. Thus, the argmin projection in Equation (4) cannot be used directly to obtain a tractable DP algorithm.

**Randomised dynamic programming.** Towards this end, we introduce a tractable *randomized dynamic programming* algorithm for the EWP representation, by using a randomized proxy $\text{BootProj}_{\kappa,m}^{\pi}$ for $\Pi_{\kappa,m}\mathcal{T}^{\pi}$, that produces accurate return distribution estimates with high probability. The method produces the following iterates,

$$\eta_{k+1}(x) = \text{BootProj}_{\kappa,m}^{\pi}\eta_k(x) := \frac{1}{m}\sum_{i=1}^{m}\delta_{r(x)+\gamma Z_i}, \qquad Z_i \sim \eta_k(X_i),\ X_i \overset{\text{iid}}{\sim} P^{\pi}(\cdot \mid x) \quad (5)$$

A similar algorithm for categorical representations was discussed in concurrent work [LK24] without convergence analysis.

The intuition is that, particularly for large $m$, the Monte Carlo error associated with the sample-based approximation to $(\mathcal{T}^{\pi}\eta)(x)$ is small, and we can therefore expect the DP process, though randomised, to be accurate with high probability. This is summarised by a key theoretical result of this section; our proof of this result in the appendix provides a general approach to proving convergence for algorithms using arbitrary accurate approximations to (4) that we expect to be useful in future work.

**Theorem 3.** *Consider a kernel $\kappa$ induced by the semimetric $\rho(x,y) = \|x-y\|_2^{\alpha}$ with $\alpha \in (0,2)$, and suppose rewards are bounded in each dimension within $[0, R_{\max}]$. For any $\eta_0$ such that $\overline{\text{MMD}}_{\kappa}(\eta_0, \eta^{\pi}) \leq D < \infty$, and any $\delta > 0$, for the sequence $(\eta_k)_{k\geq 0}$ defined in Equation (5), with probability at least $1 - \delta$ we have*

$$\overline{\text{MMD}}_{\kappa}(\eta_K, \eta^{\pi}) \in \widetilde{O}\left(\frac{d^{\alpha/2}R_{\max}^{\alpha}}{(1-\gamma^{\alpha/2})(1-\gamma)^{\alpha}\sqrt{m}}\log\left(\frac{|\mathcal{X}|\delta^{-1}}{\log\gamma^{-\alpha}}\right)\right).$$

*where $\eta_{k+1} = \text{BootProj}_{\kappa,m}^{\pi}\eta_k$ and $K = \lceil\frac{\log m}{\log\gamma^{-\alpha}}\rceil$, and where $\widetilde{O}$ omits logarithmic factors in $m$.*

This shows that our novel randomised DP algorithm with EWP representations can tractably compute accurate approximations to the true multivariate return distributions, with only polynomial dependence on the dimension $d$. Appendix C illustrates explicitly how this procedure is more memory efficient than unprojected EWP dynamic programming. However, the guarantees associated with this algorithm hold only in high probability, and are weaker than the pointwise convergence guarantees of one-dimensional distributional DP algorithms [RBD+18, RMA+24, BDR23]. Consequently, these guarantees do not provide a clear understanding of the EWP-TD method described at the beginning of this section. However, in the sequel, we introduce DP and TD algorithms based on *categorical representations*, for which we derive dynamic programming and TD-learning convergence bounds.

The proof of Theorem 3 is hinges on the following proposition, which demonstrates that convergence of projected EWP dynamic programming is controlled by how far return distributions are transported under the projection map.

**Proposition 1** (Convergence of EWP Dynamic Programming)**.** *Consider a kernel satisfying the hypotheses of Theorem 2, suppose rewards are globally bounded in each dimension in $[0, R_{\max}]$, and let $\{\Pi_{\kappa,m}^{(k)}\}_{k\geq 0}$ be a sequence of maps $\Pi : \mathcal{P}([0, (1-\gamma)^{-1}R_{\max}]^d)^{\mathcal{X}} \rightarrow \mathscr{C}_{\text{EWP},m}$ satisfying*

$$\text{MMD}_{\kappa}((\Pi_{\kappa,m}^{(k)}\eta)(x), \eta(x)) \leq f(d,m) < \infty \qquad \forall k \geq 0. \quad (6)$$

*Then the iterates $(\eta_k)_{k\geq 0}$ given by $\eta_{k+1} = \Pi_{\kappa,m}^{(k)}\mathcal{T}^{\pi}\eta_k$ with $\overline{\text{MMD}}_{\kappa}(\eta_0, \eta^{\pi}) = D < \infty$ converge to a set $\boldsymbol{\eta}_{\text{EWP},\kappa}^m \subset \overline{B}(\eta^{\pi}, (1-\gamma^{c/2})^{-1}f(d,m))$ in $\overline{\text{MMD}}_{\kappa}$, where $\overline{B}$ denotes the closed ball in $\overline{\text{MMD}}_{\kappa}$,*

$$\overline{B}(\eta, R) \triangleq \left\{\eta' \in \mathcal{P}(\mathbf{R}^d)^{\mathcal{X}} : \overline{\text{MMD}}_{\kappa}(\eta, \eta') \leq R\right\}.$$

As an immediate corollary of Proposition 1 and Theorem 3, we can derive an error rate for projected dynamic programming with $\Pi_{\text{EWP},\kappa}^m$ as well.

**Corollary 1.** *For any kernel $\kappa$ satisfying the hypotheses of Theorem 3, and for any $\eta_0 \in \mathscr{C}_{\text{EWP},m}$ for which $\overline{\text{MMD}}_{\kappa}(\eta_0, \eta^{\pi}) \leq D < \infty$, the iterates $\eta_{k+1} = \Pi_{\text{EWP},\kappa}^m\mathcal{T}^{\pi}\eta_k$ converge to a set $\boldsymbol{\eta}_{\text{EWP},\kappa}^m \subset \mathscr{C}_{\text{EWP},m}$, where*

$$\boldsymbol{\eta}_{\text{EWP},\kappa}^m \subset \overline{B}\left(\eta^{\pi}, \frac{6d^{\alpha/2}R_{\max}^{\alpha}}{(1-\gamma^{\alpha/2})(1-\gamma)^{\alpha}\sqrt{m}}\right).$$

# 5  Categorical Multivariate Distributional Dynamic Programming

Our next contribution is the introduction of a convergent multivariate distributional dynamic programming algorithm based on a *categorical* representation of return distribution functions, generalizing the algorithms and analysis of [RBD$^+$18] to the multivariate setting.

**Categorical representations.** As outlined above, to model the multi-return distribution function in practice, it is necessary to restrict each multi-return distribution to a finitely-parameterized class. In this work, we take inspiration from successful distributional RL algorithms [BDM17b, RBD$^+$18] and employ a *categorical representation*. The work of [WUS23] proposed a categorical representation for multivariate DRL, but their categorical projection was not justified theoretically, and it required a particular choice of fixed support. We propose a novel categorical representation with a finite (possibly state-dependent) support $\mathcal{R}(x) = \{\xi(x)_i\}_{i=1}^{N(x)} \subset \mathbf{R}^d$, that models the multi-return distribution function $\eta$ such that $\eta(x) \in \Delta_{\mathcal{R}(x)}$ for each $x \in \mathcal{X}$. The notation $\xi(x)_i$ simply refers to the $i$th support point at state $x$ specified by $\mathcal{R}$, and $\Delta_A$ denotes the probability simplex on the finite set $A$. We refer to the mapping $\mathcal{R}$ as the *support map*[2] and we denote the class of multi-return distribution functions under the corresponding categorical representation as $\mathscr{C}_{\mathcal{R}} \triangleq \prod_{x \in \mathcal{X}} \Delta_{\mathcal{R}(x)}$.

**Categorical projection.** Once again, the distributional Bellman operator is not generally closed over $\mathscr{C}_{\mathcal{R}}$, that is, $\mathcal{T}^{\pi} \mathscr{C}_{\mathcal{R}} \not\subset \mathscr{C}_{\mathcal{R}}$. As such, we cannot actually employ the procedure described in Theorem 2 – rather, we need to project applications of $\mathcal{T}^{\pi}$ back onto $\mathscr{C}_{\mathcal{R}}$. Roughly, we need an operator $\Pi : \mathcal{P}(\mathbf{R}^d)^{\mathcal{X}} \to \mathscr{C}_{\mathcal{R}}$ for which $\Pi|_{\mathscr{C}_{\mathcal{R}}} = \mathrm{id}$. Given such an operator, as in the literature on categorical distributional RL [BDM17b, RBD$^+$18], we will study the convergence of iterates $\eta_{k+1} = \Pi \mathcal{T}^{\pi} \eta_k$.

Projection operators used in the scalar categorical distributional RL literature are specific to distributions over $\mathbf{R}$, so we must introduce a new projection. We propose a projection similar to (4),

$$(\Pi_{C,\kappa}^{\mathcal{R}} \eta)(x) = \underset{p \in \Delta_{\mathcal{R}(x)}}{\operatorname{arginf}} \ \mathrm{MMD}_{\kappa}(p, \eta(x)). \tag{7}$$

We will now verify that $\Pi_{C,\kappa}^{\mathcal{R}}$ is well-defined, and that it satisfies the aforementioned properties.

**Lemma 1.** *Let $\kappa$ be a kernel induced by a semimetric $\rho$ on $[0, (1-\gamma)^{-1} R_{\max}]^d$ with strong negative type (cf. Theorem 1). Then $\Pi_{C,\kappa}^{\mathcal{R}}$ is well-defined, $\mathrm{Ran}(\Pi_{C,\kappa}^{\mathcal{R}}) \subset \mathscr{C}_{\mathcal{R}}$, and $\Pi_{C,\kappa}^{\mathcal{R}}|_{\mathscr{C}_{\mathcal{R}}} = \mathrm{id}$.*

It is worth noting that beyond simply ensuring the well-posedness of the projection $\Pi_{C,\kappa}^{\mathcal{R}}$, Lemma 1 also certifies an efficient algorithm for computing the projection — namely, by solving the appropriate quadratic program (QP), as observed by [SZS$^+$08]. We demonstrate pseudocode for computing the projected Bellman operator $\Pi_{C,\kappa}^{\mathcal{R}} \mathcal{T}^{\pi}$ with a QP solver QPSolve in Algorithm 1.

---

**Algorithm 1** Projected Categorical Dynamic Programming

---

**Require:** Support map $\mathcal{R}$, kernel $\kappa$, transition kernel $P^{\pi}$, reward function $r$, discount $\gamma$
**Require:** Return distribution function $\eta \in \mathscr{C}_{\mathcal{R}}$
  **for** $x \in \mathcal{X}$ **do**
    $(\mathcal{T}^{\pi} \eta)_x \leftarrow \sum_{x' \in \mathcal{X}} \sum_{\xi \in \mathcal{R}(x')} P^{\pi}(x' \mid x) \eta_{x'}(\xi) \delta_{r(x) + \gamma \xi}$
    $K_{i,j}^x \leftarrow \kappa(\xi_i, \xi_j)$ for each $(\xi_i, \xi_j) \in \mathcal{R}(x)^2$
    $q_j^x \leftarrow \sum_{\xi \in \mathrm{supp}\,(\mathcal{T}^{\pi} \eta)_x} (\mathcal{T}^{\pi} \eta)_x(\xi) \kappa(\xi_j, \xi)$ for each $\xi_j \in \mathcal{R}(x)$
    $p \leftarrow \mathsf{QPSolve}\left(\min_{p \in \mathbf{R}^{|\mathcal{R}(x)|}} \left[p^{\top} K^x p - 2 p^{\top} q\right] \text{ subject to } p \succeq 0, \ \sum_i p_i = 1\right)$
    $(\Pi_{C,\kappa}^{\mathcal{R}} \mathcal{T}^{\pi} \eta)_x \leftarrow \sum_{\xi_i \in \mathcal{R}(x)} p_i \delta_{\xi_i}$
  **end for**
  **return** $\Pi_{C,\kappa}^{\mathcal{R}} \mathcal{T}^{\pi} \eta$

---

**Lemma 2.** *Under the conditions of Lemma 1, $\Pi_{C,\kappa}^{\mathcal{R}}$ is a nonexpansion in $\overline{\mathrm{MMD}}_{\kappa}$. That is, for any $\eta_1, \eta_2 \in \mathcal{P}([0, (1-\gamma)^{-1} R_{\max}]^d)^{\mathcal{X}}$, we have $\overline{\mathrm{MMD}}_{\kappa}(\Pi_{C,\kappa}^{\mathcal{R}} \eta_1, \Pi_{C,\kappa}^{\mathcal{R}} \eta_2) \leq \overline{\mathrm{MMD}}_{\kappa}(\eta_1, \eta_2)$.*

**Categorical multivariate distributional dynamic programming.** As an immediate consequence of Lemma 2, it follows that projected dynamic programming under the projection $\Pi_{C,\kappa}^{\mathcal{R}}$ is convergent;

---

[2]In many applications, the most natural support map is constant across the state space.

this is because $\mathcal{T}^\pi$ is a contraction in $\overline{\mathrm{MMD}}_\kappa$ and $\Pi_{\mathrm{C},\kappa}^\mathcal{R}$ is a nonexpansion in $\overline{\mathrm{MMD}}_\kappa$, so the projected operator $\Pi_{\mathrm{C},\kappa}^\mathcal{R}\mathcal{T}^\pi$ is a contraction in $\overline{\mathrm{MMD}}_\kappa$; a standard invocation of the Banach fixed point theorem appealing to the completenes of $\overline{\mathrm{MMD}}_\kappa$ certifies that repeated application of $\Pi_{\mathrm{C},\kappa}^\mathcal{R}\mathcal{T}^\pi$ will result in convergence to a unique fixed point.

**Corollary 2.** *Let $\kappa$ be a kernel satisfying the conditions of Theorem 2. Then for any $\eta_0 \in \mathscr{C}_\mathcal{R}$, the iterates $\{\eta_k\}_{k=1}^\infty$ given by $\eta_{k+1} = \Pi_{\mathrm{C},\kappa}^\mathcal{R}\mathcal{T}^\pi\eta_k$ converge geometrically to a unique fixed point.*

Beyond the result of Theorem 2, Corollary 2 illustrates an algorithm for estimating $\eta^\pi$ in $\overline{\mathrm{MMD}}_\kappa$ provided knowledge of the transition kernel and the reward function, which is *computationally tractable* in tabular MDPs. Indeed, the iterates $(\eta_k)_{k\geq 0}$ all lie in $\mathscr{C}_\mathcal{R}$, having finitely-many degrees of freedom. Algorithm 1 outlines a computationally tractable procedure for learning $\eta_{\mathrm{C},\kappa}^\pi$ in this setting.

We note that the work of [WUS23] provided an alternative multivariate categorical algorithm, which was not analyzed theoretically. Moreover, our method provides the additional ability to support state-dependent arbitrary support maps, while theirs requires support maps to be uniform grids.

**Accurate approximations.** We now provide bounds showing that the fixed point $\eta_{\mathrm{C},\kappa}^\pi$ from Corollary 2 can be made arbitrarily accurate by increasing the number of atoms.

To derive a bound on the quality of the fixed point, we provide a reduction via partitioning the space of returns to the covering number of this space. Proceeding, we define a class of partitions $\mathscr{P}_{\mathcal{R}(x)}$, where each $P \in \mathscr{P}_{\mathcal{R}(x)}$ satisfies

1. $|P| = N(x)$;
2. For any $\theta_1, \theta_2 \in P$, either $\theta_1 \cap \theta_2 = \emptyset$ or $\theta_1 = \theta_2$;
3. $\cup_{\theta \in P} \theta = \mathcal{P}([0, (1-\gamma)^{-1}R_{\max}]^d)$;
4. Each element $\theta_i \in P$ contains exactly one element $z_i \in \mathcal{R}(x)$.

For any partition $P$, we define a notion of *mesh size* relative to a kernel $\kappa$ induced by a semimetric $\rho$,

$$\mathsf{mesh}(P; \kappa) = \max_{\theta \in P} \sup_{y_1, y_2 \in \theta} \rho(y_1, y_2). \tag{8}$$

The following result confirms that $\eta_{\mathrm{C},\kappa}^\pi$ recovers $\eta^\pi$ as the mesh decreases.

**Theorem 4.** *Let $\kappa$ be a kernel induced by a $c$-homogeneous and shift-invariant semimetric $\rho$ conforming to the conditions of Theorem 2. Then the fixed point $\eta_{\mathrm{C},\kappa}^\pi$ of $\Pi_{\mathrm{C},\kappa}^\mathcal{R}\mathcal{T}^\pi$ satisfies*

$$\overline{\mathrm{MMD}}_\kappa(\eta_{\mathrm{C},\kappa}^\pi, \eta^\pi) \leq \frac{1}{1-\gamma^{c/2}} \sup_{x \in \mathcal{X}} \inf_{P \in \mathscr{P}_{\mathcal{R}(x)}} \sqrt{\mathsf{mesh}(P; \kappa)}. \tag{9}$$

Thus, for any sequence of supports $\{\mathcal{R}(x)_m\}_{m\geq 1}$ for which the maximal space (in $\rho$) between any two points in $\mathcal{R}(x)_m$ tends to 0 as $m \to \infty$, the fixed point $\eta_{\mathrm{C},\kappa}^\pi$ approximates $\eta^\pi$ to arbitrary precision for large enough $m$. The next corollary illustrates this in a familiar setting.

**Corollary 3.** *Let $\mathcal{R}(x) = U_m$, where $U_m$ is a set of $m$ uniformly-spaced support points on $[0, (1-\gamma)^{-1}R_{\max}]$. For $\kappa$ induced by the semimetric $\rho(x, y) = \|x - y\|_2^\alpha$ for $\alpha \in (0, 2)$,*

$$\overline{\mathrm{MMD}}_\kappa(\eta_{\mathrm{C},\kappa}^\pi, \eta^\pi) \leq \frac{1}{(1-\gamma^{\alpha/2})(1-\gamma)^{\alpha/2}} \frac{d^{\alpha/4}R_{\max}^{\alpha/2}}{(m^{1/d} - 2)^{\alpha/2}}.$$

With $\alpha = 1$ and $d = 1$, the MMD in Corollary 3 is equivalent to the Cramér metric [SR13], the setting in which categorical (scalar) distributional dynamic programming is well understood. Our rate matches the known $\Theta(m^{-1/2})$ rate shown by [RBD+18] in this setting. Thus, our results offer a new perspective on categorical DRL, and naturally generalizes the theory to the multivariate setting.

Theorem 4 relies on the following lemma about the approximation quality of the categorical MMD projection, which may be of independent interest.

**Lemma 3.** *Let $\kappa$ be kernel satisfying the conditions of Lemma 1, and for any finite $\mathcal{R} \subset \mathbf{R}^d$, define $\Pi : \mathcal{P}(\mathbf{R}^d) \to \Delta_\mathcal{R}$ via $\Pi p = \mathrm{arginf}_{q \in \Delta_\mathcal{R}} \mathrm{MMD}_\kappa(p, q)$. Then $\mathrm{MMD}_\kappa^2(\Pi p, p) \leq \inf_{P \in \mathscr{P}_\mathcal{R}} \mathsf{mesh}(P; \kappa)$.*

At this stage, we have shown definitively that categorical dynamic programming converges in the multivariate case. In the sequel, we build on these results to provide a convergent multivariate categorical TD-learning algorithm.

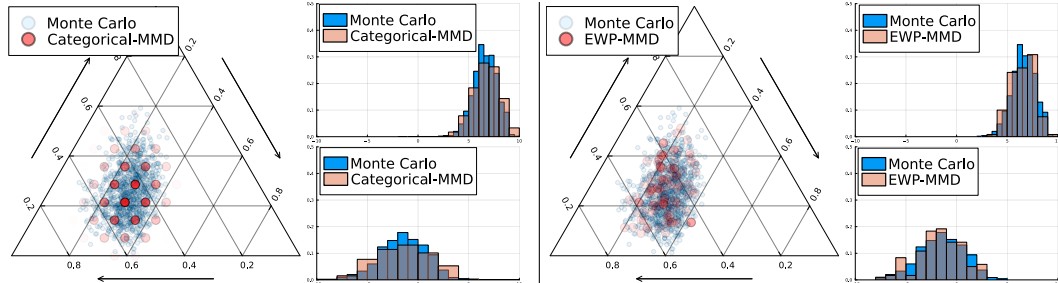

Figure 1: Distributional SMs and associated predicted return distributions with the categorical (left) and EWP (right) representations. Simplex plots denote the distributional SM. Histograms denote the associated return distributions, predicted from a pair of held-out reward functions.

## 5.1 Simulation: The Distributional Successor Measure

As a preliminary example, we consider 3-state MDPs with the cumulants $r(i) = (1 - \gamma)e_i, i \in [3]$ for $e_i$ the $i$th basis vector. In this setting, $\eta^\pi$ encodes the distribution over trajectory-wise discounted state occupancies, which was discussed in the recent work of [WFG$^+$24] and called the *distributional successor measure* (DSM). Particularly, [WFG$^+$24] showed that $x \mapsto \text{Law}\left(G_x^\top \tilde{r}\right)$ for $G_x \sim \eta^\pi(x)$ is the return distribution function for any scalar reward function $\tilde{r}$. Figure 1 shows that the projected categorical dynamic programming algorithm accurately approximates the distribution over discounted state occupancies as well as distributions over returns on held-out reward functions.

## 6 Multivariate Distributional TD-Learning

Next, we devise an algorithm for approximating the multi-return distribution function when the transition kernel and reward function are not known, and are observed only through samples. Indeed, this is a strong motivation for TD-learning algorithms [Sut88], wherein state transition data alone is used to solve the Bellman equation. In this section, we devise a TD-learning algorithm for multivariate DRL, leveraging our results on categorical dynamic programming in $\overline{\text{MMD}}_\kappa$.

**Relaxation to signed measures.** In the $d = 1$ setting, the categorical projection presented above is known to be affine [RBD$^+$18], making scalar categorical TD-learning amenable to common techniques in stochastic approximation theory. However, the projection is *not* affine when $d \geq 2$; we give an explicit example in Appendix D. Thus, we relax the categorical representation to include *signed* measures, which will provide us with an affine projection [BRCM19]—this is crucial for proving our main result, Theorem 6. We denote by $\mathcal{M}^1(\mathcal{Y})$ the set of all signed measures $\mu$ over $\mathcal{Y}$ with $\mu(\mathcal{Y}) = 1$. We begin by noting that the MMD endows $\mathcal{M}^1(\mathcal{Y})$ with a metric structure.

**Lemma 4.** *Let $\kappa : \mathcal{Y} \times \mathcal{Y} \to \mathbf{R}$ be a characteristic kernel over some space $\mathcal{Y}$. Then $\text{MMD}_\kappa : \mathcal{M}^1(\mathcal{Y}) \times \mathcal{M}^1(\mathcal{Y}) \to \mathbf{R}_+$ given by $(p, q) \mapsto \|\mu_p - \mu_q\|_\mathcal{H}$ defines a metric on $\mathcal{M}^1(\mathcal{Y})$, where $\mu_p$ denotes the usual mean embedding of $p$ and $\mathcal{H}$ is the RKHS with kernel $\kappa$.*

We define the relaxed projection $\Pi_{\text{SC},\kappa}^\mathcal{R} : \mathcal{M}^1([0, (1-\gamma)^{-1}R_{\max}]^d)^\mathcal{X} \to \prod_{x \in \mathcal{X}} \mathcal{M}^1(\mathcal{R}(x)) =: \mathscr{S}_\mathcal{R}$,

$$\left(\Pi_{\text{SC},\kappa}^\mathcal{R}\eta\right)(x) \in \underset{p \in \mathcal{M}^1(\mathcal{R}(x))}{\text{arginf}} \ \text{MMD}_\kappa(p, \eta(x)). \tag{10}$$

Note that (10) is very similar to the definition of the categorical MMD projection in (7)—the only difference is that the optimization occurs over the larger class of signed mass-1 measures. It is also worth noting that the distributional Bellman operator can be applied directly to signed measures, which yields the following convenient result.

**Lemma 5.** *Under the conditions of Corollary 2, the projected operator $\Pi_{\text{SC},\kappa}^\mathcal{R}\mathcal{T}^\pi : \mathscr{S}_\mathcal{R} \to \mathscr{S}_\mathcal{R}$ is affine, is contractive with contraction factor $\gamma^{c/2}$, and has a unique fixed point $\eta_{\text{SC},\kappa}^\pi$.*

While we have "relaxed" the projection, the fixed point $\eta_{\text{SC},\kappa}^\pi$ is a good approximation of $\eta^\pi$.

**Theorem 5.** *Under the conditions of Lemma 5, we have that*

1. $\overline{\mathrm{MMD}}_\kappa(\eta_{\mathrm{SC},\kappa}^\pi, \eta^\pi) \leq \frac{1}{1-\gamma^{c/2}} \sup_{x \in \mathcal{X}} \inf_{P \in \mathscr{P}_{\mathcal{R}(x)}} \sqrt{\mathrm{mesh}(P; \kappa)}$; and

2. $\overline{\mathrm{MMD}}_\kappa(\Pi_{\mathrm{C},\kappa}^{\mathcal{R}} \eta_{\mathrm{SC},\kappa}^\pi, \eta^\pi) \leq (1 + \frac{1}{1-\gamma^{c/2}}) \sup_{x \in \mathcal{X}} \inf_{P \in \mathscr{P}_{\mathcal{R}(x)}} \sqrt{\mathrm{mesh}(P; \kappa)}$.

Notably, the second statement of Theorem 5 states that projecting $\eta_{\mathrm{SC},\kappa}^\pi$ back onto the space of multi-return distribution functions yields approximately the same error as $\eta_{\mathrm{C},\kappa}^\pi$ when $\gamma$ is near 1.

In the remainder of the section, we assume access to a stream of MDP transitions $\{T_t\}_{t=1}^\infty \subset \mathcal{X} \times \mathcal{A} \times \mathbf{R}^d \times \mathcal{X}$ consisting of elements $T_t = (X_t, A_t, R_t, X_t')$ with the following structure,

$$X_t \sim \mathbf{P}(\cdot \mid \mathcal{F}_{t-1}) \qquad A_t \sim \pi(\cdot \mid X_t) \qquad R_t = r(X_t) \qquad X_t' \sim P(\cdot \mid X_t, A_t) \qquad (11)$$

where $\mathbf{P}$ is some probability measure and $\{\mathcal{F}_t\}_{t=1}^\infty$ is the canonical filtration $\mathcal{F}_t = \sigma(\cup_{t=1}^t T_t)$. Based on these transitions, we can define stochastic distributional Bellman backups by

$$\widehat{\mathcal{T}}_t^\pi \eta(x) = \begin{cases} (\mathrm{b}_{R_t, \gamma})_\sharp \eta(X_t') & x = X_t \\ \eta(x) & \text{otherwise} \end{cases}, \qquad (12)$$

which notably can be computed exactly without knowledge of $P, r$. Due to the stronger convergence guarantees shown for projected multivariate distributional dynamic programming, we introduce an asynchronous incremental algorithm leveraging the categorical representation, which produces iterates $\{\widehat{\eta}_t\}_{t=1}^\infty$ according to

$$\widehat{\eta}_{t+1} = (1 - \alpha_t)\widehat{\eta}_t + \alpha_t \Pi_{\mathrm{SC},\kappa}^{\mathcal{R}} \widehat{\mathcal{T}}_t^\pi \widehat{\eta}_t \qquad (13)$$

for $\widehat{\eta}_0 \in \mathscr{C}_{\mathcal{R}}$, where $\{\alpha_t\}_{t=1}^\infty$ is any sequence of (possibly) random step sizes adapted to the filtration $\{\mathcal{F}_t\}_{t=1}^\infty$. The iterates of (13) closely resemble those of classic stochastic approximation algorithms [RM51] and particularly asynchronous TD learning algorithms [JJS93, Tsi94, BT96], but with iterates taking values in the space of state-indexed signed measures. Indeed, our next result draws on the techniques from these works to establish convergence of TD-learning on $\mathscr{S}_{\mathcal{R}}$ representations.

**Theorem 6.** *For a kernel $\kappa$ induced by a semimetric $\rho$ of strong negative type, the sequence $\{\widehat{\eta}_t\}_{t=1}^\infty$ given by (11)-(13) converges to $\eta_{\mathrm{SC},\kappa}^\pi$ with probability 1.*

### 6.1 Simulations: Distributional Successor Features

To illustrate the behavior of our categorical TD algorithm, we employ it to learn the multi-return distributions for several tabular MDPs with random cumulants. We focus on the case of 2- and 3-dimensional cumulants, which is the setting studied in recent works regarding multivariate distributional RL [ZCZ+21, WUS23]. Interpreting the multi-return distributions as joint distributions over successor features [BDM+17a, SFs], we additionally evaluate the return distributions for random reward functions in the span of the cumulants. We compare our categorical TD approach with a tabular implementation of the EWP TD algorithm of [ZCZ+21], for which no convergence bounds are known.

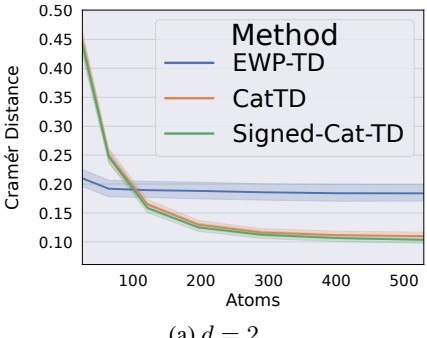
(a) $d = 2$

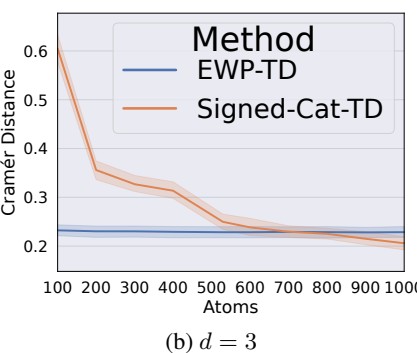
(b) $d = 3$

Figure 2: Error of zero-shot return distribution predictions over random MDPs, measured by Cramér distance, and showing 95% confidence intervals.

Figure 2a compares TD learning approaches based on their ability to accurately infer (scalar) return distributions on held out reward functions, averaged over 100 random MDPs, with transitions drawn

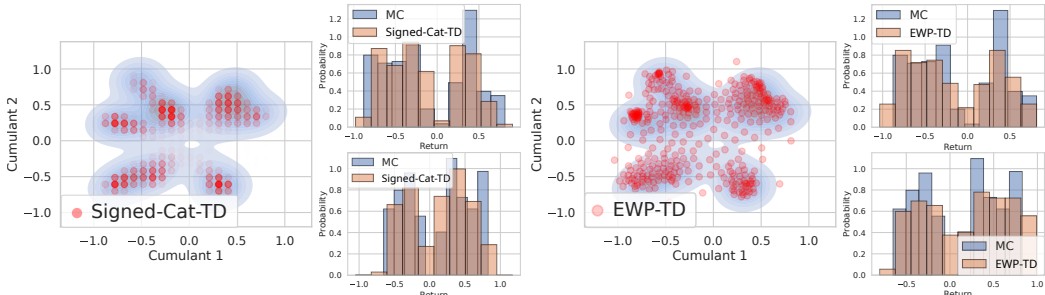

Figure 3: Distributional SFs and predicted return distributions with $m = 400$ atoms, in a random MDP with known rectangular bound on cumulants. **Left**: Categorical TD. **Right**: EWP TD.

from Dirichlet priors and 2-dimensional cumulants drawn from uniform priors. The performance of the categorical algorithms sharply increases as the number of atoms increases. On the other hand, the EWP TD algorithm performs well with few atoms, but does not improve very much with higher-resolution representations. We posit this is due to the algorithm getting stuck in local minima, given the non-convexity of the EWP MMD objective. This hypothesis is supported as well by Figure 3, which depicts the learned distributional SFs and return distribution predictions.

Particularly, we observe that the learned particle locations in the EWP TD approach tend to cluster in some areas or get stuck in low-density regimes, which suggests the presence of a local optimum. On the other hand, our provably-convergent categorical TD method learns a high fidelity quantization of the true multi-return distributions.

Naturally, however, the benefits of the $\text{poly}(d)$ bounds for EWP suggested by Theorem 3 become more present as we increase the cumulant dimension. Figure 2b repeats the experiment of Figure 2a with $d = 3$, using randomized support points for the categorical algorithm to avoid a cubic growth in the cardinality of the supports. Notably, our method is the first capable of supporting such unstructured supports. While the categorical TD approach can still outperform EWP, a much larger number of atoms is required. This is unsurprising in light of our theoretical results.

# 7 Conclusion

We have provided the first provably convergent and computationally tractable algorithms for learning multivariate return distributions in tabular MDPs. Our theoretical results include convergence guarantees that indicate how accuracy scales with the number of particles $m$ used in distribution representations, and interestingly motivate the use of signed measures to develop provably convergent TD algorithms.

While it is difficult to scale categorical representations to high-dimensional cumulants, our algorithm is highly performant in the low $d$ setting, which has been the focus of recent work in multivariate distributional RL. Notably, even the $d = 2$ setting has important applications—indeed, efforts in safe RL depend on distinguishing a cost signal from a reward signal (see, e.g., [YSTS23]), which can be modeled by bivariate distributional RL. In this setting, our method can easily be scaled to large state spaces by approximating the categorical signed measures with neural networks; an illustrated example is given in Appendix F.

On the other hand, the prospect of learning multi-return distributions for high-dimensional cumulants also has many important applications, such as modeling close approximations to distributional successor measures [WFG$^+$24] for zero-shot risk-sensitive policy evaluation. In this setting, we believe EWP-based multivariate DRL will be highly impactful. Our results concerning EWP dynamic programming provide promising evidence that the accuracy of EWP representations scales gracefully with $d$ for a fixed number of atoms. Thus, we believe that understanding convergence of EWP TD-learning algorithms is a very interesting and important open problem for future investigation.

## Acknowledgements

The authors would like to thank Yunhao Tang, Tyler Kastner, Arnav Jain, Yash Jhaveri, Will Dabney, David Meger, and Marc Bellemare for their helpful feedback, as well as insightful suggestions from anonymous reviewers. This work was supported by the Fonds de Recherche du Québec, the National Sciences and Engineering Research Council of Canada, and the compute resources provided by Mila (`mila.quebec`).

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

# Appendices

## A   In-Depth Summary of Related Work

In Sections 1 and 2, we provided a high-level synopsis of the state of existing work in multivariate distributional RL. In this section, we elaborate further.

**Analysis techniques.** Our results in this paper mostly drawn on the analysis of one-dimensional distributional RL algorithms such as categorical and quantile dynamic programming, as well as their temporal-difference learning counterparts [RBD+18, DRBM18, RMA+24, BDR23]. The proof techniques in these works themselves are related to contraction-based arguments for reinforcement learning with function approximation [Tsi94, BT96, TVR97].

**Multivariate distributional RL algorithms.** Several prior works have contributed algorithms for multivariate distributional reinforcement learning, along with empirical demonstrations and theoretical analysis, though as we note in the main paper, the approaches proposed in this paper are the first algorithms with strong theoretical guarantees and efficient tabular implementations. [FSMT19] propose a deep-learning-based approach in which generative adversarial networks are used to model multivariate return distributions, and use these predictions to inform exploration strategies. [ZCZ+21] propose the TD algorithm combing equally-weighted particle representations and an MMD loss that we recall in Equation (3). They demonstrate the effectiveness of this algorithm in combination with deep learning function approximators, though do not analyze it. [WUS23] propose a family of algorithms for multivariate distributional RL termed fitted likelihood evaluation. These methods mirror LSTD algorithms [BB96], iteratively minimising a batch objective function (in this case, a negative log-likelihood, NLL) over a growing dataset. [WUS23] demonstrate that these algorithms are performant in low-dimensional settings empirically, and provide theoretical analysis for FLE algorithms, assuming an oracle which can approximately optimise the NLL objective at each algorithm step. [SFS24] also propose a TD learning algorithm for one-dimensional distributional RL using categorical support and an MMD-based loss. They demonstrate strong performance of this algorithm in classic RL domains such as CartPole and Mountain Car, but do not analyze the algorithm. Our analysis in this paper suggests our novel relaxation to mass-1 signed measures may be crucial to obtaining a straightforwardly analyzable TD algorithm.

Finally, the concurrent work of [LK24] studied distributional Bellman operators for Banach-space-valued reward functions. Their work focuses on analyzing how well the fixed point of a distributional finite-dimensional multivariate Bellman equation can approximate the fixed point of a distributional Banach-space-valued Bellman equation. In contrast, our work only studies finite-dimensional reward functions, but provides explicit convergence rates and approximation bounds when the *distribution representations* are finite dimensional, unlike [LK24]. Moreover, [LK24] considers a similar algorithm to that discussed in Theorem 3 but for categorical representations, though its convergence is not proved. Furthermore, [LK24] did not prove convergence of any TD-learning algorithms, although they did propose some TD-learning algorithms which achieved interesting results in simulation.

# B  Proofs

## B.1  Multivariate Distributional Dynamic Programming: Section 3

In this section, we will state some lemmas building up to the proof of Theorem 2. These lemmas generalize corresponding results of [NTGV20] that were specific to the scalar reward setting. We begin with a lemma that demonstrates a notion of convexity for the MMD induced by a conditional positive definite kernel.

**Lemma 6.** *Let $(p_a)_{a \in \mathcal{I}} \subset \mathcal{P}(\mathcal{Y})$ and $(q_a)_{a \in \mathcal{I}} \subset \mathcal{P}(\mathcal{Y})$ be collections of probability measures indexed by an index set $\mathcal{I}$. Suppose $T \in \mathcal{P}(\mathcal{I})$. Then for any characteristic kernel $\kappa$, the following holds,*

$$\mathrm{MMD}_\kappa(\mathbf{E}_{a \sim T}[p_a], \mathbf{E}_{a' \sim T}[q_{a'}]) \leq \sup_{a \in \mathcal{I}} \mathrm{MMD}_\kappa(p_a, q_a)$$

*Proof.* It is known from [Sch00] that characteristic kernels generate RKHSs $\mathcal{H}$ into which probability measures can be embedded. As such, it holds that

$$\mathrm{MMD}_\kappa(p, q) = \|\mu_p - \mu_q\|$$

where $\|\cdot\|$ is the norm in the Hilbert space $\mathcal{H}$ and $\mu_p$ is the *mean embedding* of $p$ – that is, the unique element of $\mathcal{H}$ such that $\mathbf{E}_{y \sim p}[f(y)] = \langle f, \mu_p \rangle$ for every $f \in \mathcal{H}$, and where $\langle \cdot, \cdot \rangle$ is the inner product in $\mathcal{H}$.

Let $T_p \triangleq \mathbf{E}_{a \sim T}[p_a]$ and define $T_q$ analogously. We claim that $\mu_{Tp} = \mathbf{E}_{a \sim T}[\mu_{p_a}] \triangleq T\mu_p$. To see this, let $f \in \mathcal{H}$, and observe that

$$\begin{aligned}
\mathbf{E}_{y \sim Tp}[f](y) &= \int f(y) Tp(\mathrm{d}y) \\
&= \iint f(y) T(\mathrm{d}a) p_a(\mathrm{d}y) \\
&= \iint f(y) p_a(\mathrm{d}y) T(\mathrm{d}a) \\
&= \int \langle f, \mu_{p_a} \rangle T(\mathrm{d}a) \\
&= \left\langle f, \int \mu_{p_a} T(\mathrm{d}a) \right\rangle \\
&= \langle f, T\mu_p \rangle,
\end{aligned}$$

where the third step invokes Fubini's theorem, and the penultimate steps leverages the linearity of the inner product. Notably, $T$ acts as a linear operator on mean embeddings. As a result, we see that

$$
\begin{aligned}
\mathrm{MMD}_\kappa(Tp, Tq) &= \|\mu_{Tp} - \mu_{Tq}\| \\
&= \|T\mu_p - T\mu_q\| \\
&= \left\|\int_{\mathcal{I}} (\mu_{p_a} - \mu_{q_a}) T(\mathrm{d}a)\right\| \\
&\leq \int \|\mu_{p_a} - \mu_{q_a}\| T(\mathrm{d}a) \\
&\leq \sup_{a \in \mathcal{I}} \|\mu_{p_a} - \mu_{q_a}\| \\
&= \sup_{a \in \mathcal{I}} \mathrm{MMD}_\kappa(\mu_{p_a}, \mu_{q_a}).
\end{aligned}
$$

where the penultimate inequality is due to Jensen's inequality, and the final inequality holds since $\sup_{a \in \mathcal{I}} \|\mu_{p_a} - \mu_{q_a}\|$ upper bounds the integrand, and the integral is a monotone operator. □

Next, we show how the $\mathrm{MMD}_\kappa$ under the kernels hypothesized in Theorem 2 behave under affine transformations to random variables.

**Lemma 7.** *Let $\kappa$ be a kernel induced by a semimetric $\rho$ of strong negative type defined over a compact subset $\mathcal{Y} \subset \mathbf{R}^d$ that is both shift invariant and $c$-homogeneous (cf. Theorem 2). Then for any $a \in \mathcal{Y}$ and $p, q \in \mathcal{P}(\mathcal{Y})$,*

$$
\mathrm{MMD}_\kappa((\mathrm{b}_{a,\gamma})_\sharp p, (\mathrm{b}_{a,\gamma})_\sharp q) \leq \gamma^{c/2} \mathrm{MMD}_\kappa(p, q).
$$

*Proof.* It is known [GBR+12] that the MMD can be expressed in terms of expected kernel evaluations, according to

$$
\mathrm{MMD}_\kappa^2(p, q) = \mathbf{E}\left[\kappa(Y, Y')\right] + \mathbf{E}\left[\kappa(Z, Z')\right] - 2\mathbf{E}\left[\kappa(Y, Z)\right] \qquad (Y, Y', Z, Z') \sim p \otimes p \otimes q \otimes q
$$

$$
\begin{aligned}
&= \mathbf{E}\left[\frac{1}{2}(\rho(Y, y_0) + \rho(Y', y_0) - \rho(Y, Y'))\right] \\
&\quad + \mathbf{E}\left[\frac{1}{2}(\rho(Z, y_0) + \rho(Z', y_0) - \rho(Z, Z'))\right] \\
&\quad - \mathbf{E}\left[\rho(Y, y_0) + \rho(Z, y_0) - \rho(Y, Z)\right] \\
&= \mathbf{E}\left[\rho(Y, Z)\right] - \frac{1}{2}\left(\mathbf{E}\left[\rho(Y, Y')\right] + \mathbf{E}\left[\rho(Z, Z')\right]\right),
\end{aligned}
$$

where the last step invokes the definition of a kernel induced by a semimetric, and the linearity of expectation. Then, defining $\tilde{Y}, \tilde{Y}'$ as independent samples from $(\mathrm{b}_{a,\gamma})_\sharp p$ and $\tilde{Z}, \tilde{Z}'$ as independent samples from $(\mathrm{b}_{a,\gamma})_\sharp q$, we have

$$
\begin{aligned}
\mathrm{MMD}_\kappa^2((\mathrm{b}_{a,\gamma})_\sharp p, (\mathrm{b}_{a,\gamma})_\sharp q) &= \mathbf{E}\left[\rho(\tilde{Y}, \tilde{Z})\right] - \frac{1}{2}\left(\mathbf{E}\left[\rho(\tilde{Y}, \tilde{Y}')\right] + \mathbf{E}\left[\rho(\tilde{Z}, \tilde{Z}')\right]\right) \\
&= \mathbf{E}\left[\rho(a + \gamma Y, a + \gamma Z)\right] - \frac{1}{2}\left(\mathbf{E}\left[\rho(a + \gamma Y, a + \gamma Y')\right] + \mathbf{E}\left[\rho(a + \gamma Z, a + \gamma Z')\right]\right) \\
&= \mathbf{E}\left[\rho(\gamma Y, \gamma Z)\right] - \frac{1}{2}\left(\mathbf{E}\left[\rho(\gamma Y, \gamma Y')\right] + \mathbf{E}\left[\rho(\gamma Z, \gamma Z')\right]\right) \\
&= \gamma^c \mathbf{E}\left[\rho(Y, Z)\right] - \frac{\gamma^c}{2}\left(\mathbf{E}\left[\rho(Y, Y')\right] + \mathbf{E}\left[\rho(Z, Z')\right]\right) \\
&= \gamma^c \mathrm{MMD}_\kappa^2(p, q),
\end{aligned}
$$

where the second step is a change of variables, the third step invokes the shift invariance of $\rho$, and the fourth step invokes the homogeneity of $\rho$.

Thus, it follows that $\mathrm{MMD}_\kappa((\mathrm{b}_{a,\gamma})_\sharp p, (\mathrm{b}_{a,\gamma})_\sharp q) \leq \gamma^{c/2} \mathrm{MMD}_\kappa(p, q)$. □

We are now ready to prove the main result of this section.

**Theorem 2** (Convergent MMD dynamic programming for the multi-return distribution function). *Let $\kappa$ be a kernel induced by a semimetric $\rho$ on $[0, (1-\gamma)^{-1}R_{\max}]^d$ with strong negative type, satisfying*

1. **Shift-invariance**. *For any $z \in \mathbf{R}^d$, $\rho(z + y_1, z + y_2) = \rho(y_1, y_2)$.*

2. **Homogeneity**. *For any $\gamma \in [0,1)$, there exists $c > 0$ for which $\rho(\gamma y_1, \gamma y_2) = \gamma^c \rho(y_1, y_2)$.*

*Consider the sequence $\{\eta_k\}_{k=1}^\infty$ given by $\eta_{k+1} = \mathcal{T}^\pi \eta_k$. Then $\eta_k \to \eta^\pi$ at a geometric rate of $\gamma^{c/2}$ in $\overline{\mathrm{MMD}_\kappa}$, as long as $\overline{\mathrm{MMD}_\kappa}(\eta_0, \eta^\pi) \le C < \infty$.*

*Proof.* We begin by showing that the distributional Bellman operator $\mathcal{T}^\pi$ is contractive in $\overline{\mathrm{MMD}_\kappa}$. We have

$$\overline{\mathrm{MMD}_\kappa}(\mathcal{T}^\pi \eta_1, \mathcal{T}^\pi \eta_2) = \sup_{x \in \mathcal{X}} \mathrm{MMD}_\kappa(\mathcal{T}^\pi \eta_1(x), \mathcal{T}^\pi \eta_2(x))$$

$$= \sup_{x \in \mathcal{X}} \mathrm{MMD}_\kappa \left( \mathop{\mathbf{E}}_{x' \sim P^\pi(\cdot | x)} \left[ (\mathrm{b}_{r(x),\gamma})_\sharp \eta_1(x') \right], \mathop{\mathbf{E}}_{x'' \sim P^\pi(\cdot | x)} \left[ (\mathrm{b}_{r(x),\gamma})_\sharp \eta_2(x'') \right] \right).$$

We apply Lemma 6 with $\mathcal{I} = \mathcal{X}$ and $T = P^\pi(\cdot \mid x)$, yielding

$$\overline{\mathrm{MMD}_\kappa}(\mathcal{T}^\pi \eta_1, \mathcal{T}^\pi \eta_2) \le \sup_{x \in \mathcal{X}} \sup_{x' \in \mathcal{X}} \mathrm{MMD}_\kappa \left( (\mathrm{b}_{r(x),\gamma})_\sharp \eta_1(x'), (\mathrm{b}_{r(x),\gamma})_\sharp \eta_2(x') \right).$$

Next, invoking Lemma 7 with the shift-invariance and $c$-homogeneity of $\kappa$, we have

$$\overline{\mathrm{MMD}_\kappa}(\mathcal{T}^\pi \eta_1, \mathcal{T}^\pi \eta_2) \le \gamma^{c/2} \sup_{x \in \mathcal{X}} \sup_{x' \in \mathcal{X}} \mathrm{MMD}_\kappa \left( \eta_1(x'), \eta_2(x') \right)$$

$$= \gamma^{c/2} \sup_{x \in \mathcal{X}} \mathrm{MMD}_\kappa \left( \eta_1(x), \eta_2(x) \right)$$

$$= \gamma^{c/2} \overline{\mathrm{MMD}_\kappa}(\eta_1, \eta_2).$$

It follows that $\overline{\mathrm{MMD}_\kappa}(\eta_{k+1}, \eta^\pi) \le \gamma^{c/2} \overline{\mathrm{MMD}_\kappa}(\mathcal{T}^\pi \eta_k, \mathcal{T}^\pi \eta^\pi) = \gamma^{c/2} \overline{\mathrm{MMD}_\kappa}(\mathcal{T}^\pi \eta_k, \eta^\pi)$, since $\eta^\pi$ is a fixed point of $\mathcal{T}^\pi$. Continuing, we see that $\overline{\mathrm{MMD}_\kappa}(\eta_k, \eta^\pi) \le \gamma^{kc/2} \overline{\mathrm{MMD}_\kappa}(\eta_0, \eta^\pi) \le \gamma^{kc/2} C \in O(\gamma^{kc/2}) \subset o(1)$. Since $\overline{\mathrm{MMD}_\kappa}$ is a metric on $\mathcal{P}([0, (1-\gamma)^{-1}R_{\max}]^d)^{\mathcal{X}}$ for any characteristic kernel $\kappa$, it follows that $\eta_k$ approaches $\eta^\pi$ at a geometric rate. $\square$

## B.2 EWP Dynamic Programming: Section 4

In this section, we provide the proof of Theorem 3. To do so, we prove an abstract, general result, regarding any projection mapping that approximates the argmin MMD projection given in Equation (4).

**Proposition 1** (Convergence of EWP Dynamic Programming). *Consider a kernel satisfying the hypotheses of Theorem 2, suppose rewards are globally bounded in each dimension in $[0, R_{\max}]$, and let $\{\Pi_{\kappa,m}^{(k)}\}_{k \ge 0}$ be a sequence of maps $\Pi : \mathcal{P}([0, (1-\gamma)^{-1}R_{\max}]^d)^{\mathcal{X}} \to \mathscr{C}_{\mathrm{EWP},m}$ satisfying*

$$\mathrm{MMD}_\kappa((\Pi_{\kappa,m}^{(k)} \eta)(x), \eta(x)) \le f(d, m) < \infty \qquad \forall k \ge 0. \tag{6}$$

*Then the iterates $(\eta_k)_{k \ge 0}$ given by $\eta_{k+1} = \Pi_{\kappa,m}^{(k)} \mathcal{T}^\pi \eta_k$ with $\overline{\mathrm{MMD}_\kappa}(\eta_0, \eta^\pi) = D < \infty$ converge to a set $\boldsymbol{\eta}_{\mathrm{EWP},\kappa}^m \subset \overline{B}(\eta^\pi, (1 - \gamma^{c/2})^{-1} f(d, m))$ in $\overline{\mathrm{MMD}_\kappa}$, where $\overline{B}$ denotes the closed ball in $\overline{\mathrm{MMD}_\kappa}$,*

$$\overline{B}(\eta, R) \triangleq \left\{ \eta' \in \mathcal{P}(\mathbf{R}^d)^{\mathcal{X}} : \overline{\mathrm{MMD}_\kappa}(\eta, \eta') \le R \right\}.$$

*Proof.* Let $\Delta_k = \overline{\mathrm{MMD}_\kappa}(\eta_k, \eta^\pi)$. Then we have

$$\Delta_k = \overline{\mathrm{MMD}_\kappa}(\Pi_{\kappa,m}^{(k)} \mathcal{T}^\pi \eta_{k-1}, \mathcal{T}^\pi \eta^\pi)$$

$$\le \overline{\mathrm{MMD}_\kappa}(\Pi_{\kappa,m}^{(k)} \mathcal{T}^\pi \eta_{k-1}, \mathcal{T}^\pi \eta_{k-1}) + \overline{\mathrm{MMD}_\kappa}(\mathcal{T}^\pi \eta_{k-1}, \mathcal{T}^\pi \eta^\pi)$$

$$\le f(d, m) + \gamma^{c/2} \overline{\mathrm{MMD}_\kappa}(\eta_{k-1}, \eta^\pi)$$

$$\therefore \Delta_k \le f(d, m) + \gamma^{c/2} \Delta_{k-1},$$

where the first step invokes the identity that $\eta^\pi$ is the fixed point of $\mathcal{T}^\pi$ (which is well-defined by Theorem 2), the second step leverages the triangle inequality, and the third step follows by the definition of $f(d, m)$ and the contractivity of $\mathcal{T}^\pi$ established in Theorem 2. Unrolling the recurrence above, we have

$$\overline{\mathrm{MMD}_\kappa}(\eta_k, \eta^\pi) = \Delta_k \le \gamma^{ck/2} \Delta_0 + f(d, m) \sum_{i=0}^{\infty} \gamma^{ci/2}$$

$$\le \gamma^{ck/2} D + \frac{f(d, m)}{1 - \gamma^{c/2}}.$$

As such, as $k \to \infty$, we have that

$$\lim_{k \to \infty} \overline{\mathrm{MMD}_\kappa}\left(\eta_k, \overline{B}\left(\eta^\pi, \frac{f(d, m)}{1 - \gamma^{c/2}}\right)\right) = 0,$$

proving our claim. $\qquad\square$

Proposition 1, despite its simplicity, reveals an interesting fact: given a good enough approximate MMD projection $\Pi_{\kappa, m}$ in the sense that $f(d, m)$ decays quickly with $m$, the dynamic programming iterates $(\eta_k)_{k \ge 0}$ will eventually be contained in a (arbitrarily) small neighborhood of $\eta^\pi$. The next result provides an example consequence of this abstract result, and establishes that $m \in \mathsf{poly}(d)$ is enough for convergence to an arbitrarily small set with projected distributional dynamic programming under the EWP representation.

Finally, we can now prove Theorem 3, which we restate below for convenience.

**Theorem 3.** *Consider a kernel $\kappa$ induced by the semimetric $\rho(x, y) = \|x - y\|_2^\alpha$ with $\alpha \in (0, 2)$, and suppose rewards are bounded in each dimension within $[0, R_{\max}]$. For any $\eta_0$ such that $\overline{\mathrm{MMD}_\kappa}(\eta_0, \eta^\pi) \le D < \infty$, and any $\delta > 0$, for the sequence $(\eta_k)_{k \ge 0}$ defined in Equation (5), with probability at least $1 - \delta$ we have*

$$\overline{\mathrm{MMD}_\kappa}(\eta_K, \eta^\pi) \in \widetilde{O}\left(\frac{d^{\alpha/2} R_{\max}^\alpha}{(1 - \gamma^{\alpha/2})(1 - \gamma)^\alpha \sqrt{m}} \log\left(\frac{|\mathcal{X}| \delta^{-1}}{\log \gamma^{-\alpha}}\right)\right).$$

*where $\eta_{k+1} = \mathsf{BootProj}_{\kappa, m}^\pi \eta_k$ and $K = \lceil \frac{\log m}{\log \gamma^{-\alpha}} \rceil$, and where $\widetilde{O}$ omits logarithmic factors in $m$.*

*Proof.* For each $x \in \mathcal{X}$ and $k \in [K]$, denote by $\mathcal{E}_{x,k}$ the event given by

$$\mathcal{E}_{x,k} = \left\{\mathrm{MMD}_\kappa(\mathsf{BootProj}_{\kappa, m}^\pi \eta_k(x), \mathcal{T}^\pi \eta_k(x)) \le \frac{2 d^{\alpha/2} R_{\max}^\alpha}{(1 - \gamma)^\alpha \sqrt{m}} + \frac{4 d^{\alpha/2} R_{\max}^\alpha \log \delta'^{-1}}{(1 - \gamma)^\alpha \sqrt{m}} =: f(d, m; \delta')\right\},$$

for $\delta' > 0$ a constant to be chosen shortly. Moreover, with $\mathcal{E} = \cap_{(x,k) \in \mathcal{X} \times [K]} \mathcal{E}_{x,k}$, it holds that under $\mathcal{E}$, $\overline{\mathrm{MMD}_\kappa}(\mathsf{BootProj}_{\kappa, m}^\pi \eta_k, \mathcal{T}^\pi \eta_k) \le f(d, m; \delta')$ for all $k \in [K]$. Following the proof of Proposition 1, we have that, conditioned on $\mathcal{E}$,

$$\overline{\mathrm{MMD}_\kappa}(\eta_k, \eta^\pi) \le \gamma^{\alpha k/2} D + \frac{f(d, m; \delta')}{1 - \gamma^{\alpha/2}}$$

$$\le \frac{2 d^{\alpha/2} R_{\max}^\alpha}{(1 - \gamma)^\alpha \sqrt{m}} + \frac{f(d, m; \delta')}{1 - \gamma^{\alpha/2}}.$$

Now, by [TSM17, Proposition A.1], event $\mathcal{E}_{x,k}$ holds with probability at least $1 - \delta'$, since each $(\mathsf{BootProj}_{\kappa, m}^\pi \eta_k)(x)$ is generated independently by sampling $m$ independent draws from the distribution $\mathcal{T}^\pi \eta_k$. Therefore, event $\mathcal{E}$ holds with probability at least $1 - |\mathcal{X}| K \delta'$. Choosing $\delta' = \delta/(|\mathcal{X}| K)$, we have that, with probability at least $1 - \delta$,

$$\overline{\mathrm{MMD}_\kappa}(\eta_K, \eta^\pi) \le \frac{2 d^{\alpha/2} R_{\max}^\alpha}{(1 - \gamma)^\alpha \sqrt{m}} + \frac{1}{1 - \gamma^{\alpha/2}} \frac{2 d^{\alpha/2} R_{\max}^\alpha}{(1 - \gamma)^\alpha \sqrt{m}} \left(1 + 2 \log(|\mathcal{X}| K \delta^{-1})\right)$$

$$\le \frac{2 d^{\alpha/2} R_{\max}^\alpha}{(1 - \gamma)^\alpha \sqrt{m}} + \frac{2 d^{\alpha/2} R_{\max}^\alpha}{(1 - \gamma^{\alpha/2})(1 - \gamma)^\alpha \sqrt{m}} \left(1 + 2 \log\left(|\mathcal{X}|\left(1 + \frac{\log m}{\log \gamma^{-\alpha}}\right) \delta^{-1}\right)\right).$$

As such, there exist universal constants $C_0, C_1 \in \mathbf{R}_+$ such that

$$\overline{\mathrm{MMD}}_\kappa(\eta_K, \eta^\pi) \leq C_1 \frac{d^{\alpha/2} R_{\max}^\alpha}{(1 - \gamma^{\alpha/2})(1 - \gamma)^\alpha \sqrt{m}} \left( 1 + 2\log\left( |\mathcal{X}| \left( 1 + \frac{\log m}{\log \gamma^{-\alpha}} \right) \delta^{-1} \right) \right)$$

$$\leq C_0 \frac{d^{\alpha/2} R_{\max}^\alpha}{(1 - \gamma^{\alpha/2})(1 - \gamma)^\alpha \sqrt{m}} \left( \log |\mathcal{X}| + \log \frac{\log m}{\log \gamma^{-\alpha}} + \log \delta^{-1} \right) \tag{14}$$

$$= C_0 \frac{d^{\alpha/2} R_{\max}^\alpha}{(1 - \gamma^{\alpha/2})(1 - \gamma)^\alpha \sqrt{m}} \left( \log\left( \frac{|\mathcal{X}|\delta^{-1}}{\log \gamma^{-\alpha}} \right) + \log m \right).$$

$\square$

**Corollary 1.** *For any kernel $\kappa$ satisfying the hypotheses of Theorem 3, and for any $\eta_0 \in \mathscr{C}_{\mathrm{EWP},m}$ for which $\overline{\mathrm{MMD}}_\kappa(\eta_0, \eta^\pi) \leq D < \infty$, the iterates $\eta_{k+1} = \Pi_{\mathrm{EWP},\kappa}^m \mathcal{T}^\pi \eta_k$ converge to a set $\boldsymbol{\eta}_{\mathrm{EWP},\kappa}^m \subset \mathscr{C}_{\mathrm{EWP},m}$, where*

$$\boldsymbol{\eta}_{\mathrm{EWP},\kappa}^m \subset \overline{B}\left( \eta^\pi, \frac{6 d^{\alpha/2} R_{\max}^\alpha}{(1 - \gamma^{\alpha/2})(1 - \gamma)^\alpha \sqrt{m}} \right).$$

*Proof.* Proposition 1 shows that projected EWP dynamic programming converges to a set with radius controlled by the quantity $f(d, m)$ that upper bounds the distance $f(d, m)$ between $\eta_k(x)$ and $\Pi_{\kappa,m}^{(k)} \eta_k(x)$ at the worst state $x$. In the proof of Theorem 3, we saw that with nonzero probability, the randomized projections satisfy $f(d, m) \leq \frac{6 d^{\alpha/2} R_{\max}^\alpha}{(1-\gamma)^\alpha \sqrt{m}}$. Thus, there exists a projection satisfying this bound. Since the projection $\Pi_{\mathrm{EWP},\kappa}^m$ is, by definition, the projection with the smallest possible $f(d, m)$, the claim follows directly by Proposition 1. $\square$

### B.3 Categorical Dynamic Programming: Section 5

**Lemma 1.** *Let $\kappa$ be a kernel induced by a semimetric $\rho$ on $[0, (1 - \gamma)^{-1} R_{\max}]^d$ with strong negative type (cf. Theorem 1). Then $\Pi_{\mathrm{C},\kappa}^{\mathcal{R}}$ is well-defined, $\mathrm{Ran}(\Pi_{\mathrm{C},\kappa}^{\mathcal{R}}) \subset \mathscr{C}_{\mathcal{R}}$, and $\Pi_{\mathrm{C},\kappa}^{\mathcal{R}}|_{\mathscr{C}_{\mathcal{R}}} = \mathrm{id}$.*

*Proof.* Firstly, note that $\Delta_{\mathcal{R}(x)}$ is a bounded, finite-dimensional subspace for each $x \in \mathcal{X}$. Thus, $\Delta_{\mathcal{R}(x)}$ is compact, and by the continuity of the MMD, the infimum in (7) is attained.

Following the technique of [SZS$^+$08], we establish that $\Pi_{\mathrm{C},\kappa}^{\mathcal{R}}$ can be computed as the solution to a particular quadratic program with convex constraints.

Let $K \in \mathbf{R}^{N(x) \times N(x)}$ denote a matrix where $K_{i,j} = \kappa(\xi(x)_i, \xi(x)_j)$. Since $\kappa$ is a positive definite kernel when $\kappa$ is characteristic [GBR$^+$12], it follows that $K$ is a positive definite matrix. Then, for any $\varrho \in \mathcal{P}([0, (1 - \gamma)^{-1} R_{\max}]^d)$, we have

$$\operatorname*{arginf}_{p \in \Delta_{\mathcal{R}(x)}} \mathrm{MMD}_\kappa(p, \varrho)$$

$$= \operatorname*{arginf}_{p \in \Delta_{\mathcal{R}(x)}} \mathrm{MMD}_\kappa^2(p, \varrho)$$

$$= \operatorname*{arginf}_{p \in \Delta_{\mathcal{R}(x)}} \left\{ \sum_{i=1}^{N(x)} \sum_{j=1}^{N(x)} p_i p_j \kappa(\xi(x)_i, \xi(x)_j) - 2 \sum_{i=1}^{N(x)} p_i \overbrace{\int \kappa(\xi(x)_i, y) \varrho(\mathrm{d}y)}^{b \in \mathbf{R}^{N(x)}} + M(\kappa, \mathcal{R}, \varrho) \right\}$$

$$= \operatorname*{arginf}_{p \in \Delta_{\mathcal{R}(x)}} \left\{ p^\top K p - 2 p^\top b \right\},$$

where $M(\kappa, \mathcal{R}, \varrho)$ is independent of $p$, so it does not influence the minimization. Now, since $K$ is positive definite (by virtue of $\kappa$ being characteristic) and $\Delta_{\mathcal{R}(x)}$ is a closed convex subset of $\mathbf{R}^{N(x)}$, it is well-known that there is unique optimum, and the infimum above is attained for some $p^\star \in \Delta_{\mathcal{R}(x)}$. Therefore, $\Pi_{\mathrm{C},\kappa}^{\mathcal{R}}$ is indeed well-defined, and its range is contained in $\Pi_{\mathrm{C},\kappa}^{\mathcal{R}}$, confirming the first two claims. Finally, since $\Pi_{\mathrm{C},\kappa}^{\mathcal{R}}$ is well-defined and since $\mathrm{MMD}_\kappa$ is nonnegative and separates points,

$\Pi_{C,\kappa}^{\mathcal{R}}$ must map elements of $\Delta_{\mathcal{R}(x)}$ to themselves – this is because $\text{MMD}_\kappa(p,p) = 0$ for the kernels we consider. $\square$

**Lemma 2.** *Under the conditions of Lemma 1, $\Pi_{C,\kappa}^{\mathcal{R}}$ is a nonexpansion in $\overline{\text{MMD}_\kappa}$. That is, for any $\eta_1, \eta_2 \in \mathcal{P}([0, (1-\gamma)^{-1} R_{\max}]^d)^{\mathcal{X}}$, we have $\overline{\text{MMD}_\kappa}(\Pi_{C,\kappa}^{\mathcal{R}}\eta_1, \Pi_{C,\kappa}^{\mathcal{R}}\eta_2) \leq \overline{\text{MMD}_\kappa}(\eta_1, \eta_2)$.*

*Proof.* Fix any $x \in \mathcal{X}$ and denote $M(x) = \{\mu_p \in \mathcal{H} : p \in \Delta_{\mathcal{R}(x)}\}$, where $\mathcal{H}$ is the RKHS induced by the kernel $\kappa$ and $\mu_p$ denotes the mean embedding of $p$ in this RKHS. It is simple to verify that $p \mapsto \mu_p$ is linear: for any $p, q \in \mathcal{P}(\mathbf{R}^d)$ and $\alpha, \beta \in \mathbf{R}$, for all $f \in \mathcal{H}$ with $\|f\| = 1$ we have

$$\langle f, \mu_{\alpha p + \beta q} \rangle = \int f(y)[\alpha p + \beta q](\mathrm{d}y) = \alpha \int f(y)p(\mathrm{d}y) + \beta \int f(y)q(\mathrm{d}y)$$
$$= \langle a, \alpha \mu_p + \beta \mu_q \rangle,$$

which implies that $\mu_{\alpha p + \beta q} = \alpha \mu_p + \beta \mu_q$. As a consequence, $M(x)$ inherits convexity from $\Delta_{\mathcal{R}(x)}$.

We claim that $M(x)$ is closed as a subset of $\mathcal{H}$. Since $p \mapsto \mu_p$ is an injection [GBR$^+$12], by Lemma 1, since there is a unique $q \in \Delta_{\mathcal{R}(x)}$ minimizing $\text{MMD}_\kappa(p,q)$, there is a unique $\mu_q \in M(x)$ attaining the infimum $\|\mu_p - \mu_q\|$ over $M(x)$. Let $\mu \in \mathcal{H} \setminus M(x)$. Then there exists $\mu_q \in M(x)$ such that $\|\mu_q - \mu\| = \inf_{\nu \in M(x)} \|\mu - \nu\|$, and since $\mu_q \neq \mu$, it follows that $\inf_{\nu \in M(x)} \|\nu - \mu\| = \epsilon > 0$. Since this is true for any $\mu \notin M(x)$, it follows that $\mathcal{H} \setminus M(x)$ is open, so $M(x)$ is closed.

We will now show that $\eta(x) \mapsto \Pi_{C,\kappa}^{\mathcal{R}}\eta(x)$ is a nonexpansion in $\mathcal{H}$. Let $\eta_1, \eta_2 \in \mathscr{C}_{\mathcal{R}}$, and denote by $\mu_1(x), \mu_2(x)$ the mean embeddings of $\eta_1(x), \eta_2(x)$. We slightly abuse notation and write $\Pi_{C,\kappa}^{\mathcal{R}}\mu_i(x)$ to denote the mean embedding of $\Pi_{C,\kappa}^{\mathcal{R}}\eta_i(x)$. Since $M(x)$ is convex, for any $\iota(x) \in M(x)$ and $\lambda \in [0,1]$ we have

$$\text{MMD}_\kappa(\eta_1(x), \Pi_{C,\kappa}^{\mathcal{R}}\eta_1(x))^2 = \|\mu_1(x) - \Pi_{C,\kappa}^{\mathcal{R}}\mu_1(x)\|^2$$
$$\leq \|\mu_1(x) - (\lambda\iota(x) + (1-\lambda)\Pi_{C,\kappa}^{\mathcal{R}}\mu_1(x))\|^2$$
$$= \|\mu_1(x) - \Pi_{C,\kappa}^{\mathcal{R}}\mu_1(x) - \lambda(\iota(x) - \Pi_{C,\kappa}^{\mathcal{R}}\mu_1(x))\|^2.$$

Now, by expanding the squared norms and taking $\lambda \downarrow 0$, since $M(x)$ is closed we have

$$\langle \mu_1(x) - \Pi_{C,\kappa}^{\mathcal{R}}\mu_1(x), \iota_1(x) - \Pi_{C,\kappa}^{\mathcal{R}}\mu_1(x) \rangle \leq 0 \qquad \forall \iota_1(x), \iota_2(x) \in M(x)$$
$$\therefore \langle \Pi_{C,\kappa}^{\mathcal{R}}\mu_2(x) - \mu_2(x), \Pi_{C,\kappa}^{\mathcal{R}}\mu_2(x) - \iota_2(x) \rangle \leq 0,$$

where the second inequality follows by applying the same logic to $\mu_2(x)$. Choosing $\iota_1(x) = \Pi_{C,\kappa}^{\mathcal{R}}\mu_2(x), \iota_2(x) = \Pi_{C,\kappa}^{\mathcal{R}}\mu_1(x) \in M(x)$ and adding these inequalities yields

$$\langle \mu_1(x) - \mu_2(x) + (\Pi_{C,\kappa}^{\mathcal{R}}\mu_2(x) - \Pi_{C,\kappa}^{\mathcal{R}}\mu_1(x)), \Pi_{C,\kappa}^{\mathcal{R}}\mu_2(x) - \Pi_{C,\kappa}^{\mathcal{R}}\mu_1(x) \rangle \leq 0.$$

Expanding, we see that

$$\text{MMD}_\kappa(\Pi_{C,\kappa}^{\mathcal{R}}\eta_1(x), \Pi_{C,\kappa}^{\mathcal{R}}\eta_2(x))^2 = \|\Pi_{C,\kappa}^{\mathcal{R}}\mu_2(x) - \Pi_{C,\kappa}^{\mathcal{R}}\mu_1(x)\|^2$$
$$\leq \langle \mu_2(x) - \mu_1(x), \Pi_{C,\kappa}^{\mathcal{R}}\mu_2(x) - \Pi_{C,\kappa}^{\mathcal{R}}\mu_1(x) \rangle$$
$$\leq \|\mu_2(x) - \mu_1(x)\|\|\Pi_{C,\kappa}^{\mathcal{R}}\mu_2(x) - \Pi_{C,\kappa}^{\mathcal{R}}\mu_1(x)\|$$
$$= \text{MMD}_\kappa(\eta_1(x), \eta_2(x))\text{MMD}_\kappa(\Pi_{C,\kappa}^{\mathcal{R}}\eta_1(x), \Pi_{C,\kappa}^{\mathcal{R}}\eta_2(x))$$
$$\therefore \text{MMD}_\kappa(\Pi_{C,\kappa}^{\mathcal{R}}\eta_1(x), \Pi_{C,\kappa}^{\mathcal{R}}\eta_2(x)) \leq \text{MMD}_\kappa(\eta_1(x), \eta_2(x)),$$

confirming that $\eta(x) \mapsto \Pi_{C,\kappa}^{\mathcal{R}}\eta(x)$ is a non-expansion. It follows that

$$\overline{\text{MMD}_\kappa}(\Pi_{C,\kappa}^{\mathcal{R}}\eta_1, \Pi_{C,\kappa}^{\mathcal{R}}\eta_2) = \sup_{x \in \mathcal{X}} \text{MMD}_\kappa(\eta_1(x), \eta_2(x))$$
$$\leq \sup_{x \in \mathcal{X}} \text{MMD}_\kappa(\eta_1(x), \eta_2(x))$$
$$= \overline{\text{MMD}_\kappa}(\eta_1, \eta_2).$$

$\square$

**Corollary 2.** *Let $\kappa$ be a kernel satisfying the conditions of Theorem 2. Then for any $\eta_0 \in \mathscr{C}_\mathcal{R}$, the iterates $\{\eta_k\}_{k=1}^\infty$ given by $\eta_{k+1} = \Pi_{C,\kappa}^\mathcal{R} \mathcal{T}^\pi \eta_k$ converge geometrically to a unique fixed point.*

*Proof.* Combining Theorem 2 and Lemma 2, we see that

$$\overline{\mathrm{MMD}_\kappa}(\Pi_{C,\kappa}^\mathcal{R} \mathcal{T}^\pi \eta_1, \Pi_{C,\kappa}^\mathcal{R} \mathcal{T}^\pi \eta_2) \leq \overline{\mathrm{MMD}_\kappa}(\mathcal{T}^\pi \eta_1, \eta_2) \leq \gamma^{c/2} \overline{\mathrm{MMD}_\kappa}(\eta_1, \eta_2)$$

for some $c > 0$. Thus, $\Pi_{C,\kappa}^\mathcal{R} \mathcal{T}^\pi$ is a contraction on $(\mathscr{C}_\mathcal{R}, \overline{\mathrm{MMD}_\kappa})$. If $\mathcal{H}$ is the RKHS induced by $\kappa$, we showed in Lemma 2 that $\mathscr{C}_\mathcal{R}$ is isometric to a product of closed, convex subsets of $\mathcal{H}$. Therefore, by the completeness of $\mathcal{H}$, $\mathscr{C}_\mathcal{R}$ is isometric to a complete subspace, and consequently $\mathscr{C}_\mathcal{R}$ is a complete subspace under the metric $\overline{\mathrm{MMD}_\kappa}$. Invoking the Banach fixed-point theorem, it follows that $\Pi_{C,\kappa}^\mathcal{R} \mathcal{T}^\pi$ has a unique fixed point $\eta_{C,\kappa}^\pi$, and $\eta_k \to \eta_{C,\kappa}^\pi$ geometrically. $\square$

### B.3.1 Quality of the Categorical Fixed Point

As we saw in our analysis of multivariate DRL with EWP representations, the distance between a distribution and its projection (as a function of $m, d$) plays a major role in controlling the approximation error in projected distributional dynamic programming. Before proving the main results of this section, we begin by analyzing this quantity by reducing it to the largest distance between points among certain partitions of the space of returns.

**Lemma 3.** *Let $\kappa$ be kernel satisfying the conditions of Lemma 1, and for any finite $\mathcal{R} \subset \mathbf{R}^d$, define $\Pi : \mathcal{P}(\mathbf{R}^d) \to \Delta_\mathcal{R}$ via $\Pi p = \operatorname{arginf}_{q \in \Delta_\mathcal{R}} \mathrm{MMD}_\kappa(p, q)$. Then $\mathrm{MMD}_\kappa^2(\Pi p, p) \leq \inf_{P \in \mathscr{P}_\mathcal{R}} \mathsf{mesh}(P; \kappa)$.*

*Proof.* Our proof proceeds by establishing approximation bounds of Riemann sums to the Bochner integral $\mu_p$, similar to [VN02]. Let $P \in \mathscr{P}_\mathcal{R}$. Abusing notation to denote by $\Pi p_i$ the probability of the $i$th atom of the discrete support under $\Pi p$, we have

$$\begin{aligned}
\mathrm{MMD}_\kappa^2(p, \Pi p) &= \|\mu_{\Pi p} - \mu_p\|^2 \\
&= \left\| \int \kappa(\cdot, y) \Pi p(\mathrm{d}y) - \int \kappa(\cdot, y) p(\mathrm{d}y) \right\|^2 \\
&= \left\| \sum_i \kappa(\cdot, z_i) \Pi p_i - \int \kappa(\cdot, y) p(\mathrm{d}y) \right\|^2,
\end{aligned}$$

where $\mathcal{R} = \{z_i\}_{i=1}^n$ for some $n \in \mathbf{N}$. Since $\Pi p$ optimizes the MMD over all probability vectors in $\Delta_\mathcal{R}$, for $q \in \Delta_\mathcal{R}$ with $q_i = p(\theta_i)$ for the $i$th element of $P$, we have

$$\begin{aligned}
\mathrm{MMD}_\kappa^2(p, \Pi p) &\leq \left\| \sum_i \kappa(\cdot, z_i) p(\theta_i) - \int \kappa(\cdot, y) p(\mathrm{d}y) \right\|^2 \\
&= \left\| \sum_i \int_{\theta_i} (\kappa(\cdot, z_i) - \kappa(\cdot, y)) p(\mathrm{d}y) \right\|^2 \\
&\leq \left\| \sum_i \sup_{y_1, y_2 \in \theta_i} \|\kappa(\cdot, y_1) - \kappa(\cdot, y_2)\| p(\theta_i) \right\|^2 \\
&\leq \sup_{\theta \in P} \sup_{y_1, y_2 \in \theta} \|\kappa(\cdot, y_1) - \kappa(\cdot, y_2)\|^2.
\end{aligned}$$

It was shown by [SSGF13] that $z \mapsto \kappa(\cdot, z)$ is an isometry from $(\mathbf{R}^d, \rho^{1/2})$ to $\mathcal{H}$, where $\mathcal{H}$ is the RKHS induced by $\kappa$. Thus, we have

$$\mathrm{MMD}_\kappa^2(p, \Pi p) \leq \sup_{\theta \in P} \sup_{y_1, y_2 \in \theta} \rho(y_1, y_2) = \mathsf{mesh}(P; \kappa).$$

Since this is true for any partition $P \in \mathscr{P}_\mathcal{R}$, the claim follows by taking the infimum over $\mathscr{P}_\mathcal{R}$. $\square$

We now move on to the main results.

**Theorem 4.** *Let $\kappa$ be a kernel induced by a $c$-homogeneous and shift-invariant semimetric $\rho$ conforming to the conditions of Theorem 2. Then the fixed point $\eta^\pi_{C,\kappa}$ of $\Pi^\mathcal{R}_{C,\kappa}\mathcal{T}^\pi$ satisfies*

$$\overline{\mathrm{MMD}}_\kappa(\eta^\pi_{C,\kappa}, \eta^\pi) \leq \frac{1}{1 - \gamma^{c/2}} \sup_{x \in \mathcal{X}} \inf_{P \in \mathscr{P}_{\mathcal{R}(x)}} \sqrt{\mathrm{mesh}(P; \kappa)}. \tag{9}$$

*Proof.* The proof begins in a similar manner to [RBD$^+$18, Proposition 3]. Given that $\Pi^\mathcal{R}_{C,\kappa}$ is a nonexpansion as shown in Lemma 2, we have

$$
\begin{aligned}
\overline{\mathrm{MMD}}_\kappa(\eta^\pi_{C,\kappa}, \eta^\pi) &= \sup_{x \in \mathcal{X}} \mathrm{MMD}_\kappa(\eta^\pi_{C,\kappa}(x), \eta^\pi(x)) \\
&\leq \sup_{x \in \mathcal{X}} \left[ \mathrm{MMD}_\kappa(\eta^\pi_{C,\kappa}(x), \Pi^\mathcal{R}_{C,\kappa}\eta^\pi(x)) + \mathrm{MMD}_\kappa(\Pi^\mathcal{R}_{C,\kappa}\eta^\pi(x), \eta^\pi(x)) \right] \\
&\leq \overline{\mathrm{MMD}}_\kappa(\eta^\pi_{C,\kappa}, \Pi^\mathcal{R}_{C,\kappa}\eta^\pi) + \overline{\mathrm{MMD}}_\kappa(\Pi^\mathcal{R}_{C,\kappa}\eta^\pi, \eta^\pi) \\
&\overset{(a)}{=} \overline{\mathrm{MMD}}_\kappa(\Pi^\mathcal{R}_{C,\kappa}\mathcal{T}^\pi\eta^\pi_{C,\kappa}, \Pi^\mathcal{R}_{C,\kappa}\mathcal{T}^\pi\eta^\pi) + \overline{\mathrm{MMD}}_\kappa(\Pi^\mathcal{R}_{C,\kappa}\eta^\pi, \eta^\pi) \\
&\overset{(b)}{\leq} \overline{\mathrm{MMD}}_\kappa(\mathcal{T}^\pi\eta^\pi_{C,\kappa}, \mathcal{T}^\pi\eta^\pi) + \overline{\mathrm{MMD}}_\kappa(\Pi^\mathcal{R}_{C,\kappa}\eta^\pi, \eta^\pi) \\
&\overset{(c)}{\leq} \gamma^{c/2}\overline{\mathrm{MMD}}_\kappa(\eta^\pi_{C,\kappa}, \eta^\pi) + \overline{\mathrm{MMD}}_\kappa(\Pi^\mathcal{R}_{C,\kappa}\eta^\pi, \eta^\pi) \\
\therefore \overline{\mathrm{MMD}}_\kappa(\eta^\pi_{C,\kappa}, \eta^\pi) &\leq \frac{1}{1 - \gamma^{c/2}}\overline{\mathrm{MMD}}_\kappa(\Pi^\mathcal{R}_{C,\kappa}\eta^\pi, \eta^\pi),
\end{aligned}
$$

where $(a)$ leverages the fact that $\eta^\pi_{C,\kappa}$ is the fixed point of $\Pi^\mathcal{R}_{C,\kappa}\mathcal{T}^\pi$ and that $\eta^\pi$ is the fixed point of $\mathcal{T}^\pi$, $(b)$ follows since $\Pi^\mathcal{R}_{C,\kappa}$ is a nonexpansion by Lemma 2, and $(c)$ follows by the contractivity of $\mathcal{T}^\pi$ established in Theorem 2. Finally, by Lemma 3, we have

$$\overline{\mathrm{MMD}}_\kappa(\eta^\pi_{C,\kappa}, \eta^\pi) \leq \frac{1}{1 - \gamma^{c/2}} \sup_{x \in \mathcal{X}} \mathrm{MMD}_\kappa(\Pi^\mathcal{R}_{C,\kappa}\eta^\pi(x), \eta^\pi(x)) \leq \frac{1}{1 - \gamma^{c/2}} \sup_{x \in \mathcal{X}} \inf_{P \in \mathscr{P}_{\mathcal{R}(x)}} \sqrt{\mathrm{mesh}(P; \kappa)}.$$

$\square$

Finally, we explicitly derive a convergence rate for a particular support map under the energy distance kernels.

**Corollary 3.** *Let $\mathcal{R}(x) = U_m$, where $U_m$ is a set of $m$ uniformly-spaced support points on $[0, (1 - \gamma)^{-1}R_{\max}]$. For $\kappa$ induced by the semimetric $\rho(x, y) = \|x - y\|_2^\alpha$ for $\alpha \in (0, 2)$,*

$$\overline{\mathrm{MMD}}_\kappa(\eta^\pi_{C,\kappa}, \eta^\pi) \leq \frac{1}{(1 - \gamma^{\alpha/2})(1 - \gamma)^{\alpha/2}} \frac{d^{\alpha/4}R_{\max}^{\alpha/2}}{(m^{1/d} - 2)^{\alpha/2}}.$$

*Proof.* We begin bounding $\mathrm{mesh}(P; \kappa)$. Assume $m = n^d$ for some $n \in \mathbf{N}$. We consider a partition $\overline{P} \subset \mathscr{P}_{U_m}$ consisting of $d$-dimensional hypercubes with side length $(1 - \gamma)^{-1}R_{\max}/(n - 1)$. By definition of $U_m$, it is clear that these hypercubes cover $[0, (1 - \gamma)^{-1}R_{\max}]^d$ and each contain exactly one support point. Now, for each $\theta \in \overline{P}$, we have

$$\sup_{y_1, y_2 \in \theta} \rho(y_1, y_2) \leq \|y - (y + (1 - \gamma)^{-1}R_{\max}/(n - 1)\vec{1}\|_2^\alpha$$

where $\vec{1}$ is the vector of all ones, and $y$ is any element in $\theta$. Expanding, we have

$$
\begin{aligned}
\sup_{y_1, y_2 \in \theta} \rho(y_1, y_2) &\leq (1 - \gamma)^{-\alpha} \left( \sum_{i=1}^d \left( \frac{R_{\max}}{n - 1} \right)^2 \right)^{\alpha/2} \\
&\leq \frac{d^{\alpha/2}R_{\max}^\alpha}{(1 - \gamma)^\alpha(n - 1)^\alpha}.
\end{aligned}
$$

Since this bound holds for any $\theta \in \overline{P}$, invoking Theorem 4 yields

$$\overline{\mathrm{MMD}}_\kappa(\eta_{\mathrm{C},\kappa}^\pi, \eta^\pi) \le \frac{1}{1-\gamma^{\alpha/2}} \sup_{x\in\mathcal{X}} \inf_{P\in\mathscr{P}_{U_m}} \sqrt{\mathsf{mesh}(P;\kappa)}$$

$$\le \frac{1}{1-\gamma^{\alpha/2}} \sup_{x\in\mathcal{X}} \sqrt{\mathsf{mesh}(\overline{P};\kappa)}$$

$$\le \frac{1}{1-\gamma^{\alpha/2}} \sup_{x\in\mathcal{X}} \sqrt{\frac{d^{\alpha/2} R_{\max}^\alpha}{(1-\gamma)^\alpha (n-2)^\alpha}}$$

$$= \frac{1}{(1-\gamma^{\alpha/2})(1-\gamma)^{\alpha/2}} \frac{d^{\alpha/4} R_{\max}^{\alpha/2}}{(n-1)^{\alpha/2}}$$

$$= \frac{1}{(1-\gamma^{\alpha/2})(1-\gamma)^{\alpha/2}} \frac{d^{\alpha/4} R_{\max}^{\alpha/2}}{(n-1)^{\alpha/2}}$$

$$= \frac{1}{(1-\gamma^{\alpha/2})(1-\gamma)^{\alpha/2}} \frac{d^{\alpha/4} R_{\max}^{\alpha/2}}{(m^{1/d}-1)^{\alpha/2}}.$$

If instead $m \in ((n-1)^d, n^d)$, we omit all but $(n-1)^d$ of the support points, and achieve

$$\overline{\mathrm{MMD}}_\kappa(\eta_{\mathrm{C},\kappa}^\pi, \eta^\pi) \le \frac{1}{(1-\gamma^{\alpha/2})(1-\gamma)^{\alpha/2}} \frac{d^{\alpha/4} R_{\max}^{\alpha/2}}{(\lfloor m^{1/d}\rfloor - 1)^{\alpha/2}}.$$

Alternatively, we may write

$$\overline{\mathrm{MMD}}_\kappa(\eta_{\mathrm{C},\kappa}^\pi, \eta^\pi) \le \frac{1}{(1-\gamma^{\alpha/2})(1-\gamma)^{\alpha/2}} \frac{d^{\alpha/4} R_{\max}^{\alpha/2}}{(m^{1/d}-2)^{\alpha/2}}.$$

$\square$

## B.4 Categorical TD Learning: Section 6

In this section, we prove results leading up to and including the convergence of the categorical TD-learning algorithm over mass-1 signed measures. First, in Section B.4.1, we show that $\mathrm{MMD}_\kappa$ is in fact a metric on the space of mass-1 signed measures, and establish that the multivariate distributional Bellman operator is contractive under these distribution representations. Subsequently, in Section B.4.2, we analyze the temporal difference learning algorithm leveraging the results from Section B.4.1.

### B.4.1 The Signed Measure Relaxation

We begin by establishing that $\mathrm{MMD}_\kappa$ is a metric on $\mathcal{M}^1(\mathcal{Y})$ for spaces $\mathcal{Y}$ under which $\mathrm{MMD}_\kappa$ is a metric on $\mathcal{P}(\mathcal{Y})$.

**Lemma 4.** *Let $\kappa : \mathcal{Y} \times \mathcal{Y} \to \mathbf{R}$ be a characteristic kernel over some space $\mathcal{Y}$. Then $\mathrm{MMD}_\kappa :$ $\mathcal{M}^1(\mathcal{Y}) \times \mathcal{M}^1(\mathcal{Y}) \to \mathbf{R}_+$ given by $(p,q) \mapsto \|\mu_p - \mu_q\|_\mathcal{H}$ defines a metric on $\mathcal{M}^1(\mathcal{Y})$, where $\mu_p$ denotes the usual mean embedding of $p$ and $\mathcal{H}$ is the RKHS with kernel $\kappa$.*

*Proof.* Naturally, since $\mathrm{MMD}_\kappa$ is given by a norm, it is non-negative, symmetric, and satisfies the triangle inequality. We must show that $\mathrm{MMD}_\kappa(p,q) = 0 \iff p = q$. Firstly, it is clear that $\mathrm{MMD}_\kappa(p,p) = 0$ by the positive homogeneity of the norm. It remains to show that $\mathrm{MMD}_\kappa(p,q) = 0 \implies p = q$.

Let $p \ne q \in \mathcal{M}^1(\mathcal{Y})$. For the sake of contradiction, assume that $\mathrm{MMD}_\kappa(p,q) = 0$. We will show that this implies that $\mathrm{MMD}_\kappa(P,Q) = 0$ for a pair of distinct probability measures, which is a contradiction since $\mathrm{MMD}_\kappa$ with characteristic $\kappa$ is known to be a metric on $\mathcal{P}(\mathcal{Y})$.

By the Hahn decomposition theorem, we may write $p = \tilde{p}^+ - \tilde{p}^-$ for non-negative measures $\tilde{p}^+, \tilde{p}^-$. Therefore, for some $a \in \mathbf{R}_+$, we may express

$$p = (a+1)p^+ - ap^-$$

where $p^+, p^- \in \mathcal{P}(\mathcal{Y})$. Likewise, there exist $b \in \mathbf{R}_+$ and probability measure $q^+, q^-$ for which $q = (b+1)q^+ - bq^-$. Since $\mathrm{MMD}_\kappa(p, q) = 0$ by hypothesis, and by linearity of $p \mapsto \mu_p$, we have

$$
\begin{aligned}
0 &= \|\mu_p - \mu_q\|_\mathcal{H} \\
&= \|(a+1)\mu_{p^+} - a\mu_{p^-} + b\mu_{q^-} - (b+1)\mu_{q^+}\|_\mathcal{H} \\
&= \|(a+1)\mu_{p^+} + b\mu_{q^-} - (a\mu_{p^-} + (b+1)\mu_{q^+})\|_\mathcal{H} \\
&= (a+b+1)\left\|\left(\lambda\mu_{p^+} + (1-\lambda)\mu_{q^-}\right) - \left(\lambda'\mu_{p^-} + (1-\lambda')\mu_{q^+}\right)\right\|, \quad \lambda = \frac{a+1}{a+b+1}, \quad \lambda' = \frac{a}{a+b+1} \\
&:= (a+b+1)\|\mu_P - \mu_Q\|_\mathcal{H},
\end{aligned}
$$

where $P = \lambda p^+ + (1-\lambda)q^-$ and $Q = \lambda'p^- + (1-\lambda')q^+$ are convex combinations of probability measures, and are therefore probability measures themselves. So, we have that

$$
\begin{aligned}
\lambda p^+ - \lambda'p^- &= (1-\lambda')q^+ - (1-\lambda)q^- \\
(a+1)\lambda p^+ - ap^- &= (b+1)q^+ - bq^- \\
\therefore p &= q,
\end{aligned}
$$

which contradicts our hypothesis. Therefore, $\mathrm{MMD}_\kappa(p, q) = 0 \iff p = q$ for any $p, q \in \mathcal{M}^1(\mathcal{Y})$, and it follows that $\mathrm{MMD}_\kappa$ is a metric. $\qquad\square$

Next, we show that the distributional Bellman operator is contractive on the space of mass-1 signed measure return distribution representations.

**Lemma 5.** *Under the conditions of Corollary 2, the projected operator $\Pi_{\mathrm{SC},\kappa}^\mathcal{R}\mathcal{T}^\pi : \mathscr{S}_\mathcal{R} \to \mathscr{S}_\mathcal{R}$ is affine, is contractive with contraction factor $\gamma^{c/2}$, and has a unique fixed point $\eta_{\mathrm{SC},\kappa}^\pi$.*

*Proof.* Indeed, $\Pi_{\mathrm{SC},\kappa}^\mathcal{R}$ is, in a sense, a simpler operator than $\Pi_{\mathrm{C},\kappa}^\mathcal{R}$. Since $\mathcal{M}^1(\mathcal{R}(x))$ is an affine subspace of $\mathcal{M}^1(\mathbf{R}^d)$, it holds that $\Pi_{\mathrm{SC},\kappa}^\mathcal{R}$ is simply a Hilbertian projection, which is known to be affine and a nonexpansion [Lax02]. Moreover, since $\mathcal{T}^\pi$ acts identically on $\mathcal{M}^1(\mathbf{R}^d)$ as it does on $\mathcal{P}(\mathbf{R}^d)$, it immediately follows that $\mathcal{T}^\pi$ is a $\gamma^{c/2}$-contraction on $\mathcal{M}^1(\mathbf{R}^d)$, inheriting the result from Theorem 2. Thus, we have that for any $\eta_1, \eta_2 \in \mathscr{S}_\mathcal{R}$,

$$
\begin{aligned}
\mathrm{MMD}_\kappa(\Pi_{\mathrm{SC},\kappa}^\mathcal{R}\mathcal{T}^\pi\eta_1, \Pi_{\mathrm{SC},\kappa}^\mathcal{R}\mathcal{T}^\pi\eta_2) &\leq \mathrm{MMD}_\kappa(\mathcal{T}^\pi\eta_1, \mathcal{T}^\pi\eta_2) \\
&\leq \gamma^{c/2}\mathrm{MMD}_\kappa(\eta_1, \eta_2)
\end{aligned}
$$

confirming that the projected operator is a contraction. Since $\mathrm{MMD}_\kappa$ is a metric on $\mathcal{M}^1(\mathcal{R}(x))$ for each $x \in \mathcal{X}$, it follows that $\overline{\mathrm{MMD}_\kappa}$ is a metric on $\mathscr{S}_\mathcal{R}$. The existence and uniqueness of the fixed point $\eta_{\mathrm{SC},\kappa}^\pi$ follows by the Banach fixed point theorem. $\qquad\square$

Finally, we show that the fixed point of distributional dynamic programming with signed measure representations is roughly as accurate as $\eta_{\mathrm{C},\kappa}^\pi$.

**Theorem 5.** *Under the conditions of Lemma 5, we have that*

1. $\overline{\mathrm{MMD}_\kappa}(\eta_{\mathrm{SC},\kappa}^\pi, \eta^\pi) \leq \frac{1}{1-\gamma^{c/2}} \sup_{x \in \mathcal{X}} \inf_{P \in \mathscr{P}_{\mathcal{R}(x)}} \sqrt{\mathsf{mesh}(P; \kappa)}$; and

2. $\overline{\mathrm{MMD}_\kappa}(\Pi_{\mathrm{C},\kappa}^\mathcal{R}\eta_{\mathrm{SC},\kappa}^\pi, \eta^\pi) \leq (1 + \frac{1}{1-\gamma^{c/2}}) \sup_{x \in \mathcal{X}} \inf_{P \in \mathscr{P}_{\mathcal{R}(x)}} \sqrt{\mathsf{mesh}(P; \kappa)}$.

*Proof.* Since $\Pi_{\mathrm{SC},\kappa}^\mathcal{R}$ is a nonexpansion in $\overline{\mathrm{MMD}_\kappa}$ by Lemma 5, following the procedure of Theorem 4, we have

$$
\overline{\mathrm{MMD}_\kappa}(\eta_{\mathrm{SC},\kappa}^\pi, \eta^\pi) \leq \frac{1}{1-\gamma^{c/2}}\overline{\mathrm{MMD}_\kappa}(\Pi_{\mathrm{SC},\kappa}^\mathcal{R}\eta^\pi, \eta^\pi).
$$

Note that $\Pi_{\mathrm{SC},\kappa}^\mathcal{R}\eta^\pi$ identifies the closest point (in $\overline{\mathrm{MMD}_\kappa}$) to $\eta^\pi$ in $\mathscr{S}_\mathcal{R} := \prod_{x \in \mathcal{X}} \mathcal{M}^1(\mathcal{R}(x))$ and $\Pi_{\mathrm{C},\kappa}^\mathcal{R}\eta^\pi$ identifies the closest point to $\eta^\pi$ in $\mathscr{C}_\mathcal{R} := \prod_{x \in \mathcal{X}} \Delta_{\mathcal{R}(x)}$. Since it is clear that $\mathscr{C}_\mathcal{R} \subset \mathscr{S}_\mathcal{R}$, it follows that

$$
\overline{\mathrm{MMD}_\kappa}(\eta_{\mathrm{SC},\kappa}^\pi, \eta^\pi) \leq \frac{1}{1-\gamma^{c/2}}\overline{\mathrm{MMD}_\kappa}(\Pi_{\mathrm{C},\kappa}^\mathcal{R}\eta^\pi, \eta^\pi).
$$

The first statement then directly follows since it was shown in Lemma 3 that $\overline{\mathrm{MMD}_\kappa}(\Pi^{\mathcal{R}}_{C,\kappa}\eta^\pi, \eta^\pi) \leq \sup_{x\in\mathcal{X}}\inf_{P\in\mathscr{P}_{\mathcal{R}(x)}}\sqrt{\mathsf{mesh}(P;\kappa)}$.

To prove the second statement, we apply the triangle inequality to yield

$$\overline{\mathrm{MMD}_\kappa}(\Pi^{\mathcal{R}}_{C,\kappa}\eta^\pi_{\mathrm{SC},\kappa}, \eta^\pi) \leq \overline{\mathrm{MMD}_\kappa}(\Pi^{\mathcal{R}}_{C,\kappa}\eta^\pi_{\mathrm{SC},\kappa}, \Pi^{\mathcal{R}}_{C,\kappa}\eta^\pi) + \overline{\mathrm{MMD}_\kappa}(\Pi^{\mathcal{R}}_{C,\kappa}\eta^\pi, \eta^\pi)$$

$$\leq \overline{\mathrm{MMD}_\kappa}(\eta^\pi_{\mathrm{SC},\kappa}, \eta^\pi) + \overline{\mathrm{MMD}_\kappa}(\Pi^{\mathcal{R}}_{C,\kappa}\eta^\pi, \eta^\pi),$$

where the second step leverages the fact that $\Pi^{\mathcal{R}}_{C,\kappa}$ is a nonexpansion in $\overline{\mathrm{MMD}_\kappa}$ by Lemma 2. Applying the conclusion of the first statement as well as the bound on $\overline{\mathrm{MMD}_\kappa}(\Pi^{\mathcal{R}}_{C,\kappa}\eta^\pi, \eta^\pi)$, we have

$$\overline{\mathrm{MMD}_\kappa}(\Pi^{\mathcal{R}}_{C,\kappa}\eta^\pi_{\mathrm{SC},\kappa}, \eta^\pi) \leq \frac{1}{1-\gamma^{c/2}}\sup_{x\in\mathcal{X}}\inf_{P\in\mathscr{P}_{\mathcal{R}(x)}}\sqrt{\mathsf{mesh}(P;\kappa)} + \sup_{x\in\mathcal{X}}\inf_{P\in\mathscr{P}_{\mathcal{R}(x)}}\sqrt{\mathsf{mesh}(P;\kappa)}$$

$$= \left(1 + \frac{1}{1-\gamma^{c/2}}\right)\sup_{x\in\mathcal{X}}\inf_{P\in\mathscr{P}_{\mathcal{R}(x)}}\sqrt{\mathsf{mesh}(P;\kappa)}.$$

$\square$

### B.4.2 Convergence of Categorical TD Learning

Convergence of the proposed categorical TD-learning algorithm will rely on studying the iterates through an isometry to an affine subspace of $\prod_{x\in\mathcal{X}}\mathbf{R}^{N(x)}$. This affine subspace is that consisting of the set of state-conditioned "signed probability vectors". We define $\mathbf{R}^n_1$ as an affine subspace of $\mathbf{R}^n$ for any $n\in\mathbf{N}$ according to

$$\mathbf{R}^n_1 = \left\{v\in\mathbf{R}^n : \sum_{i=1}^n v_i = 1\right\}. \tag{15}$$

We note that any element $\eta$ of $\mathscr{S}_\mathcal{R}$ can be encoded in $\prod_{x\in\mathcal{X}}\mathbf{R}^{N(x)}_1$ by expresing $\eta(x)$ as the sequence of signed masses associated to each atom of $\mathcal{R}(x)$. In Lemma 8, we exhibit an isometry $\mathcal{I}$ between $\mathscr{S}_\mathcal{R}$ and $\prod_{x\in\mathcal{X}}\mathbf{R}^{N(x)}_1$.

**Lemma 8.** *Let $\kappa$ be a characteristic kernel. There exists an affine isometric isomorphism $\mathcal{I}$ between $\mathscr{S}_\mathcal{R}$ and an affine subspace $\prod_{x\in\mathcal{X}}\mathbf{R}^{N(x)}_1$ (cf. (15)).*

*Proof.* Since $\kappa$ is characteristic, it is positive definite [GBR+12]. Thus, for each $x\in\mathcal{X}$, define $K_x\in\mathbf{R}^{N(x)\times N(x)}$ according to

$$(K_x)_{i,j} = \kappa(z_i, z_j) \qquad \mathcal{R}(x) = \{z_k\}^{N(x)}_{k=1}.$$

Then each $K_x$ is positive definite since $\kappa$ is a positive definite kernel. Let $p, q\in\Delta_{\mathcal{R}(x)}$, and define $P\in\mathbf{R}^{N(x)}$ and $Q\in\mathbf{R}^{N(x)}$ such that $P_i = p(z_i)$ and $Q_i = q(z_i)$. Then, we have

$$\mathrm{MMD}^2_\kappa(p,q) = \|\mu_p - \mu_q\|^2_{\mathcal{H}}$$

$$= \left\|\sum_{i=1}^{N(x)}\kappa(\cdot, z_i)p(z_i) - \sum_{i=1}^{N(x)}\kappa(\cdot, z_i)q(z_i)\right\|^2_{\mathcal{H}}$$

$$= \left\langle\sum_{i=1}^{N(x)}\kappa(\cdot, z_i)(p(z_i) - q(z_i)), \sum_{j=1}^{N(x)}\kappa(\cdot, z_j)(p(z_j) - q(z_j))\right\rangle_{\mathcal{H}}$$

$$= \sum_{i=1}^{N(x)}\sum_{j=1}^{N(x)}(p(z_i) - q(z_i))(p(z_j) - q(z_j))\kappa(z_i, z_j)$$

$$= (P - Q)^\top K_x(P - Q)$$

$$= \|P - Q\|^2_{K_x}.$$

Since $K_x$ is positive definite, $\|\cdot\|_{K_x}$ is a norm on $\mathbf{R}^{N(x)}$. Therefore, the map $\mathcal{I}^1_x : (\Delta_{\mathcal{R}(x)}, \mathrm{MMD}_\kappa) \to (\mathbf{R}^{N(x)}, \|\cdot\|_{K_x})$ given by $\mathcal{I}^1_x(p) = \overline{P}$ is a linear isometric isomorphism

onto the affine subspace of $\mathbf{R}^{N(x)}$ with entries summing to 1, which we denote $\mathbf{R}_1^{N(x)}$. Moreover, since $(\mathbf{R}_1^{N(x)}, \|\cdot\|_{K_x})$ is a finite dimensional Hilbert space, it is well known that there exists a linear isometric isomorphism $\mathcal{I}_x^2 : (\mathbf{R}_1^{N(x)}, \|\cdot\|_{K_x}) \to \mathbf{R}_1^{N(x)}$ with the usual $L^2$ norm. Thus, $\mathcal{I}_x = \mathcal{I}_x^2 \mathcal{I}_x^1 : (\Delta_{\mathcal{R}(x)}, \mathrm{MMD}_\kappa) \to \mathbf{R}_1^{N(x)}$ is a linear isometric isomorphism. Consequently, it follows that $\mathcal{I} : (\mathscr{C}_\mathcal{R}, \overline{\mathrm{MMD}_\kappa}) \to \prod_{x \in \mathcal{X}} \mathbf{R}_1^{N(x)}$ given by $\mathcal{I} = (\prod_{x \in \mathcal{X}} \mathbf{R}^{N(x)}, \|\cdot\|_{2,\infty})$ is a linear isometric isomorphism, where $\|v\|_{2,\infty} = \sup_{x \in \mathcal{X}} \|v(x)\|_2$. $\qquad\square$

**Lemma 9.** *Define the operator* $\mathcal{U}^\pi : \prod_{x \in \mathcal{X}} \mathbf{R}_1^{N(x)} \to \prod_{x \in \mathcal{X}} \mathbf{R}_1^{N(x)}$ *by* $\mathcal{U}^\pi = \mathcal{I}\Pi_{\mathrm{SC},\kappa}^{\mathcal{R}} \mathcal{T}^\pi \mathcal{I}^{-1}$, *where* $\mathcal{I}$ *is the isometry of Lemma* 8. *Let* $\{U_t\}_{t=1}^\infty$ *be given by* $U_t = \mathcal{I}\eta_t$, *where* $\{\eta_t\}_{t=1}^\infty$ *are the dynamic programming iterates* $\eta_{t+1} = \Pi_{\mathrm{SC},\kappa}^{\mathcal{R}} \mathcal{T}^\pi \eta_t$. *Then* $U_{t+1} = \mathcal{U}^\pi U_t$. *Moreover,* $\mathcal{U}^\pi$ *is contractive whenever* $\Pi_{\mathrm{SC},\kappa}^{\mathcal{R}} \mathcal{T}^\pi$ *is, and in this case,* $U_t \to U^\star$, *where* $U^\star$ *is the unique fixed point of* $\mathcal{U}^\pi$.

*Proof.* By definition, we have

$$\begin{aligned} U_{t+1} &= \mathcal{I}\eta_{t+1} \\ &= \mathcal{I}(\Pi_{\mathrm{C},\kappa}^{\mathcal{R}} \mathcal{T}^\pi \eta_t) \\ &= \mathcal{I}\Pi_{\mathrm{C},\kappa}^{\mathcal{R}} \mathcal{T}^\pi (\mathcal{I}^{-1} U_t) \\ &= \mathcal{U}^\pi U_t, \end{aligned}$$

which proves the first claim. Moreover, for $U_1 = \mathcal{I}\eta_1$ and $U_2 = \mathcal{I}\eta_2$, we have

$$\begin{aligned} \|\mathcal{U}^\pi U_1 - \mathcal{U}^\pi U_2\|_{2,\infty} &= \left\| \mathcal{I}\Pi_{\mathrm{C},\kappa}^{\mathcal{R}} \mathcal{T}^\pi \eta_1 - \mathcal{I}\Pi_{\mathrm{C},\kappa}^{\mathcal{R}} \mathcal{T}^\pi \eta_2 \right\|_{2,\infty} \\ &= \overline{\mathrm{MMD}_\kappa}(\Pi_{\mathrm{C},\kappa}^{\mathcal{R}} \mathcal{T}^\pi \eta_1, \Pi_{\mathrm{C},\kappa}^{\mathcal{R}} \mathcal{T}^\pi \eta_2), \end{aligned}$$

where the second step transforms the $\|\cdot\|_{2,\infty}$ to $\overline{\mathrm{MMD}_\kappa}$ since $\mathcal{I}$ is an isometry between those metric spaces. Therefore, if $\Pi_{\mathrm{C},\kappa}^{\mathcal{R}} \mathcal{T}^\pi$ is contractive with contraction factor $\beta \in (0,1)$, we have

$$\begin{aligned} \|\mathcal{U}^\pi U_1 - \mathcal{U}^\pi U_2\|_{2,\infty} &\leq \beta \overline{\mathrm{MMD}_\kappa}(\eta_1, \eta_2) \\ &= \beta \|\mathcal{I}\eta_1 - \mathcal{I}\eta_2\|_{2,\infty} \\ &= \beta \|U_1 - U_2\|_{2,\infty}, \end{aligned}$$

so that $\mathcal{U}^\pi$ has the same contraction factor as $\Pi_{\mathrm{C},\kappa}^{\mathcal{R}} \mathcal{T}^\pi$. Consequently, by the Banach fixed point theorem, $\mathcal{U}^\pi$ has a unique fixed point $U^\star$ whenever $\Pi_{\mathrm{C},\kappa}^{\mathcal{R}} \mathcal{T}^\pi$ is contractive, and $U_t \to U^\star$ at the same rate as $\eta_t \to \eta^\pi$. $\qquad\square$

The main theorem in this section is that temporal difference learning on the finite dimensional representations $\mathcal{I}(\eta_t)$ converges.

**Proposition 2** (Convergence of categorical temporal difference learning)**.** *Let* $\{U_t\}_{t=1}^\infty \subset \prod_{x \in \mathcal{X}} \mathbf{R}_1^{N(x)}$ *be given by* $U_t = \mathcal{I}\widehat{\eta}_t$, *and let* $\kappa$ *be a kernel satisfying the conditions of Theorem* 2. *Suppose that, for each* $x \in \mathcal{X}$, *the states* $\{X_t\}_{t=1}^\infty$ *and step sizes* $\{\alpha_t\}_{t=1}^\infty$ *satisfy the Robbins-Munro conditions*

$$\sum_{t=0}^\infty \alpha_t \mathbf{1}_{[X_t = x]} = \infty \qquad \sum_{t=0}^\infty \alpha_t^2 \mathbf{1}_{[X_t = x]} < \infty.$$

*Then, with probability 1,* $U_k \to U^\star$, *where* $U^\star$ *is the fixed point of* $\mathcal{U}^\pi$.

The proof of this result as a natural extension of the convergence analysis of Categorical TD Learning given in [BDR23] to the multivariate return setting under the supremal MMD metric. The analysis hinges on the following general lemma.

**Lemma 10** ([BDR23, Theorem 6.9])**.** *Let* $\mathcal{O} : (\mathbf{R}^n)^\mathcal{X} \to (\mathbf{R}^n)^\mathcal{X}$ *be a contractive operator with respect to* $\|\cdot\|_{2,\infty}$ *with fixed point* $Z^\star$, *and let* $(\Omega, \mathcal{F}, \{F_k\}_{k=1}^\infty, \mathbf{P})$ *be a filtered probability space. Define a map* $\widehat{\mathcal{O}} : (\mathbf{R}^n)^\mathcal{X} \times \mathcal{X} \times \Omega \to (\mathbf{R}^n)^\mathcal{X}$ *such that*

$$\mathbf{E_P}\left[\widehat{\mathcal{O}}(Z, X, \omega) \mid X = x\right] = (\mathcal{O}Z)(x). \tag{16}$$

*For a stochastic process $\{\xi_k\}_{k=1}^\infty$ adapted to $\{\mathcal{F}_k\}_{k=1}^\infty$ with $\xi_k = X_k \oplus \omega_k$, consider a sequence $\{Z_k\}_{k=1}^\infty \subset (\mathbf{R}^n)^\mathcal{X}$ given by*

$$Z_{k+1}(x) = \begin{cases} (1 - \alpha_k)Z_k(x) + \alpha_k\widehat{\mathcal{O}}(Z_k, X_k, \omega_k)(x) & X_k = x \\ Z_k(x) & otherwise \end{cases} \tag{17}$$

*where $\{\alpha_k\}_{k=1}^\infty$ is adapted to $\{\mathcal{F}_k\}_{k=1}^\infty$ and satisfies the Robbins-Munro conditions for each $x \in \mathcal{X}$,*

$$\sum_{k=1}^\infty \alpha_k \mathbf{1}_{[X_t=x]} = \infty, \qquad \sum_{k=1}^\infty \alpha_k^2 \mathbf{1}_{[X_t=x]} < \infty.$$

*Finally, for the processes $\{w(x)_k\}_{k=1}^\infty$ where $w(x)_k = (\widehat{\mathcal{O}}(Z_k, X_k, \omega_k) - (\mathcal{O}Z_k)(X_k))\mathbf{1}_{[X_k=x]}$, assume the following moment condition holds,*

$$\mathbf{E_P}\left[\|w(x)_k\|^2 \mid \mathcal{F}_k\right] \le C_1 + C_2\|Z_k\|_{2,\infty}^2 \tag{18}$$

*for finite universal constants $C_1, C_2$. Then, with probability 1, $Z_k \to Z^\star$.*

The operator $\mathcal{O}$ of Lemma 10 will be substituted with $\mathcal{U}^\pi$, governing the dynamics of the encoded iterates of the multi-return distribution. The stochastic operator $\widehat{\mathcal{O}}$ will be substituted with the corresponding stochastic TD operator for $\mathcal{U}^\pi$, given by

$$\widehat{\mathcal{U}}^\pi(U, x_1, (R, x_2))(x) = \begin{cases} \mathcal{I}\left(\Pi_{\mathrm{SC},\kappa}^\mathcal{R}(\mathrm{b}_{R,\gamma})_\sharp\mathcal{I}^{-1}U(x_2)\right)(x_1) & x_1 = x \\ U(x) & \text{otherwise.} \end{cases} \tag{19}$$

This corresponds to applying a Bellman backup from a stochastic reward $R$ and next state $x_2$, followed by projecting back onto $\mathscr{S}_\mathcal{R}$, and applying the isometry back to $\prod_{x \in \mathcal{X}} \mathbf{R}_1^{N(x)}$.

*Proof of Proposition 2.* Let $n = \max_{x \in \mathcal{X}} N(x)$. Note that for any $m \le n$, $\mathbf{R}^m$ can be isometrically embedded into $\mathbf{R}^n$ by zero-padding. Thus, $\prod_{x \in \mathcal{X}} \mathbf{R}_1^{N(x)}$ can be isometrically embedded into $(\mathbf{R}_1^n)^\mathcal{X}$, so without loss of generality, we will assume that $N(x) \equiv n$.

Define the map $\widehat{\mathcal{U}}^\pi : (\mathbf{R}_1^n)^\mathcal{X} \times \mathcal{X} \times (\mathbf{R}^d \times \mathcal{X}) \to (\mathbf{R}_1^n)^\mathcal{X}$ according to

$$(\widehat{\mathcal{U}}^\pi(U, x_1, (R, x_2)))(x) = \begin{cases} \mathcal{I}(\Pi_{\mathrm{SC},\kappa}^\mathcal{R}(\mathrm{b}_{R,\gamma})_\sharp\mathcal{I}^{-1}U(x_2))(x_1) & x_1 = x \\ U(x) & \text{otherwise} \end{cases} \tag{20}$$

Then, defining $\widehat{\mathcal{U}}_k^\pi U = \widehat{\mathcal{U}}^\pi(U, X_k, (R_k, X_k'))$ with $(X_k, A_k, R_k, X_k') = T_k \sim \mathbf{P}$ as in (11), we have

$$\begin{aligned} U_{k+1}(x) &= (\mathcal{I}\widehat{\mathcal{T}}^\pi\widehat{\eta}_{k+1})(x) \\ &= \mathcal{I}\left(\mathbf{1}_{[X_k=x]}\Pi_{\mathrm{SC},\kappa}^\mathcal{R}(\mathrm{b}_{R_k,\gamma})_\sharp\widehat{\eta}_k(X_k') + \mathbf{1}_{[X_k\neq x]}\widehat{\eta}_k(x)\right) \\ &= \mathbf{1}_{[X_k=x]}\mathcal{I}\Pi_{\mathrm{SC},\kappa}^\mathcal{R}(\mathrm{b}_{R_k,\gamma})_\sharp\widehat{\eta}_k(X_k') + \mathbf{1}_{[X_k\neq x]}U_k(x) \\ &= (\widehat{\mathcal{U}}_k^\pi U_k)(x). \end{aligned}$$

Note that, since $\Pi_{\mathrm{SC},\kappa}^\mathcal{R}$ is a Hilbert projection onto an affine subspace, it is affine [Lax02]. Consequently, $\widehat{\mathcal{U}}^\pi$ is an unbiased estimator of the operator $\mathcal{U}^\pi$ in the following sense,

$$\begin{aligned} \mathbf{E_P}\left[\widehat{\mathcal{U}}^\pi(U, X_k, (R_k, X_k')) \mid X_k = x\right] &= \mathbf{E_P}\left[\mathcal{I}\Pi_{\mathrm{SC},\kappa}^\mathcal{R}(\mathrm{b}_{R_k,\gamma})_\sharp\mathcal{I}^{-1}U(X_k')\right] \\ &= \mathcal{I}\Pi_{\mathrm{SC},\kappa}^\mathcal{R}\mathbf{E}_{X_k' \sim P^\pi(\cdot|x)}\left[(\mathrm{b}_{r(x),\gamma})_\sharp\mathcal{I}^{-1}U(X_k')\right] \\ &= \mathcal{I}\Pi_{\mathrm{SC},\kappa}^\mathcal{R}\mathcal{T}^\pi\mathcal{I}^{-1}U(x) = (\mathcal{U}^\pi U)(x), \end{aligned}$$

where the first step invokes the linearity of $\Pi^{\mathcal{R}}_{\mathrm{SC},\kappa}$, the second step invokes the linearity of the isometry $\mathcal{I}$ established in Lemma 8 and the third step is due to the definition of $\mathcal{T}^\pi$. As a result, we see that the conditions (16) and (17) of Lemma 10 are satisfied by $\widehat{\mathcal{U}}^\pi$, the iterates $\{U_k\}_{k=1}^\infty$, and the step sizes $\{\alpha_k\}_{k=1}^\infty$. Moreover, for $w_k(x)$ defined by

$$w_k(x) = \left(\widehat{\mathcal{U}}^\pi(U_k, X_k, (R_k, X_k')) - (\mathcal{U}^\pi U_k)(X_k)\right)\mathbf{1}_{[X_k=x]}$$

we have $\|w_k(x)\|^2 \leq C_1 + C_2\|U(x)\|^2$ for universal constants $C_1, C_2$—this is shown in Lemma 11. As such, the condition of (18) is satisfied.

Finally, since $\mathcal{U}^\pi$ inherits contractivity from $\Pi^{\mathcal{R}}_{\mathrm{SC},\kappa}\mathcal{T}^\pi$ as shown in Lemma 5, we may invoke Lemma 10, which ensures that $U_k \to U$ with probability 1, where $U = \mathcal{U}^\pi U$ is a unique fixed point. $\qquad\square$

**Theorem 6.** *For a kernel $\kappa$ induced by a semimetric $\rho$ of strong negative type, the sequence $\{\widehat{\eta}_t\}_{t=1}^\infty$ given by* (11)-(13) *converges to $\eta^\pi_{\mathrm{SC},\kappa}$ with probability 1.*

*Proof.* By Proposition 2, the sequence $\{U_t\}_{t=1}^\infty$ with $U_t = \mathcal{I}\eta_t$ converges to a unique fixed point $U$ with probability 1. Note that

$$U^\star = \mathcal{U}^\pi U^\star$$
$$\mathcal{I}^{-1}U^\star = \mathcal{I}^{-1}\mathcal{U}^\pi U^\star$$
$$= \Pi^{\mathcal{R}}_{\mathrm{SC},\kappa}\mathcal{T}^\pi(\mathcal{I}^{-1}U^\star).$$

Therefore, $\mathcal{I}^{-1}U^\star$ is a fixed point of $\Pi^{\mathcal{R}}_{\mathrm{SC},\kappa}\mathcal{T}^\pi$. Since it was shown in Lemma 5 that $\Pi^{\mathcal{R}}_{\mathrm{SC},\kappa}\mathcal{T}^\pi$ has a unique fixed point, it follows that $\mathcal{I}^{-1}U^\star = \eta^\pi_{\mathrm{SC},\kappa}$. Since $\mathcal{I}$ is an isometry, $\widehat{\eta}_t = \mathcal{I}^{-1}U_t \to \mathcal{I}^{-1}U^\star$ with probability 1, so indeed $\widehat{\eta}_t \to \eta^\pi_{\mathrm{SC},\kappa}$ with probability 1. $\qquad\square$

To conclude, we prove Lemma 11, which was invoked in the proof of Proposition 2.

**Lemma 11.** *Under the conditions of Proposition 2, it holds that for any $x \in \mathcal{X}$ and $U \in \prod_{x\in\mathcal{X}}\mathbf{R}^{N(x)-1}$,*

$$\mathop{\mathbf{E}}_{X\sim P^\pi(\cdot|x)}\left[\left\|\mathcal{U}^\pi U(x) - \widehat{\mathcal{U}}^\pi(U, x, (R(x), X))(x)\right\|^2\right] \leq C_1 + C_2\|U(x)\|^2$$

*for finite constants $C_1, C_2 \in \mathbf{R}_+$.*

*Proof.* Since $\mathcal{I}$ is an isometry, we have that

$$\left\|\mathcal{U}^\pi U(x) - \widehat{\mathcal{U}}^\pi(U, x, (r, x'))\right\|^2 = \left\|\Pi^{\mathcal{R}}_{\mathrm{SC},\kappa}\mathcal{T}^\pi\mathcal{I}^{-1}U(x) - \Pi^{\mathcal{R}}_{\mathrm{SC},\kappa}\left((b_{r,\gamma})_\sharp\mathcal{I}^{-1}(x')\right)(x)\right\|^2_{\mathcal{H}},$$

where $\mathcal{H}$ is the RKHS induced by the kernel $\kappa$. Moreover, since $\Pi^{\mathcal{R}}_{\mathrm{SC},\kappa}$ is a nonexpansion in $\|\cdot\|_{\mathcal{H}}$ as argued in Lemma 5, we have that

$$\mathop{\mathbf{E}}_{X\sim P^\pi(\cdot|x)}\left[\left\|\mathcal{U}^\pi U(x) - \widehat{\mathcal{U}}^\pi(U, x, (R(x), X))(x)\right\|^2\right]$$
$$\leq \mathop{\mathbf{E}}_{X\sim P^\pi(\cdot|x)}\left[\left\|\mathcal{T}^\pi\mathcal{I}^{-1}U(x) - \left((b_{R,\gamma})_\sharp\mathcal{I}^{-1}U(X)\right)(x)\right\|^2_{\mathcal{H}}\right]$$
$$\leq 2\underbrace{\|\mathcal{T}^\pi\mathcal{I}^{-1}U(x)\|^2_{\mathcal{H}}}_{A} + 2\underbrace{\mathop{\mathbf{E}}_{X\sim P^\pi(\cdot|x)}\left[\left\|\left((b_{R,\gamma})_\sharp\mathcal{I}^{-1}U(X)\right)(x)\right\|^2_{\mathcal{H}}\right]}_{B}.$$

Proceeding, we will bound the terms $A, B$. To bound $A$, we simply have

$$A \leq \|\mathcal{T}^\pi\mathcal{I}^{-1}U(x) - \eta^\pi(x)\|^2_{\mathcal{H}} + \|\eta^\pi(x)\|^2_{\mathcal{H}}$$
$$\leq \gamma^{c/2}\|\mathcal{I}^{-1}U(x) - \eta^\pi(x)\|^2_{\mathcal{H}} + \|\eta^\pi(x)\|^2_{\mathcal{H}},$$

where we invoke the contraction of $\mathcal{U}^\pi$ in $\mathrm{MMD}_\kappa$ from Theorem 2. Note that $\eta^\pi(x) \in \mathcal{P}([0, (1 - \gamma)^{-1}R_{\max}]^d)$, so it follows that $\|\eta^\pi\|_{\mathcal{H}}^2 \leq D_{1,1}$ for some constant $D_{1,1}$ since the kernel $\kappa$ is bounded in compact domains. Expanding the norm of the difference above yields

$$A \leq (1 + \gamma^{c/2})D_{1,1} + D_{1,2}\|\mathcal{I}^{-1}U(x)\|_{\mathcal{H}}^2 = (1 + \gamma^{c/2})D_{1,1} + D_{1,2}\|U(x)\|^2$$

for a finite constant $D_{1,2}$, again invoking the isometry $\mathcal{I}$ in the last step.

Our bound for $B$ is similar. Choose any $x' \in \operatorname{supp} P^\pi(\cdot \mid x)$. We consider the operator $\widetilde{\mathcal{T}}_{x'} : \mathcal{P}([0, (1-\gamma)^{-1}R_{\max}]^d) \to \mathcal{P}([0, (1-\gamma)^{-1}R_{\max}]^d)$ given by

$$(\widetilde{\mathcal{T}}_{x'}\eta)(x) = (\mathrm{b}_{R(x),\gamma})_\sharp \eta(x').$$

This operator is a contraction in $\mathrm{MMD}_\kappa$, and correspondingly has a fixed point $\eta_{x'}^\pi$. To see this, we note that $\widetilde{T}_{x'}$ is simply a special case of $\mathcal{U}^\pi$ for the case $P^\pi(\cdot \mid x) = \delta_{x'}$, so the contractivity and existence of the fixed point are inherited from Theorem 2. Proceeding in a manner similar to the bound on $A$, we have

$$
\begin{aligned}
\left\|\left((\mathrm{b}_{R,\gamma})_\sharp \mathcal{I}^{-1}U(x')\right)(x)\right\|_{\mathcal{H}}^2 &= \left\|\widetilde{\mathcal{T}}_{x'}\mathcal{I}^{-1}U(x)\right\|_{\mathcal{H}}^2 \\
&\leq \left\|\widetilde{\mathcal{T}}_{x'}\mathcal{I}^{-1}U(x) - \eta_{x'}\right\|_{\mathcal{H}}^2 + \|\eta_{x'}(x)\|_{\mathcal{H}}^2 \\
&\leq \gamma^{c/2}\|\mathcal{I}^{-1}U(x) - \eta_{x'}\|_{\mathcal{H}}^2 + \|\eta_{x'}(x)\|_{\mathcal{H}}^2 \\
&\leq (1 + \gamma^{c/2})D_{2,1} + D_{2,2}\|U(x)\|^2
\end{aligned}
$$

where the final step mirrors the bound on $A$. Therefore, we have shown that

$$
\begin{aligned}
\mathop{\mathbf{E}}_{X \sim P^\pi(\cdot|x)} &\left[\left\|\mathcal{U}^\pi U(x) - \widehat{\mathcal{U}}^\pi(U, x, (R(x), X))(x)\right\|^2\right] \\
&\leq 2(1 + \gamma^{c/2})(D_{1,1} + D_{2,1}) + (D_{1,2} + D_{2,2})\|U(x)\|^2,
\end{aligned}
$$

completing the proof. $\qquad\square$

## C  Memory Efficiency of Randomized EWP Dynamic Programming

In Section 4, we argued for the necessity of considering a projection operator in EWP dynamic programming. While we provided a randomized projection, Theorem 3 requires that we apply only a finite amount of DP iterations. Thus, one might ask if, given that we apply only finitely many iterations, the naive unprojected EWP dynamic programming can produce accurate enough approximations of $\eta^\pi$ without costing too much in memory.

In this section, we demonstrate that, in fact, the algorithm described in Theorem 3 can approximate $\eta^\pi$ to any desired accuracy with many fewer particles. Suppose our goal is to derive some $\eta$ such that

$$\overline{\mathrm{MMD}_\kappa}(\eta, \eta^\pi) \leq \epsilon$$

for some $\epsilon > 0$. We will derive bounds on the number of required particles to attain such an approximation with unprojected EWP dynamic programming (denoting the number of particles $m_{\mathsf{unproj}}$) as well as with our algorithm described in Theorem 3 (denoting the number of particles $m_{\mathsf{proj}}$. In both cases, we will compute iterates starting with some $\eta_0 \in \mathscr{C}_{\mathrm{EWP},m}$ with $\overline{\mathrm{MMD}_\kappa}(\eta_0, \eta^\pi) \leq D < \infty$. For simplicity, we will consider the energy distance kernel with $\alpha = 1$.

The remainder of this section will show that the dependence of the number of atoms on both $\epsilon$ and $|\mathcal{X}|$ is substantially worse in the unprojected case (that is, $m_{\mathsf{proj}} \ll m_{\mathsf{unproj}}$ for large state spaces or low error tolerance). We demonstrate this with concrete lower bounds on $m_{\mathsf{unproj}}$ and upper bounds on $m_{\mathsf{proj}}$ below; note that these bounds are not optimized for tightness or generality, and are instead aimed to provide straightforward evidence of our core points above.

We will begin by bounding $m_{\mathsf{unproj}}$. In the best case, $\eta_0(x)$ is supported on 1 particle for each $x$. If any state can be reached from any other state in the MDP with non-zero probability, then applying the distributional Bellman operator to $\eta_0$ will result in $\eta_1(x)$ having support on $|\mathcal{X}|$ atoms at each

state $x$ (due to the mixture over successor states in the Bellman backup). Consequently, the iterate $\eta_k(x)$ will be supported on $|\mathcal{X}|^k$ atoms. Since $\overline{\mathrm{MMD}}_\kappa(\eta_k, \eta^\pi) \leq \gamma^{1/2}D$ by Theorem 2, we require

$$K \geq \frac{2\log(D/\epsilon)}{\log\gamma^{-1}}$$

to ensure that $\overline{\mathrm{MMD}}_\kappa(\eta_K, \eta^\pi) \leq \epsilon$. Thus, we have

$$m_{\mathsf{unproj}} \geq |\mathcal{X}|^{\frac{2\log(D/\epsilon)}{\log\gamma^{-1}}}.$$

On the other hand, the following lemma bounds $m_{\mathsf{proj}}$; we prove the lemma at the end of this section.

**Lemma 12.** *Let $\eta_{m_{\mathsf{proj}}}$ denote the output of the projected EWP algorithm described by Theorem 3 with $m = m_{\mathsf{proj}}$ particles. Then under the assumptions of Theorem 3 and with the energy distance kernel with $\alpha = 1$, $\overline{\mathrm{MMD}}_\kappa(\eta_{m_{\mathsf{proj}}}, \eta^\pi) \leq \epsilon$ is achievable with*

$$m_{\mathsf{proj}} \in \Theta\left(\epsilon^{-2}\frac{dR_{\max}^2}{(1-\sqrt{\gamma})^2(1-\gamma)^2}\mathsf{polylog}\left(\frac{1}{\epsilon}, \frac{1}{\delta}, |\mathcal{X}|, d, R_{\max}, \frac{1}{1-\sqrt{\gamma}}\right)\right). \tag{21}$$

For any fixed MDP with $|\mathcal{X}| \geq 4$ and $\gamma \geq 1/2$, we have that

$$m_{\mathsf{unproj}} \geq \exp\left(2\log|\mathcal{X}|\frac{\log\epsilon^{-1}}{\log\gamma^{-1}}\right)\exp\left(2\log|\mathcal{X}|\frac{\log D}{\log\gamma^{-1}}\right)$$

$$= \exp\left(2\log|\mathcal{X}|\frac{\log D}{\log\gamma^{-1}}\right)\epsilon^{-2\frac{\log|\mathcal{X}|}{\log\gamma^{-1}}}$$

$$\in \Omega(\epsilon^{-4})$$

since $D > 0$ and does not depend on $\epsilon$. Meanwhile, we have $m_{\mathsf{proj}} \in \Theta(\epsilon^{-2}\mathsf{polylog}(\epsilon^{-1}))$ by Lemma 12, indicating a much more graceful dependence on $\epsilon$ relative to the unprojected algorithm.

On the other hand, for any fixed tolerance $\epsilon \leq \gamma D$, we immediately have

$$m_{\mathsf{unproj}} \in \Omega(|\mathcal{X}|^2)$$
$$m_{\mathsf{proj}} \in \Theta(d \cdot \mathsf{polylog}(d, |\mathcal{X}|)).$$

In the worst case, we may have $d \in \Theta(|\mathcal{X}|)$ (any larger $d$ will induce linearly dependent cumulants). Thus, we have

$$\frac{m_{\mathsf{proj}}}{m_{\mathsf{unproj}}} \in \begin{cases} \widetilde{O}(|\mathcal{X}|^{-1}) & d \in \omega(1) \\ \widetilde{O}(|\mathcal{X}|^{-2}) & d \in \Theta(1), \end{cases}$$

so the projected algorithm scales much more gracefully with $|\mathcal{X}|$ as well.

**Proof of Lemma 12**

Finally, we prove Lemma 12, which determines the number of atoms required to achieve an $\epsilon$-accurate return distribution estimate with the algorithm of Theorem 3.

**Lemma 12.** *Let $\eta_{m_{\mathsf{proj}}}$ denote the output of the projected EWP algorithm described by Theorem 3 with $m = m_{\mathsf{proj}}$ particles. Then under the assumptions of Theorem 3 and with the energy distance kernel with $\alpha = 1$, $\overline{\mathrm{MMD}}_\kappa(\eta_{m_{\mathsf{proj}}}, \eta^\pi) \leq \epsilon$ is achievable with*

$$m_{\mathsf{proj}} \in \Theta\left(\epsilon^{-2}\frac{dR_{\max}^2}{(1-\sqrt{\gamma})^2(1-\gamma)^2}\mathsf{polylog}\left(\frac{1}{\epsilon}, \frac{1}{\delta}, |\mathcal{X}|, d, R_{\max}, \frac{1}{1-\sqrt{\gamma}}\right)\right). \tag{21}$$

*Proof.* Note that, by Theorem 3, increasing $m_{\mathsf{proj}}$ can only decrease the error $\epsilon$ as long as $m_{\mathsf{proj}} \geq 1$. Therefore, as shown in (14) in the proof of Theorem 3, there exists a universal constant $C_0 > 0$ such that

$$\epsilon := C_0 \frac{1}{\sqrt{m_{\mathsf{proj}}}} \underbrace{\frac{d^{\alpha/2}R_{\max}^\alpha}{(1-\gamma^{\alpha/2})(1-\gamma)^\alpha}}_{c_1}\left(\underbrace{\log\left(\frac{|\mathcal{X}|\delta^{-1}}{\log\gamma^{-\alpha}}\right)}_{c_2} + \log m_{\mathsf{proj}}\right). \tag{22}$$

Now, we write $c_3 = C_0 c_1 c_2$, $c_4 = C_0 c_1$, and $u := \sqrt{m_{\mathsf{proj}}}$, yielding

$$\epsilon = \frac{c_3}{u} + c_4 \frac{\log u^2}{u}$$
$$= \frac{c_3}{u} + 2c_4 \frac{\log u}{u}.$$

Then, after isolating the logarithmic term and exponentiating, we see that

$$u = \exp\left(\frac{u\epsilon - c_3}{2c_4}\right).$$

We now rearrange this expression and invoke the identity $W(z)e^{W(z)} = z$ where $W$ is a Lambert $W$-function [CGH$^+$96]:

$$u e^{c_3/2c_4} \exp\left(-\frac{u\epsilon}{2c_4}\right) = 1$$

$$-\frac{u\epsilon}{2c_4} \exp\left(-\frac{u\epsilon}{2c_4}\right) = -\frac{e^{-c_3/2c_4}\epsilon}{2c_4} = -\frac{e^{-c_2/2}\epsilon}{2c_4}$$

$$\therefore z e^z = -\frac{e^{-c_2/2}\epsilon}{2c_4} \qquad\qquad z := -\frac{u\epsilon}{2c_4}.$$

There are two branches of the Lambert $W$-function on the reals, namely $W_0$ and $W_{-1}$. These two branches satisfy $W_0(ze^z) = z$ when $z \geq -1$ and $W_{-1}(ze^z) = z$ when $z \leq -1$. In our case, we know that $z$ is negative, and it is known [CGH$^+$96] that $|W_0(z)| \leq 1$ when $z \in [-1, 0]$. Consequently, when $z \geq -1$, we have $|\frac{u\epsilon}{2c_4}| \leq 1$, and substituting $m_{\mathsf{proj}} = u^2$, we have

$$m_{\mathsf{proj}} \leq \frac{4c_4^2}{\epsilon^2} \text{ when } z \geq -1. \tag{23}$$

On the other hand, when $z \leq -1$, we have

$$z = W_{-1}\left(-\frac{e^{-c_2/2}\epsilon}{2c_4}\right)$$

$$\therefore -\frac{u\epsilon}{2c_4} = W_{-1}\left(-\frac{e^{-c_2/2}\epsilon}{2c_4}\right)$$

$$\therefore m_{\mathsf{proj}} = \frac{4c_4^2}{\epsilon^2} W_{-1}^2\left(-\frac{e^{-c_2/2}\epsilon}{2c_4}\right), \quad z \leq -1.$$

Since it is known [CGH$^+$96, Equations 4.19, 4.20] that $W_{-1}^2(-\overline{z}) \in \mathsf{polylog}(1/\overline{z})$, incorporating (23), we have that

$$m_{\mathsf{proj}} \leq \begin{cases} \frac{4c_4^2}{\epsilon^2} & z \geq -1 \\ \frac{4c_4^2}{\epsilon^2} W_{-1}^2\left(-\frac{e^{c_2/2}\epsilon}{2c_4}\right) & z \leq -1 \end{cases}$$

$$\leq \frac{4c_4^2}{\epsilon^2} \max\left(1, \mathsf{polylog}(c_4 e^{c_2/2}\epsilon^{-1})\right)$$

$$\leq \frac{4C_0^2 d R_{\max}^2}{(1 - \sqrt{\gamma})^2 (1 - \gamma)^2 \epsilon^2} \mathsf{polylog}\left(\frac{1}{\epsilon}, \frac{1}{\delta}, |\mathcal{X}|, d, R_{\max}, \frac{1}{1 - \sqrt{\gamma}}\right).$$

The upper bound given above will generally not be an integer. However, increasing $m_{\mathsf{proj}}$ can only improve the approximation error, as shown in Theorem 3 since $\log m/\sqrt{m}$ decreases monotonically when $m > 7$. So, we can round $m_{\mathsf{proj}}$ up to the nearest integer (or round it down when $m \leq 7$) incurring a penalty of at most one atom. It follows that the randomized EWP dynamic programming algorithm of Theorem 3 run with $m_{\mathsf{proj}}$ given by (21) produces a return distribution function $\eta_{m_{\mathsf{proj}}}$ for which $\overline{\mathrm{MMD}}_\kappa(\eta_{m_{\mathsf{proj}}}, \eta^\pi) \leq \epsilon$. $\qquad\square$

Table 1: Certificate that $\Pi^{\mathcal{R}}_{C,\kappa}$ is not affine

| Support point $\xi \in \mathcal{R}$ | $q_1(\xi)$ | $q_2(\xi)$ |
|:---:|:---:|:---:|
| $(0,0)$ | 0 | 0 |
| $(0,1)$ | 0 | 0 |
| $(0,2)$ | 0 | 0 |
| $(0,3)$ | 0 | 0 |
| $(1,0)$ | 0 | 0 |
| $(1,1)$ | 0.1999 | 0.2057 |
| $(1,2)$ | 0.1999 | 0.1959 |
| $(1,3)$ | 0 | 0 |
| $\mathbf{(2,0)}$ | **0.0937** | **0.07957** |
| $\mathbf{(2,1)}$ | **0.2062** | **0.2413** |
| $(2,2)$ | 0.1999 | 0.2026 |
| $(2,3)$ | 0 | 0 |
| $\mathbf{(3,0)}$ | **0.0937** | **0.0787** |
| $(3,1)$ | 0.0063 | 0 |
| $(3,2)$ | 0 | 0 |
| $(3,3)$ | 0 | 0 |

# D    Nonlinearity of the Categorical MMD Projection

In Section 6, we noted that the categorical projection $\Pi^{\mathcal{R}}_{C,\kappa}$ is non-affine. Here, we provide an explicit example certifying this phenomenon.

We consider a single-state MDP, since the nonlinearity issue is independent of the cardinality of the state space (the projection is applied to each state-conditioned distribution independently). We write $\mathcal{R} = \{0,\ldots,3\}^2$, and consider the kernel $\kappa$ induced by $\rho(x,y) = \|x-y\|_2$—this resulting MMD is known as the energy distance, which is what we used in our experiments. We consider two distributions, $p_1 = \delta_{[1.5,1.5]}$ and $p_2 = \delta_{[2.5,0]}$.

We consider $\lambda = 0.8$ and compare $q_1 = \Pi^{\mathcal{R}}_{C,\kappa}(\lambda p_1 + (1-\lambda)p_2)$ with $q_2 = \lambda \Pi^{\mathcal{R}}_{C,\kappa} p_1 + (1-\lambda)\Pi^{\mathcal{R}}_{C,\kappa} p_2$, and we note that $q_1 \neq q_2$; confirming that $\Pi^{\mathcal{R}}_{C,\kappa}$ is not an affine map. The results are tabulated in Table 1, with bolded entries depicting the atoms with non-negligible differences in probability under $q_1, q_2$.

# E    Experiment Details

TD-learning experiments were conducted on a NVidia A100 80G GPU to parallelize experiments. Methods were implemented in Jax [BFH+18], particularly with the help of JaxOpt [BBC+21] for vectorizing QP solutions — this was helpful for computing the categorical projections discussed in this work. SGD was used for optimization, using an annealed learning rate schedule $(\lambda_k)_{k\geq 0}$ with $\lambda_k = k^{-3/5}$, satisfying the conditions of Lemma 10. Experiments with constant learning rates yielded similar results, but were less stable—this validates that the choice of learning rate schedule did not impede learning.

The dynamic programming experiments were implemented in the Julia programming language [BEKS17].

In all experiments, we used the kernel induced by $\rho(x,y) = \|x-y\|_2$ with reference point 0 for MMD optimization—this corresponds to the energy distance, and satisfies the requisite assumptions for convergent multivariate distributional dynamic programming outlined in Theorem 2.

# F    Neural Multivariate Distributional TD-Learning

For the sake of illustration, in this section, we demonstrate that the signed categorical TD learning algorithm presented in Section 6 can be scaled to continuous state spaces with neural networks. We will consider an environment with visual (pixel) observations of a car in a parking lot, an example observation is shown in Figure 4.

Here, we consider 2-dimensional cumulants, where the first dimension tracks the $x$ coordinate of the car, and the second dimension is an indicator that is $1$ if and only if the car is parked in the parking spot. We learn a multivariate return distribution function with transitions sampled from trajectories that navigate around the obstacle to the parking spot. Notably, the successor features (expectation of multivariate return distribution) will be zero in the first dimension, since the set of trajectories is horizontally symmetric. Thus, from the successor features alone, one cannot distinguish the observed policy from one that traverses straight through the obstacle!

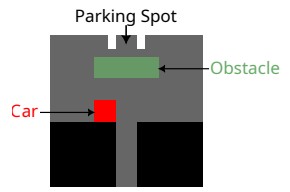

Figure 4: Example state in the parking environment.

Fortunately, when modeling a distribution over multivariate returns, we should see that the support of the multivariate return distribution does not include points with vanishing first dimension.

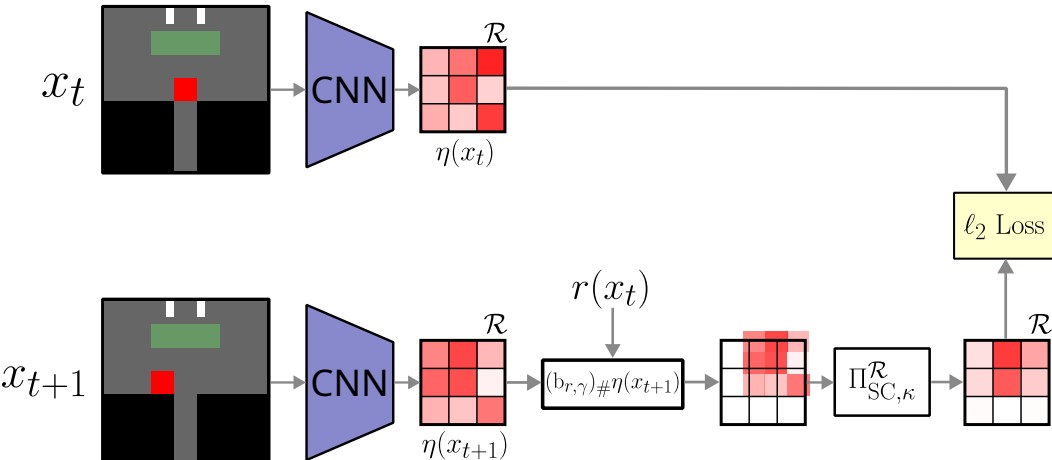

Figure 5: Neural architecture for modeling multi-return distributions from images.

To learn the multivariate return distribution function from images, we use a convolutional neural architecture as shown in Figure 5.

Notably, we simply use convolutional networks to model the signed masses for the fixed atoms of the categorical representation. The projection $\Pi^{\mathcal{R}}_{\mathrm{SC},\kappa}$ is computed by a QP solver as discussed in Section 5, and is applied only to the target distributions (thus we do not backpropagate through it).

We compared the multi-return distributions learned by our signed categorical TD method with that of [ZCZ+21]. Our results are shown in Figure 6. We see that both TD-learning methods accurately estimate the distribution over multivariate returns, indicating that no multivariate return will have a vanishing lateral component. Quantitatively, we see that the EWP algorithm appears to be stuck in a local optimum, with some particles lying in regions of low probability mass.

Moreover, on the right side of Figure 6, we show predicted return distributions for two randomly sampled reward vectors, and quantitatively evaluate the two methods. The leftmost reward vector incentivizes the agent to take paths conservatively avoiding the obstacle on the left. The rightmost reward vector incentivizes the agent to get to the parking spot as quickly as possible. We see that the EWP TD learning algorithm of [ZCZ+21] more accurately estimates the return distribution function corresponding to the latter reward vector, while our signed categorical TD algorithm more accurately estimates the return distribution function corresponding to the former reward vector. In both cases, both methods produce accurate estimations.

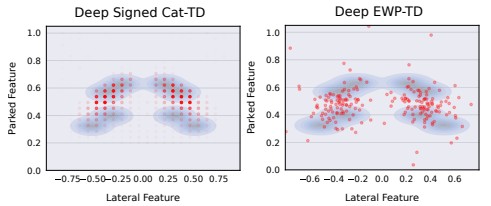 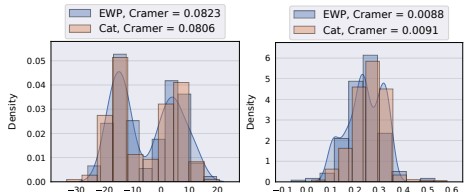

Figure 6: Multi-return distributions learned by signed categorical TD and EWP TD, as well as examples of predicted return distributions on two randomly sampled reward functions.

