# OpenReview forum: "Foundations of Multivariate Distributional Reinforcement Learning"
_NeurIPS.cc/2024/Conference — NeurIPS 2024 poster_

### Official Review · Reviewer_9dnf · 2024-07-05

**Soundness:** 2
**Presentation:** 2
**Contribution:** 1
**Rating:** 3
**Confidence:** 3

**Summary:**

This paper studies the theoretical foundations of multivariate distributional RL, particularly providing the convergence proof under the MMD distance. The paper first investigates the aspect of particle-based multivariate dynamic programming in Section 4 and then shifts the attention to categorical representation-based multivariate dynamic programming in Section 5 and Temporal difference learning in Section 5. Experiments are conducted mainly on the distributional successor measure setting.

**Strengths:**

* The convergence proof of extending existing univariate DP or TD to its multivariate settings is technically sound, although the corresponding conclusions are straightforward.

* The writing, including the proof, is rigorous, and relevant works are sufficiently discussed.

**Weaknesses:**

* **Straightforward motivation and extension**. Personally, my main concern is that the contribution of this paper is within a limited scope. Multivariate distributional RL was first studied by [ZCZ+21], where the corresponding Bellman operator with the convergence proof under the Wasserstein distance is already provided. Having this knowledge, this paper mainly investigates the convergence under the MMD distance with either particle or categorical representation. This seems incremental to the theoretical values, given that the univariate version under MMD and categorical representation (with Cramer distance) have also been studied earlier. Therefore, a similar conclusion, which applies to the multivariate version and is highly based on existing results and proof techniques, is straightforward and easy to expect, in my opinion. Additionally, I am not fully convinced by the motivation of multi-dimensional distributional RL and whether it should be valued sufficiently in practice. The authors are suggested to emphasize the practical motivations, for example, by providing concrete examples or providing more experiments in real applications. Without these, I am not sure whether investigating the foundations of this setting is really useful (in an incremental way in this paper).


* **Less concentrated organization**. The paper first studies the particle-based MMD distributional RL, which is a straightforward extension of [NGV20]. However, the authors suddenly shift their attention to the categorical representation albeit still equipped with MMD. I am not sure the motivation behind this kind of paper organization, but it indeed made me feel that some components are unnaturally combined together, where each component seems to rely highly on the corresponding existing works, e.g., univariate MMD or categorical distributional RL. This makes it difficult to posit the contribution of this paper.


* **Limited and less general experiments**. While I understand some of this paper's motivations come from the distributional successor representation, it would be advisable to concentrate on more acceptable experiments in distributional RL, like Atari games in [ZCZ+21] or Mujoco environments. I acknowledge that this paper is theory-oriented, but it is more suggested to provide some general experimental results like [ZCZ+21] since this paper is highly based on [ZCZ+21].

**Questions:**

I personally believe [ZCZ+21] indeed provides many convergence guarantees, which is inconsistent with the statement in Line 30 on Page 1. Thus, I think this kind of statement may not be accurate or proper.

I personally do not think the projection operator is generally necessary. Imagine both the current Bellman statistics and the TD target ones are within the real set or unbounded, e.g., the particle representation. Another concern is why we need to introduce the randomized projector, which seems not common in the existing literature.

I partially disagree with the main contribution statement of the paper in lines 33 and 67: it designs (1) a computationally tractable and (2) a theoretically justified algorithm with convergence guarantee. Firstly, the theoretical part is highly reliance on existing conclusions, which may be overclaimed by this paper. Also, the authors did not extensively verify the computational efficiency by conducting large-scale experiments.

How the size of $\eta_k$ increases exponentially with k in line 62 of Page 2.

Some writings are not clear. For example, what are the two ideas mentioned in Line 24 on Page 1?

Is the word cumulants referring to statistical cumulants or general statistics for a distribution? If it is the latter, a more careful choice of this word is suggested.

The equally weighted particle seems inaccurate. In non-parametric statistics, we draw some samples to characterize the empirical distribution, and some samples are assigned a signal to contribute to the mass. I am not sure why the authors emphasize the equal weight of each sample, which seems unrelated to the following analysis.

More explanation about the QR programming should be given on Page 5. The current version of the Algorithm makes it difficult for me to understand the details.

In summary, this paper gives me the feeling that 1) the theoretical contribution is within a limited scope as most conclusions and extensions seem incremental, 2) the motivation may not be practical, and it lacks sufficient experiments to demonstrate the corresponding statements.

**Limitations:**

yes, the authors mentioned some limitations of their work in Section 7

---

> ### Author Rebuttal · Authors · 2024-08-06
>
> We thank the reviewer for the detailed assessment of our work.
>
> We appreciate your comments about the scalability of our approach and the motivations for studying foundations of multivariate DRL; we have discussed these in detail in the general response.
>
> **Q1**: The reviewer claims that our work is incremental given the results of [ZCZ+21]. We respectfully disagree, and we hope the following clarify our novelty:
>
> 1. [ZCZ+21] only analyzes the contractivity of the *unprojected* distributional Bellman operator. This result alone does not prove convergence of any practical algorithm. Our work is the first to provide analysis for the type of distributional operator that we can deploy in practice, and associated TD-learning algorithms, unlike [ZCZ+21].
> 2. [ZCZ+21] only provides convergence analysis for dynamic programming (again, with the unprojected / intractable operator), which we also generally cannot perform in practice without knowledge of the MDP. Our work goes beyond this and provides convergence guarantees for *TD-learning*, which can be computed from samples of the MDP (and no knowledge of the transition kernel or reward function). This is a much more difficult result to achieve, and we actually require new proof techniques and models to accomplish this (e.g., we must represent distributions as signed measures). Our novel analysis inspired a new algorithm which has not yet been studied even in the $d=1$ case, and is the first provably convergent TD-learning algorithm in the $d>1$ setting.
> 3. While we have done much more than extend the analysis to MMD, this itself is a bonus. There are fundamental difficulties to performing TD-learning in Wasserstein space due to biased gradients. This is another facet under which our analysis applies to a practically-relevant algorithm, unlike [ZCZ+21].
>
> To be clear, [ZCZ+21] is fantastic and was certainly an inspiration, but our results are not simply incremental modifications of theirs.
>
> **Q2**: Projections are necessary to avoid exponential blowup in the number of particles in return distributions (see Q4) in DP methods.
> Some TD algorithms do not explicitly compute projections, just passing gradients through a fixed representation, such as equally weighted particles.
> However, to *prove convergence* of such algorithms, one must show that these updates track applications of a projected operator (see e.g. [RMA+23; Sec 5-6]). Crucially, convergence of TD algorithms does not follow from contractivity of the unprojected Bellman operator, and analysis of projected operators is essential. Thus, while [ZCZ+21] does not explicitly compute a projection, this is one sense in which their paper does not theoretically justify their algorithm.
>
> Randomized projections uncommon in the literature: this is a novel contribution that we introduced to tackle issues with computing the EWP projection for DP. In the case of $d>1$, the MMD projection onto the space of EWP representations is non-convex, and could be extremely expensive (if at all possible) to compute in practice. The randomized projection that we introduce allows us to design a tractable algorithm for approximating the multi-return distribution function with EWP representations, which enjoys dimension-free theoretical convergence bounds.
>
> **Q3**: As described above, our theoretical results are *not* simple corollaries of existing results. They required novel proof techniques and algorithms (Sec 4-6). Each step of the algorithms described is objectively computationally tractable. With our statements on line 33/67, we are comparing against existing analyses of multivariate distributional RL which rely on regression oracles (for which no tractable algorithm generally exists). See also our general response, which demonstrates that our proposed TD-learning algorithm can be straightforwardly scaled with neural networks.
>
> **Q4**: Imagine an MDP with $n$ states and a nonzero probability of transition between any 2 states, and $\eta_0(x)$ is supported on 1 point for each $x$. Then, after one iteration of DP, $\eta_1(x)$ will be supported on $n$ points (contributions from each of the $n$ successor states which have return distributions supported on $1$ point).  Now, $\eta_2(x)$ will be supported on $n^2$ points, since it mixes contributions of $n$ return distributions each having support on $n$ points. Continuing, we see that $\eta_k(x)$ is generally supported on $n^k$ points.
>
> **Q5**: We will make this more clear in our revision; the two ideas are modeling return distributions and modeling multivariate returns.
>
> **Q6**: Good question. Cumulant here refers to the multivariate return. While this is standard terminology in this niche of RL research (see e.g. *Bootstrapped Representations for Reinforcement Learning*, Le Lan et al. 2023), it is worth clarifying as you point out.
>
> **Q7**: Alternatively, one could model the positions of $m$ particles and their probability masses, as opposed to modeling empirical distributions. This requires more memory and is often more difficult to optimize (see [BDR23]). Can you clarify in what sense the EWP representation is "inaccurate"? While modeling atom probabilities provides more flexibility, EWP representations are valid probability measures, and we provide strong bounds on the quality of our EWP approximation to the true (nonparametric) multivariate return distributions (Theorem 3).
>
> **Q8**: We would be happy to explain this further, but it would help us if you could specify more precisely which part is difficult to understand. The proof of Lemma 1 explicitly constructs the QP to be solved for computing the projection. This QP is described completely in the pseudocode on the line with QPSolve -- this is a minimization of a quadratic over a convex set. We can use efficient QP solvers to solve this. It also does not preclude function approximation, since we only need to solve the QP for the target returns (which gradients are not propagated through).

---

> ### Author Response · Authors · 2024-08-11
>
> Thank you for your detailed review of our submission. We appreciate the time you've taken to evaluate our work.
> In our rebuttal, we addressed your concerns regarding the novelty of our theoretical results and provided clarification on how our work differs from existing literature. We also included results from larger-scale experiments as per your suggestion.
> If you've had a chance to review our rebuttal, we'd be interested to know if our explanations and additional results have helped address your concerns; and if so, we would be grateful if you could consider increasing your score. If you have any further questions, we are eager to discuss them.

---

> > ### Comment · Area_Chair_Ecuu · 2024-08-11
> >
> > Dear Reviewer 9dnf, the authors have written a comprehensive response regarding the details of their contribution. Do their comments change your assessment of what the field will learn from this paper? It would be great if you can comment on this and raise additional or remaining concerns while we still can ask the authors for clarification.

---

> > > ### Comment · Reviewer_9dnf · 2024-08-11
> > >
> > > Thanks for the authors' response. Some concerns in Questions have been clarified; however, other major drawbacks in the Weakness part have not been addressed, and I have not changed the original impression this paper left on me. In general, positing the contribution to the foundations of multivariate distributional RL is very incremental in contribution based on the limited scope of existing literature, and lacking large-scale experiments is also one of the main limitations of the paper's current version. Thus, I keep my score and suggest the authors improve the manuscript substantially.

---

> > > > ### Author Response · Authors · 2024-08-12
> > > >
> > > > **Questions section**. The reviewer mentioned that some of their concerns in the Questions section have been clarified. We are glad to hear this, though we believe our response addresses *all* the questions raised. Please let us know which specific concerns remain and we'll be happy to provide further explanation.
> > > >
> > > > **Weaknesses section**. Again, we would appreciate if the reviewer can clarify which points we can address further. To summarize,
> > > > 1. 'Simple motivation and extension': Our response highlights that our algorithms and theoretical analysis go far beyond what is present in the literature, and that existing literature does not supply theoretical analysis for any practical algorithms for multi-dimensional distributional RL.
> > > > 2. 'Less concentrated organization': This is something that will be relatively simple to fix (and we will be happy to do so) in our revision;
> > > > 3. 'Limited and less general experiments': The purpose of the paper was to study the theory of multivariate DRL. But, we did demonstrate that our algorithms are amenable to larger scale applications in the general response.

---

### Official Review · Reviewer_wmwv · 2024-07-09

**Soundness:** 3
**Presentation:** 4
**Contribution:** 3
**Rating:** 6
**Confidence:** 4

**Summary:**

The authors propose a tractable and convergence-guaranteed method, called randomized dynamic programming, for multivariate distributional reinforcement learning (distRL). They also introduce practical algorithms, multivariate EWP-TD and signed-categorical-TD, provide an upper bound on MMD with respect to dimension $d$, and empirically compare performance using the Cramer distance in a tabular MDP environment.

**Strengths:**

- Despite the extensive background knowledge and theoretical understanding required in this field, the authors have proficiently written the paper to be accessible to first-time readers, using standard and clear notations.

- The visualizations for the proposed EWP-TD and Categorical-TD are particularly excellent and intuitive.

- The authors clearly explain the issues and limitations arising from finite parameterization in distRL theory and naturally introduce the novel concept of randomized DP for EWP representation.

- The theory regarding categorical representation in multivariate cases is novel and informative, and the convergence rate matches the results in univariate cases.

**Weaknesses:**

While the contributions of the authors are clear, it is uncertain whether the proposed theory aligns with the contributions. The proposal of a randomized DP for a tractable EWP is interesting, but its convergence and the existence of a unique fixed point do not seem to be clearly described. The detailed theory appears to focus more on the categorical representation. See the Questions section for further details.

**Questions:**

-	In Theorem 3, the symbol seems to be $\leq$ instead of $\in$.

-	I understand that the randomized DP is a sampling-based algorithm that circumvents the EWP representation in terms of MMD. However, the projection that minimizes MMD does not guarantee the uniqueness of the fixed point. How can its relaxation, the randomized DP, have a unique fixed point? Theorem 3 seems to imply a bound at $K=O(\log m)$, suggesting that the number of atoms increases exponentially. Can Theorem 3 be analyzed with $K \rightarrow \infty$ for a given number of atoms $m$?

-	The depiction of Cramer distance with respect to the number of atoms for each algorithm in Figures 2 and 3 is very interesting. While EWP-TD is hardly affected by the increase in the number of atoms, signed-Cat-TD shows a consistent decrease. When the dimension increases, will EWP-TD always have a smaller Cramer distance compared to signed-Cat-TD?


-	In Line 337, can you elaborate on "using randomized support points for the categorical algorithm"? It is unclear whether this refers to EWP-TD or signed-Cat-TD.

Since there are still some aspects I don't fully understand, I'm willing to raise the score if the authors can clearly address my questions.

**Limitations:**

The authors have clearly stated the assumptions and limitations of their theory.

---

> ### Author Rebuttal · Authors · 2024-08-06
>
> We thank the reviewer for their thorough assessment and for their interest in our work. We hope the responses below address the queries raised in the review, please let us know if you have any further questions.
>
> **Q1**: With regard to $\leq$ vs $\in$ in Theorem 3 -- this was a stylistic choice. We were invoking the definition of $\widetilde{O}(\cdot)$ as a set-valued function -- all elements of this set satisfy the upper bound that you are describing. This is the convention used, for example, in the classic CLRS *Introduction to Algorithms* text; its meaning coincides with what you are describing.
>
> **Q2**: You are correct that there is not a unique fixed point of the projected operator. The theorem is simply claiming that, after enough ($K$) iterations, we can guarantee that the resulting $\eta_K$ will be within a very small margin of error from $\eta^\pi$ w.r.t. the MMD. Notably, $\eta^\pi$ is the unique fixed-point of the (unprojected) distributional Bellman operator, which is well-defined (by Theorem 2). Convergence results for dynamic programming algorithms without unique fixed points also exist in previous works, such as [WUS23]. Note we do not require exponentially-many atoms; the free parameter here is $m$, not $K$. Therefore, for a desired tolerance level, we require polylog-many iterations ($K$) of dynamic programming.
>
> **Q3**: This is a fantastic question. The point to note is that the convergence bound for EWP is *dimension-free*, while those of the categorical algorithms are not (they depend on $d$). However, in the case of EWP-TD, the algorithm is prone to convergence to local minima (see our discussions on lines 146 and 324). Thus, we suspect that the EWP-TD examples are likely stuck in local minima, but notably that the quality of these local minima is roughly uniformly high for enough particles (e.g., as many as we use in our experiments). As you suggest, we would expect that EWP-TD should perform favorably to the Cat-TD algorithms with high $d$ -- however this is not guaranteed. Particularly, there is no known convergence guarantee for the EWP-TD algorithm when $d > 1$. Moreover, we could potentially leverage prior knowledge of the supports of the return distributions to enhance the resolution of the Cat-TD representations in relevant areas of the space of multi-returns, which could result in improved performance.
>
> **Q4**: This was in reference to the Signed-Cat-TD algorithm. In previous experiments, the Cat-TD algorithms were representing categorical distributions on evenly-spaced points in the space of multi-returns, which necessarily requires an exponential number of support points as a function of $d$. On line 337, we meant to say that we instead fixed a number $m$ of support points and generated supports on $m$ randomly-chosen points in the space of multi-returns to avoid the exponential blowup (at the cost of lower resolution). Notably, previous theory and algorithms for categorical distributional RL do not accommodate such supports; this is a novel feature of our algorithms and theory.

---

> ### Author Response · Authors · 2024-08-11
>
> Thank you again for your thorough review and your enthusiasm for our work.
> We appreciated your thoughtful questions and have addressed them in our rebuttal.
> We wanted to check if our responses have adequately clarified your questions. If so, we would be grateful if you could consider increasing your score. If there are any remaining points you would like us to clarify, we would be eager to discuss further.

---

> ### Comment · Reviewer_wmwv · 2024-08-13
>
> Thank you for the author's response. However, after carefully reading the proof of Theorem 3, I'm not convinced by the meaning of the theorem and its proof.
>
> 1. The author claims that the free parameter is $m$, but this makes Theorem 3 seem to only propose an upper bound on MMD at a specific iteration $K = \lceil \frac{\log m}{2 \log \gamma^{-c/2}} \rceil$.
> In short, for a given $m$, the phrase "after enough $K$ iterations" does not seem valid.
> In my view, the theorem should be expressed to indicate that for any arbitrary $K$ and if there are $m = O(\exp{K})$ atoms, then an upper bound on MMD can be provided, and that as $K \rightarrow \infty$, the bound can be sufficiently reduced to approach zero.
>
> 2. The expression in Proposition 1, Line 600, seems mathematically awkward. Based on the results in Lines 599-600, the MMD as $k \rightarrow \infty$ implies $\frac{f(d,m)}{1- \gamma^{c/2}}$, but Line 600 expresses this in terms of a distribution and a ball, which has not been well-defined for MMD in this context.
> While this appears to be a minor expression issue that does not affect the result of Theorem 3, it still seems necessary to correct it.
>
> 3. It seems that $\eta$ on Line 617 should be changed to $\eta_k$, and $\eta$ on Line 621 should be changed to $\eta_K$.
>
> After reading reviewer 9dnf's question and the author's answer, the author claims to have avoided an “exponential blowup”.
> However, at this point, I do not believe that Theorem 3 successfully solves this problem.
> I think more explanation is needed from the author, and thus I decreased the score to 5.

---

> ### Author Response · Authors · 2024-08-13
>
> Thanks to the reviewer for the questions. We really appreciate your engagement and your effort reading further into the proofs.
>
> **Proposition 1, Line 600**. Thank you for pointing this out, we will clarify this. Since $\overline{\mathrm{MMD}}$ is a metric on multi-return distribution functions, we employ the usual notion of a ball in a metric space.
>
> **Line 617 and 621**. Again, thank you, these are typos. Indeed on line 617 we will fix $\eta$ to $\eta_k$ and on line 621 we will fix $\eta$ to $\eta_K$.
>
> **Blowup of number of particles**. There are two points to consider here. We can first ask how many large the representations become after a certain number of iterations. Additionally, we can ask, for a given error tolerance $\epsilon$, how large does $m$ need to be.
>
> With regard to the first question, the number of particles in Theorem 3 remains fixed over the course of all iterations. With the unprojected DP algorithm, this number will increase at an exponential rate as discussed with 9dnf. You are correct that after a finite number of iterations, we will still have finitely many particles; see our discussion for the next point that shows that this number will still be much larger than what we get with Theorem 3.
>
> The problem that Theorem 3 solves is the following: "Given a budget of $m$ particles, give me an $m$-particle EWP representation of that achieves error at most $\epsilon$". Theorem 3 says that, to accomplish this, you need
>
> \begin{align*}
> m_{\mathrm{ours}} \geq \widetilde{O}\left(\frac{1}{\epsilon^2}\frac{d^\alpha R^{2\alpha}_{\max}}{(1-\gamma^{\alpha/2})^2(1-\gamma)^{2\alpha}}\log^2\left(\frac{|\mathcal{X}|\delta^{-1}}{\log\gamma^{-\alpha/2}}\right)\right).
> \end{align*}
>
> The number of iterates $K$ is an algorithmic detail for accomplishing this in our case -- we solve this problem in polynomial time ($K$ is polynomially large).
>
> Suppose instead you try the method shown in the example of exponential blowup. You start with a 1-particle EWP representation. Note that the distributional Bellman operator is a $\gamma^{\alpha/2}$-contraction, so you'll have $\overline{\mathrm{MMD}}(\eta_k, \eta^\pi)\leq \gamma^{\alpha k/2}D$, where $D =\overline{\mathrm{MMD}}(\eta_0, \eta^\pi)$. So, if you want at most $\epsilon$ error, you need $K\geq \frac{2\log(D/\epsilon)}{\alpha\log\gamma^{-1}}$. Then, since each iteration blows up the number of particles by a factor of $|\mathcal{X}|$, this results in
>
> \begin{align*}
> m_{\mathrm{unprojected}} \geq |\mathcal{X}|^{\frac{2\log (D/\epsilon)}{\alpha\log\gamma^{-1}}}
> \end{align*}
>
> To summarize, as a function of $\epsilon$ and $\mathcal{X}$, we have $m_{\mathrm{ours}} = \widetilde{O}(\log^2(\lvert\mathcal{X}\rvert)\epsilon^{-2})$ while $m_{\mathrm{unprojected}} = O(\lvert\mathcal{X}\rvert^2\epsilon^{-2})$, which is much worse.
>
> The reason why we need to specify $K$ is because we're using randomized projections; applying too many raises the (low) probability of sampling a bad projection. We found the "just right" $K$ that avoids this with arbitrarily high probability. But again, $K$ is not to be interpreted as a user-chosen parameter here. It is an algorithmic quantity that gets us $\epsilon$-approximate return distributions for much smaller $m$ than required for unprojected DP.

---

> > ### Comment · Reviewer_wmwv · 2024-08-14
> >
> > Thank you for the detailed explanation, it has completely cleared up my misunderstanding.
> > The comparison with unprojected DP provided by the authors is quite convincing, and I would appreciate it if this could be reflected in the main text. I have no further concerns and will restore my original score of 6.

---

> > > ### Author Response · Authors · 2024-08-14
> > >
> > > We're glad to hear that the explanation has cleared everything up. Thank you very much again for your detailed reading of the technical aspects of the paper, we really appreciate it.
> > > We'll be very happy to include this discussion and comparison against the unprojected operator in the final version of the paper.
> > >
> > > We also just wanted to check in reference to the original review, you mentioned you would be willing to increase your score above 6 if your questions are addressed.
> > > As the discussion period is now drawing to an end, we'd like to ask if you would consider increasing your score in light of the discussion we've had, or if you have any further queries.

---

> > > > ### Comment · Reviewer_wmwv · 2024-08-14
> > > >
> > > > As I wrote in the weaknesses, the paper focuses on the theoretical analysis of Categorical-TD rather than EWP-TD.
> > > > Although the author's response clears up my misunderstanding of Theorem 3, I still feel that the theoretical analysis of EWP-TD is not sufficient, and I will keep the score at 6.

---

> > > > > ### Author Response · Authors · 2024-08-14
> > > > >
> > > > > Thanks again for your engagement.
> > > > >
> > > > > While we do not have enough time to query more about "theoretical analysis of EWP-TD is not sufficient", we would like to point out that traditionally theoretical analysis of EWP in distributional RL has come many years after categorical analysis due to the complexity of the projection. Indeed, the first analysis of $d=1$ projected distributional RL [RBD+18] included *no* results about EWP, with $d=1$ EWP TD analysis only coming several years later in [RMA+23] (and applied to only one particular kernel, as opposed to our work that generalizes this). Deviating from this trend, our work already does provide results for EWP in the $d>1$ setting. Establishing results for TD-learning with EWP is a major challenge due to the highly non-convex projection when $d>1$ as we discussed in section 3, which actually leads to local optima as we see in our experiments in section 6. Thus, it is not even clear that strong results *can* be proved for EWP in the TD case for $d>1$ in general, and we believe this to be a very exciting avenue for future work.

---

### Official Review · Reviewer_FNo5 · 2024-07-10

**Soundness:** 3
**Presentation:** 3
**Contribution:** 3
**Rating:** 5
**Confidence:** 3

**Summary:**

In this submission, the authors combine a multivariate reward with distributional learning. They rely on a Maximum Mean Discrepancy based projection operator to obtain the first efficient and provably convergent algorithm in this setting. The key in their work is the extension to the multivariate setting of the contraction property of the Bellman operator with respect to the MMD “metric”. They use this property in two algorithms. One based on a randomized dynamic programming which maintains the number of atoms/particles through sampling and can be proved to converge in the MDD metric. The second rely on a categorical representation for which there is a MMD compatible “projection”. Finally, they show how to make these algorithms compatible with a Temporal Difference approach. All those algorithms are illustrated with numerical experiments on toy (random MDP) examples.

**Strengths:**

- The results are new and the proofs seem correct .

- They exploit the possible dependency between the coordinates of the multivariate reward.

- The results are supported by numerical experiments.

**Weaknesses:**

- The experiments are made on very simple toy examples (random MDP). It would be interesting to see if their numerical results also hold with more realistic examples.

Typos:
- 614 \widebar{MMD} -> MMD

**Questions:**

- Could the authors comment why they did not observe in practice the convergence suggested by their theorem in the EWP-based technique?

- In Monte Carlo approximation, the price of increasing the dimensionality is often hidden in the “variance”. Here it seems that the price is moderate with a polynomial term. Do the authors have an intuition on why such a phenomenon and why it is not related to the correlation structure between the coordinates of the reward?

**Limitations:**

As mentioned by the authors, the multivariate setting can only be used in the evaluation part in MDP and Reinforcement Learning and is of interest when there are more than one possible reward, which is not the most classical setting.

---

> ### Author Rebuttal · Authors · 2024-08-06
>
> We thank the reviewer for their careful assessment of our work.
>
> > Could the authors comment on why they did not observe in practice the convergence suggested by their theorem in the EWP-based technique?
>
> We did in fact observe convergence of the EWP-based algorithms in practice -- can you please point us more specifically to where the text or results  suggest otherwise?
>
> > In Monte Carlo approximation, the price of increasing the dimensionality is often hidden in the “variance”...
>
> This is a very interesting question! We suspect that it may be possible to provide "instance-dependent" bounds that sharpen with more favorable correlation structure between reward dimensions. Our results leverage the boundedness of the reward function and the structure of strongly negative-definite kernels to provide worst-case bounds. The polynomial scaling of the bound with dimensionality is inherited from the fact that the worst-case values taken on by the MMDs we consider also scale polynomially with dimension.
>
> Regarding experimental results, please see our general response, in which we demonstrate that the signed categorical TD algorithm proposed in this paper is straightforward to combine with neural network function approximation.

---

> ### Author Response · Authors · 2024-08-11
>
> Thank you again for your thorough review and your positive comments on our theoretical results.
> In our rebuttal, we addressed your questions and provided a demonstration of a larger scale application of our algorithm as per your suggestion. We hope these additional results have helped address your concerns about the scalability and practicality our work.
> We wanted to respectfully check if you've had a chance to review our rebuttal. We're keen to understand if the new illustration results and our responses to your questions have adequately addressed your concerns. If so, we would be grateful if you could consider increasing your score. If there are any remaining points you'd like us to clarify, we would be eager to discuss further.

---

> > ### Comment · Reviewer_FNo5 · 2024-08-12
> >
> > I am grateful for your response and the supplementary experiment, and I will maintain my score.

---

### Official Review · Reviewer_SWh2 · 2024-07-13

**Soundness:** 3
**Presentation:** 2
**Contribution:** 3
**Rating:** 5
**Confidence:** 2

**Summary:**

This paper studied the multivariate distributional reinforcement learning (RL) problem, in which the goal is to learn probability distribution of accumulated multi-dimensional rewards of the RL system. First, dynamic programming for multivariate distributional dynamic programming is established, then a randomized particle-based dynamic programming solution method is proposed. Furthermore, a projection-based dynamic programming algorithm is proposed for categorical multivariate distributional. Some extensions to RL setting were discussed and additional empirical observations are provided.

**Strengths:**

This paper studied the fundamental problem of multivariate distributional dynamic programming and RL problems. Furthermore, convergence results have been achieved for both settings under maximum mean discrepancy distance.

**Weaknesses:**

* Some important notations are not defined for better understanding. For example, 1) the right-hand-side (RHS) of the first equation in Eq. (2) is not clearly explained or defined; 2) the RHS of R(x) in Line 187 and the $\Delta$ notation on Line 188 are not explained or defined. This writing style makes it difficult to understand many technical details of the paper.
* Typo: Equation (10), $X’_r$ -> $X’_t$

**Questions:**

* What does the right-hand-side (RHS) of the first equation in Eq. (2) mean?
* What does the right-hand-side (RHS) of the RHS of R(x) in Line 187? Please explain all the terms including, $\xi$, $N(x)$ and index $i$.
* Define $\Delta$ notation on Line 188.

**Limitations:**

See the weaknesses.

---

> ### Author Rebuttal · Authors · 2024-08-06
>
> We thank the reviewer for their feedback on the paper. We address the queries on notation below, and will include clarifications in the revised draft.
>
> * The RHS of equation (2) is simply the set of all empirical distributions on $m$ points. That is, the distributions obtained by picking $m$ points $\theta_i\in\mathbf{R}^d$ and constructing the distributions supported on those points with equal mass. This is often referred to as a $m$-quantile representation in the $d=1$ case (see e.g. [BDR23]) and is equivalent to the model of [ZCZ+21].
> * The object $\mathcal{R}$ in line 187 is defining a *state-dependent* support map. That is, for any input state $x$, $\mathcal{R}(x)$ outputs a finite set of support points $\{\xi_i(x)\}_{i=1}^{N(x)}\subset\mathbf{R}^d$. In the $d=1$ case, these would be the locations of the atoms (bins) of the categorical return distributions (see e.g. [BDM17b]) -- we generalize the notion here to have categorical supports that can depend on the state. The quantities $N(x)$ describe how many support points are in the categorical support of state $x$ -- indeed, we additionally generalize the model beyond that of [BDM17b] to accommodate state-dependent *resolution* of categorical supports. Thus, at state $x$, our model consists of $N(x)$ support points in $\mathbf{R}^d$, and these are indexed by $i$ in the equation in question.
> * The $\Delta$ on line 188 represents a simplex. In particular, $\Delta_A$ for a finite set $A$ is the set of probability mass functions on $A$.
>
> Thank you for pointing out these clarity issues, and we will clarify these in the revised draft. Should you have any other questions, we would be happy to discuss further.

---

> > ### Comment · Reviewer_SWh2 · 2024-08-11
> > **Response to rebuttal**
> >
> > The reviewer would like to thank the authors on their efforts for rebuttal and new empirical results. I have two more questions for a better understanding of the paper.
> >
> > 1. One of the reasons that the randomized projection is proposed due to the fact that original projection in (4) may not have a fixed point after applying Bellman operator. Leaving aside the computational issue, I am wondering that does the original projection may have certain statistical concentration near $\eta^{\pi}$ like in Theorem 3, either empirically at a small-size problem or at an intuitive level?
> >
> > 2. What are the key challenges of establishing a finite-time convergence results under TD setting in comparison with under dynamic programming setting?

---

> > > ### Author Response · Authors · 2024-08-11
> > >
> > > Thanks to the reviewer for their response.
> > >
> > > **Q1**: This is a nice question. If we could replace the randomized projection with the exact projection, we would achieve a similar convergence bound as Theorem 3. This can be seen from Proposition 1 (Appendix B), which shows that the concentration is controlled by how close $\Pi\eta$ is to $\eta$, where $\Pi$ is a projection onto the space of EWP representations. Since the exact projection is the "best" such projection (in the sense that it brings $\eta$ to the closest EWP representation), Proposition 1 asserts ensures that the concentration of exact projection DP algorithm iterates to $\eta^\pi$ can be no worse than the bound of Theorem 3. In particular, the exact projection would assert that the projected distributions are at least as good as those under the event $\mathcal{E}$ on line 614, so we immediately get a bound of $O(\frac{d^{\alpha/2}R^\alpha_{\max}}{(1-\gamma^{c/2})(1-\gamma)^\alpha\sqrt{m}})$.
> > >
> > > The main reason we do not apply the exact projection is that it is computationally prohibitive, but Theorem 3 asserts that the randomized projection is a good substitute.
> > >
> > > Having said that, we believe your question and its answer via Proposition 1 are interesting and important, and we will gladly include them in the main body of our revision.
> > >
> > > **Q2**: The main challenge is that, with TD, we can never apply the exact operator we're interested in -- rather, we can only apply a noisy update based on transition samples. Since the iterates of the return distribution function evolve as a dynamical system, we are therefore trying to track a dynamical system with a noisy stochastic version, which opens up the possibility of quick divergence. The standard method for ensuring this does not happen involves showing that the TD update is equal to a DP update in expectation and demonstrating a bound on the variance of the noise in the TD updates.
> > >
> > > In the particular case of our work, the usual technique for preventing this fails.
> > > Namely, since the standard categorical projection does not commute with the expectation over transition samples, we cannot even show that a TD update is equal to a DP update in expectation. To handle this, we had to introduce the projection onto the set of signed measures, which *does* commute with the expectation. However, analysis of signed measure representations of return distributions has its own complications; for instance, signed measures (even with total mass 1) may not be bounded on individual measurable sets, unlike probability measures. Subsequently, it remained to show that performing TD in the space of signed measures allows us to still project the outcome onto the space of probability measures and get a close approximation to $\eta^\pi$.
> > >
> > > Notably, this is not necessary in the case of dynamic programming, since with DP we are not estimating the mixture over next-state return distributions with transition samples. As such, it doesn't matter that the projection and the expectation do not commute, since in the DP case we can compute the mixture over next-state return distributions exactly and then apply the projection.

---

> ### Author Response · Authors · 2024-08-11
>
> Thank you for taking the time to review our submission. We appreciate your feedback regarding the notation used in our paper. We have addressed your questions in our rebuttal and hope that our clarifications have been helpful.
> If there are any other aspects that you would like further clarification on, we would be more than happy to discuss further; please let us know. If we have addressed all of your concerns, we would be grateful if you could consider increasing your score.

---

> > ### Comment · Area_Chair_Ecuu · 2024-08-11
> >
> > Dear Reviewer SWh2, are there other concerns you have about this paper that you wish to raise with the authors? Your main concern seemed that the writing lacks important technical details. Could you add what details are essential for the paper's claims and supporting evidence to be more clear?
> >
> > Thanks,
> > Your AC

---

> ### Comment · Reviewer_SWh2 · 2024-08-12
> **Thanks for the response**
>
> Once again, thanks for the efforts on rebuttal. I have adjusted my score. However, I do want to emphasize to the authors on the clarity of definitions, notations and motivations(in particular, categorical representation section) in the future revisions.

---

> > ### Author Response · Authors · 2024-08-14
> >
> > Thank you very much for your engagement and the constructive discussion. We will definitely expand more on these points (as well as the points we discussed above) in the final version of the paper.

---

### Author Rebuttal · Authors · 2024-08-06

# General Response
We thank all authors for their assessments.

Reviewers praised our convergence theory (SWh2, FNo5, wmwv), rigorous proofs and discussions (FNo5, wmwv, 9dnf), and illustrative experiments (FNo5, wmwv); while the simplicity of the numerical experiments and the motivation for formalizing multivariate distributional RL were commented on by FNo5 and 9dnf.

To address comments about scalability and motivation, we illustrate in our rebuttal PDF that our Signed Categorical TD method can be scaled to large (pixel-based) state spaces, just by directly representing the multi-return distribution function with a deep neural network trained with gradient descent for the update rule of equation (12). Our illustration also demonstrates how learning distributional successor features (multivariate return distributiosn) can be very useful in practice.

Further details about our experiment setup are given below.

## Illustration Details
Henceforth, we refer to figures in the rebuttal PDF.
Our environment consists of a car navigating to a parking spot (see Figure 1 for a depiction of an observation in this environment). The reward function is two-dimensional, with dimens
1. **Lateral feature**: The $x$ coordinate of the car.
2. **Parking feature**: A sparse reward that is $1$ when the car is parked in the correct location, and $0$ otherwise.

We train the multivariate return distribution function from pixels using a convolutional neural network (architecture shown in Figure 2 of the rebuttal document) from data generated by trajectories that circumvent the obstacle in the middle of the map. Note that, by symmetry, the (expected) successor features for such data would be roughly $0$ in the lateral feature, which is impossible to distinguish from a policy that drives straight through the obstacle.

Figures 3 and 4 show the learned multivariate return distributions (e.g., distributional successor features) from the method of [ZCZ+21] and ours, respectively. In both cases, the distribution provides crucial information for distinguishing the policy from one that drives through the obstacle (unlike if we learned SFs). Both algorithms produce multimodal distributions depicting trajectories that circumvented the obstacle on either side. However, our approach based on signed categorical TD has two major advantages:
1. Unlike the local optimum found by the EWP algorithm, its probability mass is roughly contained in the support of the true multi-return distribution;
2. It is based on an algorithm whose convergence is well understood (as we prove in this paper).

Figure 5 quantitatively demonstrates the accuracy of the multivariate return distributions learned by our algorithm and that of [ZCZ+21]. Here we examine two held-out projections on the space of multivariate returns (corresponding to held-out scalar reward functions) and evaluate the Cramer error of the projected return distributions relative to Monte Carlo estimates. These two projections correspond to a diverse pair of objectives: on the left, we incentivize paths circumventing the obstacle conservatively on the left (with a large negative lateral component) and on the right we incentivize "riskier" paths that get to the parking spot quickly (with a large positive parking component and near-zero lateral component). In both cases, both algorithms achieve remarkably low Cramer error. In the left case, our method achieves lower Cramer error by a non-negligible margin.

## Utility of Multivariate Distributional RL

This illustration also motivates multivariate distributional RL as a tool for inverse RL. Many inverse RL methods aim to match successor features to a demonstration, however in this case, this would result in learning a policy that drives through the obstacle. Matching distributional SFs (using multivariate distributional RL) would prevent such behavior.

Beyond this, a formal understanding of multivariate distributional RL can provide insights for zero-shot risk-sensitive RL: the problem of predicting arbitrary statistics of return distributions for arbitrary reward functions without further training. This concept was recently explored by [WFG+24] through the distributional successor measure (DSM), but much like [ZCZ+21], the convergence of their practical algorithm was not analyzed. Our results in fact immediately provide convergence guarantees for learning the DSM in tabular MDPs, and we demonstrate this further in Section 5.1.

---

### Decision · Program_Chairs · 2024-09-25

**Decision:**

Accept (poster)

**Comment:**

Brief summary: this paper provides convergence analysis for policy evaluation in distributional RL with multiple reward functions. There are k reward functions and the task is to predict the distribution of the returns for all k and from all states when running a fixed policy (evaluation setting). The paper derives convergence rates for distributional dynamic programming and then extends to temporal-difference methods. Theoretical results are complemented with simple experiments.

The paper was extensively debated and the point of discussion ultimately came down to the subjective point of research taste in terms of how significant the theoretical contribution of the paper is w.r.t. prior work. All reviewers agreed that the paper is sound and the contributions are novel. All reviewers also agree that the empirical study presented is not a major contribution.

After reading the reviews, discussion, paper, and most closely related paper, I believe the paper makes a non-trivial contribution to the foundations of distributional RL in terms of convergence analysis of both dynamic programming and TD-operators. Consequently, I think it is preferable to accept the paper.